# Optimistic Active Exploration of Dynamical Systems

**Bhavya Sukhija**[1]    **Lenart Treven**[1]    **Cansu Sancaktar**[2]    **Sebastian Blaes**[2]
**Stelian Coros**[1] **Andreas Krause**[1]
ETH Zürich [1]    MPI for Intelligent Systems[2]
{sukhijab,trevenl,scoros,krausea}@ethz.ch
{cansu.sancaktar,sebastian.blae}@tuebingen.mpg.de

## Abstract

Reinforcement learning algorithms commonly seek to optimize policies for solving one particular task. How should we explore an unknown dynamical system such that the estimated model globally approximates the dynamics and allows us to solve multiple downstream tasks in a zero-shot manner? In this paper, we address this challenge, by developing an algorithm – OPAX– for active exploration. OPAX uses well-calibrated probabilistic models to quantify the epistemic uncertainty about the unknown dynamics. It optimistically—w.r.t. to plausible dynamics—maximizes the information gain between the unknown dynamics and state observations. We show how the resulting optimization problem can be reduced to an optimal control problem that can be solved at each episode using standard approaches. We analyze our algorithm for general models, and, in the case of Gaussian process dynamics, we give a first-of-its-kind sample complexity bound and show that the epistemic uncertainty *converges to zero*. In our experiments, we compare OPAX with other heuristic active exploration approaches on several environments. Our experiments show that OPAX is not only theoretically sound but also performs well for zero-shot planning on novel downstream tasks.

## 1   Introduction

Most reinforcement learning (RL) algorithms are designed to maximize cumulative rewards for a single task at hand. Particularly, model-based RL algorithms, such as (Chua et al., 2018; Kakade et al., 2020; Curi et al., 2020), excel in efficiently exploring the dynamical system as they direct the exploration in regions with high rewards. However, due to the directional bias, their underlying learned dynamics model fails to generalize in other areas of the state-action space. While this is sufficient if only one control task is considered, it does not scale to the setting where the system is used to perform several tasks, i.e., under the same dynamics optimized for different reward functions. As a result, when presented with a new reward function, they often need to relearn a policy from scratch, requiring many interactions with the system, or employ multi-task (Zhang and Yang, 2021) or transfer learning (Weiss et al., 2016) methods. Traditional control approaches such as trajectory optimization (Biagiotti and Melchiorri, 2008) and model-predictive control (García et al., 1989) assume knowledge of the system's dynamics. They leverage the dynamics model to solve an optimal control problem for each task. Moreover, in the presence of an accurate model, important system properties such as stability and sensitivity can also be studied. Hence, knowing an accurate dynamics model bears many practical benefits. However, in many real-world settings, obtaining a model using just physics' first principles is very challenging. A promising approach is to leverage data for learning the dynamics, i.e., system identification or active learning. To this end, the key question we investigate in this work is: *how should we interact with the system to learn its dynamics efficiently?*

While active learning for regression and classification tasks is well-studied, active learning in RL is much less understood. In particular, active learning methods that yield strong theoretical and practical results, generally query data points based on information-theoretic criteria (Krause et al., 2008; Settles, 2009; Balcan et al., 2010; Hanneke et al., 2014; Chen et al., 2015). In the context of

dynamical systems, this requires querying arbitrary transitions (Berkenkamp et al., 2017; Mehta et al., 2021). However, in most cases, *querying a dynamical system at any state-action pair is unrealistic*. Rather, we can only execute policies on the real system and observe the resulting trajectories. Accordingly, an active learning algorithm for RL needs to suggest policies that are "informative" for learning the dynamics. This is challenging since it requires planning with unknown dynamics.

**Contributions**  In this paper, we introduce a new algorithm, *Optimistic Active eXploration (*OPAX*)*, designed to actively learn nonlinear dynamics within continuous state-action spaces. During each episode, OPAX plans an exploration policy to gather the most information possible about the system. It learns a statistical dynamics model that can quantify its epistemic uncertainty and utilizes this uncertainty for planning. The planned trajectory targets state-action pairs where the model's epistemic uncertainty is high, which naturally encourages exploration. In light of unknown dynamics, OPAX uses an optimistic planner that picks policies that optimistically yield maximal information. We show that this *optimism paradigm* plays a crucial role in studying the theoretical properties of OPAX. Moreover, we provide a general convergence analysis for OPAX and prove convergence to the true dynamics for Gaussian process (GP) dynamics models. Theoretical guarantees for active learning in RL exist for a limited class of systems (Simchowitz et al., 2018; Wagenmaker and Jamieson, 2020; Mania et al., 2020), but lack for a more general and practical class of dynamics (Chakraborty et al., 2023; Wagenmaker et al., 2023). We are, to the best of our knowledge, the first to give convergence guarantees for a rich class of nonlinear dynamical systems.

We evaluate OPAX on several simulated robotic tasks with state dimensions ranging from two to 58. The empirical results provide validation for our theoretical conclusions, showing that OPAX consistently delivers strong performance across all tested environments. Finally, we provide an efficient implementation[1] of OPAX in JAX (Bradbury et al., 2018).

## 2 Problem Setting

We study an *unknown* discrete-time dynamical system $\boldsymbol{f}^*$, with state $\boldsymbol{x} \in \mathcal{X} \subset \mathbb{R}^{d_x}$ and control inputs $\boldsymbol{u} \in \mathcal{U} \subset \mathbb{R}^{d_u}$.

$$\boldsymbol{x}_{k+1} = \boldsymbol{f}^*(\boldsymbol{x}_k, \boldsymbol{u}_k) + \boldsymbol{w}_k. \tag{1}$$

Here, $\boldsymbol{w}_k$ represents the stochasticity of the system for which we assume $\boldsymbol{w}_k \overset{i.i.d.}{\sim} \mathcal{N}\left(\boldsymbol{0}, \sigma^2 \boldsymbol{I}\right)$ (Assumption 3). Most common approaches in control, such as trajectory optimization and model-predictive control (MPC), assume that the dynamics model $\boldsymbol{f}^*$ is known and leverages the model to control the system state. Given a cost function $c : \mathcal{X} \times \mathcal{U} \to \mathbb{R}$, such approaches formulate and solve an optimal control problem to obtain a sequence of control inputs that drive the system's state

$$\underset{\boldsymbol{u}_{0:T-1}}{\arg\min} \, \mathbb{E}_{\boldsymbol{w}_{0:T-1}} \left[ \sum_{t=0}^{T-1} c(\boldsymbol{x}_t, \boldsymbol{u}_t) \right], \tag{2}$$

$$\boldsymbol{x}_{t+1} = \boldsymbol{f}^*(\boldsymbol{x}_t, \boldsymbol{u}_t) + \boldsymbol{w}_t \quad \forall 0 \leq t \leq T.$$

Moreover, if the dynamics are known, many important characteristics of the system such as stability, and robustness (Khalil, 2015) can be studied. However, in many real-world scenarios, an accurate dynamics model $\boldsymbol{f}^*$ is not available. Accordingly, in this work, we consider the problem of actively learning the dynamics model from data a.k.a. system identification (Åström and Eykhoff, 1971). Specifically, we are interested in devising a cost-agnostic algorithm that focuses solely on learning the dynamics model. Once a good model is learned, it can be used for solving different downstream tasks by varying the cost function in Equation (2).

We study an episodic setting, with episodes $n = 1, \ldots, N$. At the beginning of the episode $n$, we deploy an exploratory policy $\boldsymbol{\pi}_n$, chosen from a policy space $\Pi$ for a horizon of $T$ on the system. Next, we obtain trajectory $\boldsymbol{\tau}_n = (\boldsymbol{x}_{n,0}, \ldots, \boldsymbol{x}_{n,T})$, which we save to a dataset of transitions $\mathcal{D}_n = \{(\boldsymbol{z}_{n,i} = (\boldsymbol{x}_{n,i}, \boldsymbol{\pi}_n(\boldsymbol{x}_{n,i})), \boldsymbol{y}_{n,i} = \boldsymbol{x}_{n,i+1})_{0 \leq i < T}\}$. We use the collected data to learn an estimate $\boldsymbol{\mu}_n$ of $\boldsymbol{f}^*$. To this end, the goal of this work is to propose an algorithm *Alg*, that at each episode $n$ leverages the data acquired thus far, i.e., $\mathcal{D}_{1:n-1}$ to determine a policy $\boldsymbol{\pi}_n \in \Pi$ for the next step of data collection, that is, $Alg(\mathcal{D}_{1:n-1}, n) \to \boldsymbol{\pi}_n$. The proposed algorithm should be consistent, i.e., $\boldsymbol{\mu}_n(\boldsymbol{z}) \to \boldsymbol{f}^*(\boldsymbol{z})$ for $n \to \infty$ for all $\boldsymbol{z} \in \mathcal{R}$, where $\mathcal{R}$ is the reachability set defined as

$$\mathcal{R} = \{\boldsymbol{z} \in \mathcal{Z} \mid \exists (\boldsymbol{\pi} \in \Pi, t \leq T), \text{ s.t.}, p(\boldsymbol{z}_t = \boldsymbol{z} | \boldsymbol{\pi}, \boldsymbol{f}^*) > 0\},$$

---

[1] https://github.com/lasgroup/opax

and efficient w.r.t. rate of convergence of $\boldsymbol{\mu}_n$ to $\boldsymbol{f}^*$.

To devise such an algorithm, we take inspiration from Bayesian experiment design (Chaloner and Verdinelli, 1995). In the Bayesian setting, given a prior over $\boldsymbol{f}^*$, a natural objective for active exploration is the mutual information (Lindley, 1956) between $\boldsymbol{f}^*$ and observations $\boldsymbol{y}_{\mathcal{D}_n}$.

**Definition 1** (Mutual Information, Cover and Thomas (2006)). *The mutual information between $\boldsymbol{f}^*$ and its noisy measurements $\boldsymbol{y}_{\mathcal{D}_n}$ for points in $\mathcal{D}_n$, where $\boldsymbol{y}_{\mathcal{D}_n}$ is the concatenation of $(\boldsymbol{y}_{\mathcal{D}_n,i})_{i<T}$ is defined as,*

$$F(\mathcal{D}_n) := I\left(\boldsymbol{f}^*; \boldsymbol{y}_{\mathcal{D}_n}\right) = H\left(\boldsymbol{y}_{\mathcal{D}_n}\right) - H\left(\boldsymbol{y}_{\mathcal{D}_n} \mid \boldsymbol{f}^*\right), \tag{3}$$

*where $H$ is the Shannon differential entropy.*

The mutual information quantifies the reduction in entropy of $\boldsymbol{f}^*$ conditioned on the observations. Hence, maximizing the mutual information w.r.t. the dataset $\mathcal{D}_n$ leads to the maximal entropy reduction of our prior. Accordingly, a natural objective for active exploration in RL can be the mutual information between $\boldsymbol{f}^*$ and the collected transitions over a budget of $N$ episodes, i.e., $I\left(\boldsymbol{f}^*; \boldsymbol{y}_{\mathcal{D}_{1:N}}\right)$. This requires maximizing the mutual information over a sequence of policies, which is a challenging planning problem even in settings where the dynamics are known (Mutny et al., 2023). A common approach is to greedily pick a policy that maximizes the information gain conditioned on the previous observations at each episode:

$$\max_{\boldsymbol{\pi} \in \Pi} \mathbb{E}_{\tau^{\boldsymbol{\pi}}} \left[ I\left(\boldsymbol{f}^*_{\tau^{\boldsymbol{\pi}}}; \boldsymbol{y}_{\tau^{\boldsymbol{\pi}}} \mid \mathcal{D}_{1:n-1}\right) \right]. \tag{4}$$

Here $\boldsymbol{f}^*_{\tau^{\boldsymbol{\pi}}} = (\boldsymbol{f}^*(\boldsymbol{z}_{n,0}), \ldots, \boldsymbol{f}^*(\boldsymbol{z}_{n,T-1}))$, $\boldsymbol{y}_{\tau^{\boldsymbol{\pi}}} = (\boldsymbol{y}_{n,0}, \ldots, \boldsymbol{y}_{n,T-1})$, $\tau^{\boldsymbol{\pi}}$ is the trajectory under the policy $\boldsymbol{\pi}$, and the expectation is taken w.r.t. the process noise $\boldsymbol{w}$.

**Interpretation in frequentist setting**  While information gain is Bayesian in nature (requires a prior over $\boldsymbol{f}^*$), it also has a frequentist interpretation. In particular, later in Section 3 we relate it to the epistemic uncertainty of the learned model. Accordingly, while this notion of information gain stems from Bayesian literature, we can use it to motivate our objective in both Bayesian and frequentist settings.

## 2.1  Assumptions

In this work, we learn a probabilistic model of the function $\boldsymbol{f}^*$ from data. Moreover, at each episode $n$, we learn the mean estimator $\boldsymbol{\mu}_n(\boldsymbol{x}, \boldsymbol{u})$ and the epistemic uncertainty $\boldsymbol{\sigma}_n(\boldsymbol{x}, \boldsymbol{u})$, which quantifies our uncertainty on the mean prediction. To this end, we use Bayesian models such as Gaussian processes (GPs, Rasmussen and Williams, 2005) or Bayesian neural networks (BNNs, Wang and Yeung, 2020). More generally, we assume our model is *well-calibrated*:

**Definition 2** (All-time calibrated statistical model of $\boldsymbol{f}^*$, Rothfuss et al. (2023)). *Let, $\boldsymbol{z} = (\boldsymbol{x}, \boldsymbol{u})$ and $\mathcal{Z} := \mathcal{X} \times \mathcal{U}$. An all-time calibrated statistical model of the function $\boldsymbol{f}^*$ is a sequence $(\boldsymbol{\mu}_n, \boldsymbol{\sigma}_n, \beta_n(\delta))_{n \geq 0}$, such that*

$$\Pr\left(\forall \boldsymbol{z} \in \mathcal{Z}, \forall l \in \{1, \ldots, d_x\}, \forall n \in \mathbb{N} : |\mu_{n,l}(\boldsymbol{z}) - f_l(\boldsymbol{z})| \leq \beta_n(\delta)\sigma_{n,l}(\boldsymbol{z})\right) \geq 1 - \delta$$

*Here $\mu_{n,l}$ and $\sigma_{n,l}$ are the $l$-th element in the vector valued functions $\boldsymbol{\mu}_n$ and $\boldsymbol{\sigma}_n$ respectively. The scalar function, $\beta_n(\delta) \in \mathbb{R}_{\geq 0}$ quantifies the width of the $1 - \delta$ confidence intervals. We assume w.l.o.g. that $\beta_n$ monotonically increases with $n$, and that $\sigma_{n,l}(\boldsymbol{z}) \leq \sigma_{\max}$ for all $\boldsymbol{z} \in \mathcal{Z}$, $n \geq 0$, and $l \in \{1, \ldots, d_x\}$.*

**Assumption 1** (Well calibration assumption). *Our learned model is an all-time-calibrated statistical model of $\boldsymbol{f}^*$, i.e., there exists a sequence of $(\beta_n(\delta))_{n \geq 0}$ such that our model satisfies the well-calibration condition, c.f., Definition 2.*

This is a natural assumption on our modeling. It states that we can make a mean prediction and also quantify how far it is off from the true one with high probability. A GP model satisfies this requirement for a very rich class of functions, c.f., Lemma 3. For BNNs, calibration methods (Kuleshov et al., 2018) are often used and perform very well in practice. Next, we make a simple continuity assumption on our function $\boldsymbol{f}^*$.

**Assumption 2** (Lipschitz Continuity). *The dynamics model $\boldsymbol{f}^*$ and our epistemic uncertainty prediction $\boldsymbol{\sigma}_n$ are $L_{\boldsymbol{f}}$ and $L_{\boldsymbol{\sigma}}$ Lipschitz continuous, respectively. Moreover, we define $\Pi$ to be the policy class of $L_{\boldsymbol{\pi}}$ Lipschitz continuous functions.*

The Lipschitz continuity assumption on $\boldsymbol{f}^*$ is quite common in control theory (Khalil, 2015) and learning literature (Curi et al., 2020; Pasztor et al., 2021; Sussex et al., 2023). Furthermore, the Lipschitz continuity of $\boldsymbol{\sigma}_n$ also holds for GPs with common kernels such as the linear or radial basis function (RBF) kernel (Rothfuss et al., 2023).

Finally, we reiterate the assumption of the system's stochasticity.

**Assumption 3** (Process noise distribution). *The process noise is i.i.d. Gaussian with variance* $\sigma^2$, *i.e.,* $\boldsymbol{w}_k \overset{i.i.d}{\sim} \mathcal{N}(\boldsymbol{0}, \sigma^2 \boldsymbol{I})$.

We focus on the setting where $\boldsymbol{w}$ is homoscedastic for simplicity. However, our framework can also be applied to the more general heteroscedastic and sub-Gaussian case (c.f., Theorem 2).

## 3 Optimistic Active Exploration

In this section, we propose our *optimistic active exploration* (OPAX) algorithm. The algorithm consists of two main contributions: *(i)* First we reformulate the objective in Equation (4) to a simple optimal control problem, which suggests policies that visit states with high epistemic uncertainty. *(ii)* We leverage the optimistic planner introduced by Curi et al. (2020) to efficiently plan a policy under *unknown* dynamics. Moreover, we show that the optimistic planner is crucial in giving theoretical guarantees for the algorithm.

### 3.1 Optimal Exploration Objective

The objective in Equation (4) is still difficult and expensive to solve in general. However, since in this work, we consider Gaussian noise, c.f., Assumption 3, we can simplify this further.

**Lemma 1** (Information gain is upper bounded by sum of epistemic uncertainties). *Let* $\boldsymbol{y} = \boldsymbol{f}^*(\boldsymbol{z}) + \boldsymbol{w}$, *with* $\boldsymbol{w} \sim \mathcal{N}(0, \sigma^2 \boldsymbol{I})$ *and let* $\boldsymbol{\sigma}_{n-1}$ *be the epistemic uncertainty after episode* $n-1$. *Then the following holds for all* $n \geq 1$ *and dataset* $\mathcal{D}_{1:n-1}$,

$$I\left(\boldsymbol{f}_{\boldsymbol{\tau}^\pi}^*; \boldsymbol{y}_{\boldsymbol{\tau}^\pi} \mid \mathcal{D}_{1:n-1}\right) \leq \frac{1}{2} \sum_{t=0}^{T-1} \sum_{j=1}^{d_x} \log\left(1 + \frac{\sigma_{n-1,j}^2(\boldsymbol{z}_t)}{\sigma^2}\right). \tag{5}$$

We prove Lemma 1 in Appendix A. The information gain is non-negative (Cover and Thomas, 2006). Therefore, if the right-hand side of Equation (5) goes to zero, the left-hand side goes to zero as well. Lemma 1 relates the information gain to the model epistemic uncertainty. Therefore, it gives a tractable objective that also has a frequentist interpretation - collect points with the highest epistemic uncertainty. We can use it to plan a trajectory at each episode $n$, by solving the following optimal control problem:

$$\boldsymbol{\pi}_n^* = \operatorname*{argmax}_{\boldsymbol{\pi} \in \Pi} J_n(\boldsymbol{\pi}) = \operatorname*{argmax}_{\boldsymbol{\pi} \in \Pi} \mathbb{E}_{\boldsymbol{\tau}^\pi}\left[\sum_{t=0}^{T-1} \sum_{j=1}^{d_x} \log\left(1 + \frac{\sigma_{n-1,j}^2(\boldsymbol{x}_t, \boldsymbol{\pi}(\boldsymbol{x}_t))}{\sigma^2}\right)\right], \tag{6}$$

$$\boldsymbol{x}_{t+1} = \boldsymbol{f}^*(\boldsymbol{x}_t, \boldsymbol{\pi}(\boldsymbol{x}_t)) + \boldsymbol{w}_t.$$

The problem in Equation (6) is closely related to previous literature in active exploration for RL. For instance, some works consider different geometries such as the sum of epistemic uncertainties (Pathak et al. (2019); Sekar et al. (2020), c.f., appendix C for more detail).

### 3.2 Optimistic Planner

The optimal control problem in Equation (6) requires knowledge of the dynamics $\boldsymbol{f}^*$ for planning, however, $\boldsymbol{f}^*$ is unknown. A common choice is to use the mean estimator $\boldsymbol{\mu}_{n-1}$ in Equation (6) instead of $\boldsymbol{f}^*$ for planning (Buisson-Fenet et al., 2020). However, in general, using the mean estimator is susceptible to model biases (Chua et al., 2018) and is provably optimal only in the case of linear systems (Simchowitz and Foster, 2020). To this end, we propose using an optimistic planner, as suggested in Curi et al. (2020), instead. Accordingly, given the mean estimator $\boldsymbol{\mu}_{n-1}$ and the epistemic uncertainty $\boldsymbol{\sigma}_{n-1}$, we solve the following optimal control problem

$$\boldsymbol{\pi}_n, \boldsymbol{\eta}_n = \operatorname*{argmax}_{\boldsymbol{\pi} \in \Pi, \boldsymbol{\eta} \in \Xi} J_n(\boldsymbol{\pi}, \boldsymbol{\eta}) = \operatorname*{argmax}_{\boldsymbol{\pi} \in \Pi, \boldsymbol{\eta} \in \Xi} \mathbb{E}_{\boldsymbol{\tau}^{\pi, \eta}}\left[\sum_{t=0}^{T-1} \sum_{j=1}^{d_x} \log\left(1 + \frac{\sigma_{n-1,j}^2(\hat{\boldsymbol{x}}_t, \boldsymbol{\pi}(\hat{\boldsymbol{x}}_t))}{\sigma^2}\right)\right], \tag{7}$$

$$\hat{\boldsymbol{x}}_{t+1} = \boldsymbol{\mu}_{n-1}(\hat{\boldsymbol{x}}_t, \boldsymbol{\pi}(\hat{\boldsymbol{x}}_t)) + \beta_{n-1}(\delta)\boldsymbol{\sigma}_{n-1}(\hat{\boldsymbol{x}}_t, \boldsymbol{\pi}(\hat{\boldsymbol{x}}_t))\boldsymbol{\eta}(\hat{\boldsymbol{x}}_t) + \boldsymbol{w}_t,$$

**Init:** Aleatoric uncertainty $\sigma$, Probability $\delta$, Statistical model $(\boldsymbol{\mu}_0, \boldsymbol{\sigma}_0, \beta_0(\delta))$
**for** episode $n = 1, \ldots, N$ **do**

$$\boldsymbol{\pi}_n = \underset{\boldsymbol{\pi} \in \Pi}{\arg\max} \ \underset{\boldsymbol{\eta} \in \Xi}{\max} \ \mathbb{E} \left[ \sum_{t=0}^{T-1} \sum_{j=1}^{d_x} \log \left( 1 + \frac{\sigma_{n-1,j}^2(\boldsymbol{x}_t, \boldsymbol{\pi}(\boldsymbol{x}_t))}{\sigma^2} \right) \right] \qquad \blacktriangleright \text{Prepare policy}$$

$\mathcal{D}_n \leftarrow \text{ROLLOUT}(\boldsymbol{\pi}_n)$         $\blacktriangleright$ Collect measurements

Update $(\boldsymbol{\mu}_n, \boldsymbol{\sigma}_n, \beta_n(\delta)) \leftarrow \mathcal{D}_{1:n}$         $\blacktriangleright$ Update model

where $\Xi$ is the space of policies $\boldsymbol{\eta} : \mathcal{X} \to [-1, 1]^{d_x}$. Therefore, we use the policy $\boldsymbol{\eta}$ to "hallucinate" (pick) transitions that give us the most information. Overall, the resulting formulation corresponds to a simple optimal control problem with a larger action space, i.e., we increase the action space by another $d_x$ dimension. A natural consequence of Assumption 1 is that $J_n(\boldsymbol{\pi}_n^*) \leq J_n(\boldsymbol{\pi}_n, \boldsymbol{\eta}_n)$ with high probability (c.f., Corollary 1 in Appendix A). That is by solving Equation (7), we get an optimistic estimate on Equation (6). Intuitively, the policy $\boldsymbol{\pi}_n$ that OPAX suggests, behaves optimistically with respect to the information gain at each episode.

## 4 Theoretical Results

We theoretically analyze the convergence properties of OPAX. We first study the regret of planning under unknown dynamics. Specifically, since we cannot evaluate the optimal exploration policy from eq. (6) and use the optimistic one, i.e., eq. (7) instead, we incur a regret. We show that due to the optimism in the face of uncertainty paradigm, we can give sample complexity bounds for the Bayesian and frequentist settings. All the proofs are presented in Appendix A.

**Lemma 2** (Regret of optimistic planning under unknown dynamics). *Let Assumption 1 hold. Furthermore, define $J_{n,k}(\boldsymbol{\pi}_n, \boldsymbol{\eta}_n, \boldsymbol{x})$ as*

$$J_{n,k}(\boldsymbol{\pi}_n, \boldsymbol{\eta}_n, \boldsymbol{x}) = \mathbb{E}_{\boldsymbol{\tau}^{\boldsymbol{\pi}_n, \boldsymbol{\eta}_n}} \left[ \sum_{t=k}^{T-1} \sum_{j=1}^{d_x} \log \left( 1 + \frac{\sigma_{n-1,j}^2(\hat{\boldsymbol{x}}_t, \boldsymbol{\pi}_n(\hat{\boldsymbol{x}}_t))}{\sigma^2} \right) \right],$$

$$\text{s.t. } \hat{\boldsymbol{x}}_{t+1} = \boldsymbol{\mu}_{n-1}(\hat{\boldsymbol{x}}_t, \boldsymbol{\pi}_n(\hat{\boldsymbol{x}}_t)) + \beta_{n-1}(\delta) \boldsymbol{\sigma}_{n-1}(\hat{\boldsymbol{x}}_t, \boldsymbol{\pi}_n(\hat{\boldsymbol{x}}_t)) \boldsymbol{\eta}_n(\hat{\boldsymbol{x}}_t) + \boldsymbol{w}_t$$

$$\text{and } \hat{\boldsymbol{x}}_0 = \boldsymbol{x}.$$

*Then, for all $n \geq 1$, with probability at least $1 - \delta$,*

$$J_n(\boldsymbol{\pi}_n^*) - J_n(\boldsymbol{\pi}_n) \leq \sum_{t=0}^{T-1} \mathbb{E}_{\boldsymbol{\tau}^{\boldsymbol{\pi}_n}} \left[ J_{n,t+1}(\boldsymbol{\pi}_n, \boldsymbol{\eta}_n, \boldsymbol{x}_{t+1}') - J_{n,t+1}(\boldsymbol{\pi}_n, \boldsymbol{\eta}_n, \boldsymbol{x}_{t+1}) \right],$$

$$\text{with } \boldsymbol{x}_{t+1} = \boldsymbol{f}^*(\boldsymbol{x}_t, \boldsymbol{\pi}_n(\boldsymbol{x}_t)) + \boldsymbol{w}_t,$$

$$\text{and } \boldsymbol{x}_{t+1}' = \boldsymbol{\mu}_{n-1}(\boldsymbol{x}_t, \boldsymbol{\pi}_n(\boldsymbol{x}_t)) + \beta_{n-1}(\delta) \boldsymbol{\sigma}_{n-1}(\boldsymbol{x}_t, \boldsymbol{\pi}_n(\boldsymbol{x}_t)) \boldsymbol{\eta}_n(\boldsymbol{x}_t) + \boldsymbol{w}_t.$$

Lemma 2 gives a bound on the regret of planning optimistically under unknown dynamics. The regret is proportional to the difference in the expected returns for $\boldsymbol{x}_t$ and $\boldsymbol{x}_t'$. Note, $\|\boldsymbol{x}_t - \boldsymbol{x}_t'\| \propto \beta_n(\delta) \boldsymbol{\sigma}_{n-1}(\boldsymbol{x}_{t-1}, \boldsymbol{\pi}_n(\boldsymbol{x}_{t-1}))$. Hence, when we have low uncertainty in our predictions, planning optimistically suffers smaller regret. Next, we leverage Lemma 2 to give a sample complexity bound for the Bayesian and frequentist setting.

**Bayesian Setting** We start by introducing a measure of model complexity as defined by Curi et al. (2020).

$$\mathcal{MC}_N(\boldsymbol{f}^*) := \max_{\mathcal{D}_1, \ldots, \mathcal{D}_N \subset \mathcal{Z} \times \mathcal{X}} \sum_{n=1}^{N} \sum_{\boldsymbol{z} \in \mathcal{D}_n} \|\boldsymbol{\sigma}_{n-1}(\boldsymbol{z})\|_2^2. \qquad (8)$$

This complexity measure captures the difficulty of learning $\boldsymbol{f}^*$ given $N$ trajectories. Mainly, the more complicated $\boldsymbol{f}^*$, the larger the epistemic uncertainties $\boldsymbol{\sigma}_n$, and in turn, the larger corresponding $\mathcal{MC}_N(\boldsymbol{f}^*)$. Moreover, if the model complexity measure is sublinear in $N$, i.e. $\mathcal{MC}_N(\boldsymbol{f}^*)/N \to 0$ for $N \to \infty$, then the epistemic uncertainties also converge to zero in the limit, which implies

convergence to the true function $\boldsymbol{f}^*$. We present our main theoretical result, in terms of the model complexity measure.

**Theorem 1.** *Let Assumption 1 and 3 hold. Then, for all $N \geq 1$, with probability at least $1 - \delta$,*

$$\mathbb{E}_{\mathcal{D}_{1:N-1}} \left[ \max_{\boldsymbol{\pi} \in \Pi} \mathbb{E}_{\boldsymbol{\tau}^{\boldsymbol{\pi}}} \left[ I \left( \boldsymbol{f}^*_{\boldsymbol{\tau}^{\boldsymbol{\pi}}}; \boldsymbol{y}_{\boldsymbol{\tau}^{\boldsymbol{\pi}}} \mid \mathcal{D}_{1:N-1} \right) \right] \right] \leq \mathcal{O} \left( \beta_N T^{3/2} \sqrt{\frac{\mathcal{MC}_N(\boldsymbol{f}^*)}{N}} \right) \tag{9}$$

Theorem 1 relates the maximum expected information gain at iteration $N$ to the model complexity of our problem. For deterministic systems, the expectation w.r.t. $\boldsymbol{\tau}^{\boldsymbol{\pi}}$ is redundant. The bound in Equation (9) depends on the Lipschitz constants, planning horizon, and dimensionality of the state space (captured in $\beta_N$ and $\mathcal{MC}_N(\boldsymbol{f}^*)$). If the right-hand side is monotonically decreasing with $N$, Theorem 1 guarantees that the information gain at episode $N$ is also shrinking with $N$, and the algorithm is converging. Empirically, Pathak et al. (2019) show that the epistemic uncertainties go to zero as more data is acquired. In general, deriving a worst-case bound on the model complexity is a challenging and active open research problem. However, in the case of GPs, convergence results can be shown for a very rich class of functions. We show this in the following for the frequentist setting.

**Frequentist Setting with Gaussian Process Models**   We extend our analysis to the frequentist kernelized setting, where $\boldsymbol{f}^*$ resides in a Reproducing Kernel Hilbert Space (RKHS) of vector-valued functions.

**Assumption 4.** *We assume that the functions $f_j^*$, $j \in \{1, \ldots, d_x\}$ lie in a RKHS with kernel $k$ and have a bounded norm $B$, that is $\boldsymbol{f}^* \in \mathcal{H}_{k,B}^{d_x}$, with $\mathcal{H}_{k,B}^{d_x} = \{\boldsymbol{f} \mid \|f_j\|_k \leq B, j = 1, \ldots, d_x\}$.*

In this setting, we model the posterior mean and epistemic uncertainty of the vector-valued function $\boldsymbol{f}^*$ with $\boldsymbol{\mu}_n(\boldsymbol{z}) = [\mu_{n,j}(\boldsymbol{z})]_{j \leq d_x}$, and $\boldsymbol{\sigma}_n(\boldsymbol{z}) = [\sigma_{n,j}(\boldsymbol{z})]_{j \leq d_x}$, where,

$$\begin{aligned}
\mu_{n,j}(\boldsymbol{z}) &= \boldsymbol{k}_n^\top(\boldsymbol{z})(\boldsymbol{K}_n + \sigma^2 \boldsymbol{I})^{-1} \boldsymbol{y}_{1:n}^j, \\
\sigma_{n,j}^2(\boldsymbol{z}) &= k(\boldsymbol{x}, \boldsymbol{x}) - \boldsymbol{k}_n^\top(\boldsymbol{z})(\boldsymbol{K}_n + \sigma^2 \boldsymbol{I})^{-1} \boldsymbol{k}_n(\boldsymbol{x}),
\end{aligned} \tag{10}$$

Here, $\boldsymbol{y}_{1:n}^j$ corresponds to the noisy measurements of $f_j^*$, i.e., the observed next state from the transitions dataset $\mathcal{D}_{1:n}$, $\boldsymbol{k}_n = [k(\boldsymbol{z}, \boldsymbol{z}_i)]_{i \leq nT}, \boldsymbol{z}_i \in \mathcal{D}_{1:n}$, and $\boldsymbol{K}_n = [k(\boldsymbol{z}_i, \boldsymbol{z}_l)]_{i,l \leq nT}, \boldsymbol{z}_i, \boldsymbol{z}_l \in \mathcal{D}_{1:n}$ is the data kernel matrix. It is known that if $\boldsymbol{f}^*$ satisfies Assumption 4, then Equation (10) yields well-calibrated confidence intervals, i.e., that Assumption 1 is satisfied.

**Lemma 3** (Well calibrated confidence intervals for RKHS, Rothfuss et al. (2023)). *Let $\boldsymbol{f}^* \in \mathcal{H}_{k,B}^{d_x}$. Suppose $\boldsymbol{\mu}_n$ and $\boldsymbol{\sigma}_n$ are the posterior mean and variance of a GP with kernel $k$, c.f., Equation (10). There exists $\beta_n(\delta)$, for which the tuple $(\boldsymbol{\mu}_n, \boldsymbol{\sigma}_n, \beta_n(\delta))$ satisfies Assumption 1 w.r.t. function $\boldsymbol{f}^*$.*

Theorem 2 presents our convergence guarantee for the kernelized case to the $T$-step reachability set $\mathcal{R}$ for the policy class $\pi \in \Pi$. In particular, $\mathcal{R}$ is defined as

$$\mathcal{R} = \{\boldsymbol{z} \in \mathcal{Z} \mid \exists (\boldsymbol{\pi} \in \Pi, t \leq T), \text{ s.t.}, p(\boldsymbol{z}_t = \boldsymbol{z} \mid \boldsymbol{\pi}, \boldsymbol{f}^*) > 0\}$$

There are two key differences from Theorem 1; (*i*) we can derive an upper bound on the epistemic uncertainties $\boldsymbol{\sigma}_n$, and (*ii*) we can bound the model complexity $\mathcal{MC}_N(\boldsymbol{f}^*)$, with the *maximum information gain* of kernel $k$ introduced by Srinivas et al. (2012), defined as

$$\gamma_N(k) = \max_{\mathcal{D}_1, \ldots, \mathcal{D}_N; |\mathcal{D}_n| \leq T} \frac{1}{2} \log \det(\boldsymbol{I} + \sigma^{-2} \boldsymbol{K}_N).$$

**Theorem 2.** *Let Assumption 3 and 4 hold, Then, for all $N \geq 1$, with probability at least $1 - \delta$,*

$$\max_{\boldsymbol{\pi} \in \Pi} \mathbb{E}_{\boldsymbol{\tau}^{\boldsymbol{\pi}}} \left[ \max_{\boldsymbol{z} \in \boldsymbol{\tau}^{\boldsymbol{\pi}}} \sum_{j=1}^{d_x} \frac{1}{2} \sigma_{N,j}^2(\boldsymbol{z}) \right] \leq \mathcal{O} \left( \beta_N T^{3/2} \sqrt{\frac{\gamma_N(k)}{N}} \right). \tag{11}$$

*If we relax noise Assumption 3 to $\sigma$-sub Gaussian. Then, if Assumption 2 holds, we have for all $N \geq 1$, with probability at least $1 - \delta$,*

$$\max_{\boldsymbol{\pi} \in \Pi} \mathbb{E}_{\boldsymbol{\tau}^{\boldsymbol{\pi}}} \left[ \max_{\boldsymbol{z} \in \boldsymbol{\tau}^{\boldsymbol{\pi}}} \sum_{j=1}^{d_x} \frac{1}{2} \sigma_{N,j}^2(\boldsymbol{z}) \right] \leq \mathcal{O} \left( \beta_N^T T^{3/2} \sqrt{\frac{\gamma_N(k)}{N}} \right). \tag{12}$$

*Moreover, if $\gamma_N(k) = \mathcal{O}\left(polylog(N)\right)$, then for all $z \in \mathcal{R}$, and $1 \leq j \leq d_x$,*

$$\sigma_{N,j}(z) \xrightarrow{a.s.} 0 \text{ for } N \to \infty. \tag{13}$$

We only state Theorem 2 for the expected epistemic uncertainty along the trajectory at iteration $N$. For deterministic systems, the expectation is redundant and for stochastic systems, we can leverage concentration inequalities to give a bound without the expectation (see Appendix A for more detail).

For the Gaussian noise case, we obtain a tighter bound by leveraging the change of measure inequality from Kakade et al. (2020, Lemma C.2.) (c.f., Lemma 6 in Appendix A for more detail). In the more general case of sub-Gaussian noise, we cannot use the same analysis. To this end, we use the Lipschitz continuity assumptions (Assumption 2) similar to Curi et al. (2020). This results in comparing the deviation between two trajectories under the same policy and dynamics but different initial states (see Lemma 2). For many systems (even linear) this can grow exponentially in the horizon $T$. Accordingly, we obtain a $\beta_N^T$ term in our bound (Equation (12)). Nonetheless, for cases where the RKHS is of a kernel with maximum information gain $\gamma_N(k) = \mathcal{O}\left(polylog(N)\right)$, we can give sample complexity bounds and an almost sure convergence result in the reachable set $\mathcal{R}$ (Equation (13)). Kernels such as the RBF kernel or the linear kernel (kernel with a finite-dimensional feature map $\phi(x)$) have maximum information gain which grows polylogarithmically with $n$ (Vakili et al. (2021)). Therefore, our convergence guarantees hold for a very rich class of functions. The exponential dependence of our bound on $T$ imposes the restriction on the kernel class. For the case of Gaussian noise, we can include a richer class of kernels, such as Matèrn.

In addition to the convergence results above, we also give guarantees on the zero-shot performance of OPAX in Appendix A.5.

## 5 Experiments

We evaluate OPAX on the Pendulum-v1 and MountainCar environment from the OpenAI gym benchmark suite (Brockman et al., 2016), on the Reacher, Swimmer, and Cheetah from the deep mind control suite (Tassa et al., 2018), and a high-dimensional simulated robotic manipulation task introduced by Li et al. (2020). See Appendix B for more details on the experimental setup.

**Baselines** We implement four baselines for comparisons. To show the benefit of our intrinsic reward, we compare OPAX to (*1*) a random exploration policy (RANDOM) which randomly samples actions from the action space. As we discuss in Section 3 our choice of objective in Equation (6) is in essence similar to the one proposed by Pathak et al. (2019) and Sekar et al. (2020). Therefore, in our experiments, we compare the optimistic planner with other planning approaches. Moreover, most work on active exploration either uses the mean planner or does not specify the planner (c.f., Section 6). We use the most common planners: (*2*) mean (MEAN-AE), and (*3*) trajectory sampling (TS-1) scheme proposed in Chua et al. (2018) (PETS-AE) as our baselines. The mean planner simply uses the mean estimate $\mu_n$ of the well-calibrated model. This is also used in Buisson-Fenet et al. (2020). Finally, we compare OPAX to (*4*) H-UCRL (Curi et al., 2020), a single-task model-based RL algorithm. We investigate the following three aspects: (*i*) *how fast does active exploration reduce model's epistemic uncertainty* $\sigma_n$ *with increasing* $n$, (*ii*) *can we solve downstream tasks with* OPAX, and (*iii*) *does* OPAX *scale to high-dimensional and challenging object manipulation tasks*? For our experiments, we use GPs and probabilistic ensembles (PE, Lakshminarayanan et al. (2017)) for modeling the dynamics. For the planning, we either the soft actor-critic (SAC, Haarnoja et al. (2018)) policy optimizer, which takes simulated trajectories from our learned model to train a policy, or MPC with the iCEM optimizer (Pinneri et al., 2021).

**How fast does active exploration reduce the epistemic uncertainty?** For this experiment, we consider the Pendulum-v1 environment. We sample transitions at random from the pendulum's *reachable* state-action space and evaluate our model's epistemic uncertainty for varying episodes and baselines. We model the dynamics with both GPs and PE. We depict the result in Figure 1. We conclude that the RANDOM agent is slower in reducing the uncertainty compared to other active exploration methods for both GP and PE models. In particular, from the experiment, we empirically validate Theorem 2 for the GP case and also conclude that empirically even when using PE models, we find convergence of epistemic uncertainty. Moreover, we notice for the PE case that OPAX reaches smaller uncertainties slightly faster than MEAN-AE and PETS-AE. We believe this is due to the additional exploration induced by the optimistic planner.

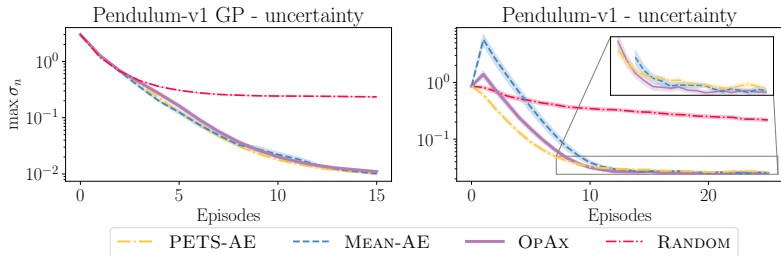

Figure 1: Reduction in maximum epistemic uncertainty in *reachable* state-action space for the Pendulum-v1 environment over 10 different random seeds. We evaluate OPAX with both GPs and PE and plot the mean performance with two standard error confidence intervals. For both, active exploration reduces epistemic uncertainty faster compared to random exploration. All active exploration baselines perform well for the GP case, whereas for the PE case OPAX gives slightly lower uncertainties.

**Can the model learnt through OPAX solve downstream tasks?**  We use OPAX and other active exploration baselines to actively learn a dynamics model and then evaluate the learned model on downstream tasks. We consider several tasks, (*i*) Pendulum-v1 swing up, (*ii*) Pendulum-v1 keep down (keep the pendulum at the stable equilibria), (*iii*) MountainCar, (*iv*) Reacher - go to target, (*v*) Swimmer - go to target, (*vi*) Swimmer - go away from target (quickly go away from the target position), (*vii*) Cheetah - run forward, (*viii*) Cheetah - run backward. For all tasks, we consider PEs, except for (*i*) where we also use GPs. Furthermore, for the MountainCar and Reacher, we give a reward once the goal is reached. Since this requires long-term planning, we use a SAC policy for these tasks. We use MPC with iCEM for the remaining tasks. We also train H-UCRL on tasks (*i*) with GPs, and (*ii*), (*iii*), (*iv*), (*v*), (*vii*) with PEs. We report the best performance across all episodes.

To make a fair comparison, we use the following evaluation procedure; first, we perform active exploration for each episode on the environment, and then after every few episodes we use the mean estimate $\boldsymbol{\mu}_n$ to evaluate our learned model on the downstream tasks.

Figure 2 shows that all active exploration variants perform considerably better than the RANDOM agent. In particular, for the MountainCar, the RANDOM agent is not able to solve the task. Moreover, PETS-AE performs slightly worse than the other exploration baselines in this environment. In general, we notice that OPAX always performs well and is able to achieve H-UCRL's performance on all the tasks for which H-UCRL is trained. However, on tasks that are new/unseen for H-UCRL, active exploration algorithms outperform H-UCRL. From this experiment, we conclude two things (*1*) apart from providing theoretical guarantees, the model learned through OPAX also performs well in downstream tasks, and (*2*) active exploration agents generalize well to downstream tasks, whereas H-UCRL performs considerably worse on new/unseen tasks. We believe this is because, unlike active exploration agents, task-specific model-based RL agents only explore the regions of the state-action space that are relevant to the task at hand.

**Does OPAX scale to high-dimensional and challenging object manipulation tasks?**  To answer this question, we consider the Fetch Pick & Place Construction environment (Li et al., 2020). We again use the active exploration agents to learn a model and then evaluate the success rate of the learned model in three challenging downstream tasks: (*i*) Pick & Place, (*ii*) Throw, and (*iii*) Flip (see Figure 4). The environment contains a 7-DoF robot arm and four 6-DoF blocks that can be manipulated. In total, the state space is 58-dimensional. The 4-dimensional actions control the end-effector of the robot in Cartesian space as well as the opening/closing of the gripper. We compare OPAX to PETS-AE, MEAN-AE, a random policy as well as CEE-US (Sancaktar et al., 2022). CEE-US is a model-based active exploration algorithm, for which Sancaktar et al. (2022) reports state-of-the-art performance compared to several other active exploration methods. In all three tasks,

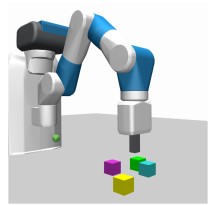

Figure 3: Fetch Pick & Place Construction environment.

OPAX is at least on par with the best-performing baselines, including CEE-US. We run OPAX and all baselines with the same architecture and hyperparameter settings. See Appendix B for more details.

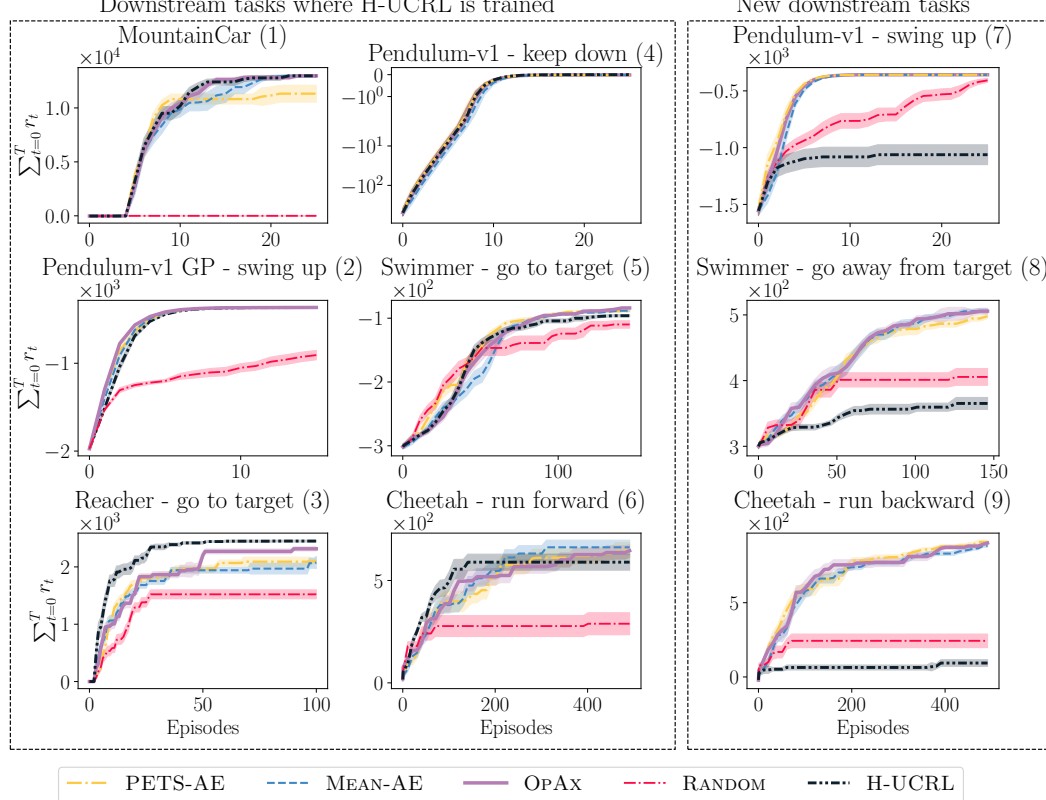

Figure 2: We evaluate the downstream performance of our agents over 10 different random seeds and plot the mean performance with two standard error confidence intervals. For all the environments we use PE as models, except plot (1), for which we use a GP model (see plot (2) in the figure above). For tasks (1)-(6), we also train H-UCRL, a model-based RL algorithm. Tasks (7)-(9) are new/unseen for H-UCRL. From the Figure, we conclude that (*i*) compared to other active exploration baselines, OPAX constantly performs well and is on par with H-UCRL, and (*ii*) on the new/unseen tasks the active exploration baselines and OPAX outperform H-UCRL by a large margin.

## 6 Related Work

System identification is a broadly studied topic (Åström and Eykhoff, 1971; Schoukens and Ljung, 2019; Schön et al., 2011; Ziemann et al., 2022; Ziemann and Tu, 2022). However, system identification from the perspective of experiment design for nonlinear systems is much less understood (Chiuso and Pillonetto, 2019). Most methods formulate the identification task through the maximization of intrinsic rewards. Common choices of intrinsic rewards are (*i*) model prediction error or "Curiosity" (Schmidhuber, 1991; Pathak et al., 2017), (*ii*) novelty of transitions (Stadie et al., 2015), and (*iii*) diversity of skills (Eysenbach et al., 2018).

A popular choice for intrinsic rewards is mutual information or entropy (Jain et al., 2018; Buisson-Fenet et al., 2020; Shyam et al., 2019; Pathak et al., 2019; Sekar et al., 2020). Jain et al. (2018) propose an approach to maximize the information gain greedily wrt the immediate next transition, i.e., one-step greedy, whereas Buisson-Fenet et al. (2020) consider planning full trajectories. Shyam et al. (2019); Pathak et al. (2019); Sekar et al. (2020) and Sancaktar et al. (2022) consider general Bayesian models, such as BNNs, to represent a probabilistic distribution for the learned model. Shyam et al. (2019) propose using the information gain of the model with respect to observed transition as the intrinsic reward. To this end, they learn an ensemble of Gaussian neural networks and represent the distribution over models with a Gaussian mixture model (GMM). A similar approach is also proposed in Pathak et al. (2019); Sekar et al. (2020); Sancaktar et al. (2022). The main difference between Shyam et al. (2019) and Pathak et al. (2019) lies in how they represent mutual information. Moreover, Pathak et al. (2019) use the model's epistemic uncertainty, that is the

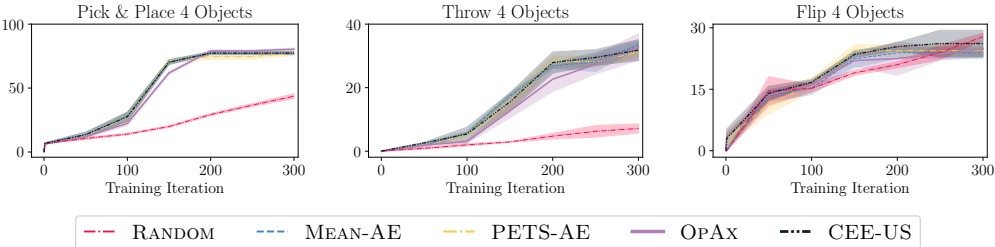

Figure 4: Success rates for pick & place, throwing and flipping tasks with four objects in the Fetch Pick & Place Construction environment for OPAX and baselines. We evaluate task performance via planning zero-shot with models learned using different exploration strategies. We report performance on three independent seeds. OPAX is on par with the best-performing baselines in all tasks.

disagreement between the ensemble models as an intrinsic reward. Sekar et al. (2020) link the model disagreement (epistemic uncertainty) reward to maximizing mutual information and demonstrate state-of-the-art performance on several high-dimensional tasks. Similarly, Sancaktar et al. (2022), use the disagreement in predicted trajectories of an ensemble of neural networks to direct exploration. Since trajectories can diverge due to many factors beyond just the model epistemic uncertainty, e.g., aleatoric noise, this approach is restricted to deterministic systems and susceptible to systems with unstable equilibria. Our approach is the most similar to Pathak et al. (2019); Sekar et al. (2020) since we also propose the model epistemic uncertainty as the intrinsic reward for planning. However, we thoroughly and theoretically motivate this choice of reward from a Bayesian experiment design perspective. Furthermore, we induce additional exploration in OPAX through our optimistic planner and rigorously study the theoretical properties of the proposed methods. On the contrary, most of the prior work discussed above either uses the mean planner (MEAN-AE) or does not discuss the planner thoroughly or provide any theoretical results. In general, theoretical guarantees for active exploration algorithms are rather immature (Chakraborty et al., 2023; Wagenmaker et al., 2023) and mostly restrictive to a small class of systems (Simchowitz et al., 2018; Tarbouriech et al., 2020; Wagenmaker and Jamieson, 2020; Mania et al., 2020). To the best of our knowledge, we are the first to give convergence guarantees for a rich class of nonlinear systems.

While our work focuses on the active learning of dynamics, there are numerous works that study exploration in the context of reward-free RL (Jin et al., 2020; Kaufmann et al., 2021; Wagenmaker et al., 2022; Chen et al., 2022). However, most methods in this setting give guarantees for special classes of MDPs (Jin et al., 2020; Kaufmann et al., 2021; Wagenmaker et al., 2022; Qiu et al., 2021; Chen et al., 2022) and result in practical algorithms. On the contrary, we focus on solely learning the dynamics. While a good dynamics model may be used for zero-shot planning, it also exhibits more relevant knowledge about the system such as its stability or sensitivity to external effects. Furthermore, our proposed method is not only theoretically sound but also practical.

## 7 Conclusion

We present OPAX, a novel model-based RL algorithm for the active exploration of unknown dynamical systems. Taking inspiration from Bayesian experiment design, we provide a comprehensive explanation for using model epistemic uncertainty as an intrinsic reward for exploration. By leveraging the *optimistic in the face of uncertainty* paradigm, we put forth first-of-their-kind theoretical results on the convergence of active exploration agents in reinforcement learning. Specifically, we study convergence properties of general Bayesian models, such as BNNs. For the frequentist case of RKHS dynamics, we established sample complexity bounds and convergence guarantees for OPAX for a rich class of functions. We evaluate the efficacy of OPAX across various RL environments with state space dimensions from two to 58. The empirical results corroborate our theoretical findings, as OPAX displays systematic and effective exploration across all tested environments and exhibits strong performance in zero-shot planning for new downstream tasks.

## Acknowledgments and Disclosure of Funding

We would like to thank Jonas Hübotter for the insightful discussions and his feedback on this work. Furthermore, we also thank Alex Hägele, Parnian Kassraie, Scott Sussex, and Dominik Baumann for their feedback.

This project has received funding from the Swiss National Science Foundation under NCCR Automation, grant agreement 51NF40 180545, and the Microsoft Swiss Joint Research Center.

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

# Contents of Appendix

# A Proofs for section 4

We first prove some key properties of our active exploration objective in Equation (6). Then, we prove Theorem 1 which holds for general Bayesian models, and finally we prove Theorem 2, which guarantees convergence for the frequentist setting where the dynamics are modeled using a GP.

**Lemma 4** (Properties of OPAX's objective). *Let Assumption 1 and 2 hold, then the following is true for all $n \geq 0$,*

$$\log \left( 1 + \frac{\sigma_{n-1,j}^2(\boldsymbol{x}_t, \boldsymbol{\pi}(\boldsymbol{x}_t))}{\sigma^2} \right) \geq 0 \tag{1}$$

$$\frac{1}{2} \sup_{\boldsymbol{\pi} \in \Pi, \boldsymbol{\eta} \in \Xi} \mathbb{E} \left[ \left( \sum_{t=0}^{T-1} \sum_{j=1}^{d_x} \log \left( 1 + \frac{\sigma_{n-1,j}^2(\boldsymbol{x}_t, \boldsymbol{\pi}(\boldsymbol{x}_t))}{\sigma^2} \right) \right)^2 \right] \leq \frac{1}{2} T^2 d_x^2 \log^2 \left( 1 + \frac{\sigma_{\max}^2}{\sigma^2} \right), \tag{2}$$

$$\left| \frac{1}{2} \sum_{j=1}^{d_x} \log \left( 1 + \frac{\sigma_{n,j}^2(\boldsymbol{z})}{\sigma^2} \right) - \log \left( 1 + \frac{\sigma_{n,j}^2(\boldsymbol{z}')}{\sigma^2} \right) \right| \leq \frac{d_x \sigma_{\max} L_{\boldsymbol{\sigma}}}{\sigma^2} \|\boldsymbol{z} - \boldsymbol{z}'\|. \tag{3}$$

*where $\sigma_{\max} = \sup_{\boldsymbol{z} \in \mathcal{Z}; i \geq 0; 1 \leq j \leq d_x} \sigma_{i,j}(\boldsymbol{z})$.*

*Proof.* The positivity of the reward follows from the positive definiteness of the epistemic uncertainty $\sigma_{n-1,j}$. For (2), the following holds

$$\mathbb{E} \left[ \left( \sum_{t=0}^{T-1} \sum_{j=1}^{d_x} \log \left( 1 + \frac{\sigma_{n-1,j}^2(\boldsymbol{x}_t, \boldsymbol{\pi}(\boldsymbol{x}_t))}{\sigma^2} \right) \right)^2 \right]$$

$$\leq \mathbb{E} \left[ \sum_{t=0}^{T-1} \sum_{j=1}^{d_x} T d_x \log^2 \left( 1 + \frac{\sigma_{n-1,j}^2(\boldsymbol{x}_t, \boldsymbol{\pi}(\boldsymbol{x}_t))}{\sigma^2} \right) \right]$$

$$\leq \mathbb{E} \left[ \left( \sum_{t=0}^{T-1} \sum_{j=1}^{d_x} T d_x \log^2 \left( 1 + \frac{\sigma_{\max}^2}{\sigma^2} \right) \right) \right]$$

$$\leq T^2 d_x^2 \log^2 \left( 1 + \frac{\sigma_{\max}^2}{\sigma^2} \right)$$

From hereon, let $J_{\max} = 1/2 T^2 d_x^2 \log^2 \left( 1 + \sigma^{-2} \sigma_{\max}^2 \right)$.

Finally, we show that this reward is Lipschitz continuous.

$$|\sigma_{n,j}^2(\boldsymbol{z}) - \sigma_{n,j}^2(\boldsymbol{z}')| = |\sigma_{n,j}(\boldsymbol{z})\sigma_{n,j}(\boldsymbol{z}) - \sigma_{n,j}(\boldsymbol{z})\sigma_{n,j}(\boldsymbol{z}') + \sigma_{n,j}(\boldsymbol{z})\sigma_{n,j}(\boldsymbol{z}') - \sigma_{n,j}(\boldsymbol{z}')\sigma_{n,j}(\boldsymbol{z}')|$$

$$\leq L_\sigma \|\boldsymbol{z} - \boldsymbol{z}'\| \sigma_{n,j}(\boldsymbol{z}) + L_\sigma \|\boldsymbol{z} - \boldsymbol{z}'\| \sigma_{n,j}(\boldsymbol{z}')$$

$$\leq 2\sigma_{\max} L_\sigma \|\boldsymbol{z} - \boldsymbol{z}'\|.$$

$$\left| \frac{1}{2} \sum_{j=1}^{d_x} \log \left( 1 + \frac{\sigma_{n,j}^2(\boldsymbol{z})}{\sigma^2} \right) - \log \left( 1 + \frac{\sigma_{n,j}^2(\boldsymbol{z}')}{\sigma^2} \right) \right| = \frac{1}{2} \left| \sum_{j=1}^{d_x} \log \left( 1 + \frac{\frac{\sigma_{n,j}^2(\boldsymbol{z}) - \sigma_{n,j}^2(\boldsymbol{z}')}{\sigma^2}}{1 + \frac{\sigma_{n,j}^2(\boldsymbol{z}')}{\sigma^2}} \right) \right|$$

$$\leq \frac{1}{2} \left| \sum_{j=1}^{d_x} \log \left( 1 + \frac{|\sigma_{n,j}^2(\boldsymbol{z}) - \sigma_{n,j}^2(\boldsymbol{z}')|}{\sigma^2} \right) \right|$$

$$\leq \frac{1}{2\sigma^2} \sum_{j=1}^{d_x} |\sigma_{n,j}^2(\boldsymbol{z}) - \sigma_{n,j}^2(\boldsymbol{z}')| \tag{*}$$

$$\leq \frac{d_x \sigma_{\max} L_{\boldsymbol{\sigma}}}{\sigma^2} \|\boldsymbol{z} - \boldsymbol{z}'\|.$$

Where (*) is true because for all $x \geq 0$, $\log(1 + x) \leq x$. $\qquad\square$

**Corollary 1** (OPAX gives an optimistic estimate on Equation (6))**.** *Let Assumption 1 hold and $\boldsymbol{\pi}_n^*$ denote the solution to Equation (6) and $J_n(\boldsymbol{\pi}_n^*)$ the resulting objective. Similarly, let $\boldsymbol{\pi}_n$ and $\boldsymbol{\eta}_n$ be the solution to Equation (7) and $J_n(\boldsymbol{\pi}_n, \boldsymbol{\eta}_n)$ the corresponding value of the objective. Then with probability at least $1 - \delta$ we have for every episode $n \in \{1, \dots, N\}$:*

$$J_n(\boldsymbol{\pi}_n^*) \leq J_n(\boldsymbol{\pi}_n, \boldsymbol{\eta}_n).$$

*Proof.* Follows directly from Assumption 1. $\qquad\square$

### A.1 Proof of Lemma 2

**Lemma 5** (Difference in Policy performance)**.** *Let $J_{r,k}(\boldsymbol{\pi}, \boldsymbol{x}_k) = \mathbb{E}_{\boldsymbol{\tau}\boldsymbol{\pi}}\left[\sum_{t=k}^{T-1} r(\boldsymbol{x}_t, \boldsymbol{\pi}(\boldsymbol{x}_t))\right]$ and $A_{r,k}(\boldsymbol{\pi}, \boldsymbol{x}, \boldsymbol{a}) = \mathbb{E}_{\boldsymbol{\tau}\boldsymbol{\pi}}\left[r(\boldsymbol{x}, \boldsymbol{a}) + J_{r,k+1}(\boldsymbol{\pi}, \boldsymbol{x}') - J_{r,k}(\boldsymbol{\pi}, \boldsymbol{x})\right]$ with $\boldsymbol{x}' = \boldsymbol{f}^*(\boldsymbol{x}, \boldsymbol{a}) + \boldsymbol{w}$. For simplicity we refer to $J_{r,0}(\boldsymbol{\pi}, \boldsymbol{x}_0) = J_r(\boldsymbol{\pi}, \boldsymbol{x}_0)$. The following holds for all $\boldsymbol{x}_0 \in \mathcal{X}$:*

$$J_r(\boldsymbol{\pi}', \boldsymbol{x}_0) - J_r(\boldsymbol{\pi}, \boldsymbol{x}_0) = \mathbb{E}_{\boldsymbol{\tau}\boldsymbol{\pi}'}\left[\sum_{t=0}^{T-1} A_{r,t}(\boldsymbol{\pi}, \boldsymbol{x}_t', \boldsymbol{\pi}'(\boldsymbol{x}_t'))\right]$$

*Proof.*

$$
\begin{aligned}
J_r(\boldsymbol{\pi}', \boldsymbol{x}_0) &= \mathbb{E}_{\boldsymbol{\tau}\boldsymbol{\pi}'}\left[\sum_{t=0}^{T-1} r(\boldsymbol{x}_t', \boldsymbol{\pi}'(\boldsymbol{x}_t'))\right] = \mathbb{E}_{\boldsymbol{\tau}\boldsymbol{\pi}'}\left[r(\boldsymbol{x}_0, \boldsymbol{\pi}'(\boldsymbol{x}_0)) + J_{r,1}(\boldsymbol{\pi}', \boldsymbol{x}_1')\right]\\
&= \mathbb{E}_{\boldsymbol{\tau}\boldsymbol{\pi}'}\left[r(\boldsymbol{x}_0, \boldsymbol{\pi}'(\boldsymbol{x}_0)) + J_{r,1}(\boldsymbol{\pi}, \boldsymbol{x}_1') + J_{r,1}(\boldsymbol{\pi}', \boldsymbol{x}_1') - J_{r,1}(\boldsymbol{\pi}, \boldsymbol{x}_1')\right]\\
&= \mathbb{E}_{\boldsymbol{\tau}\boldsymbol{\pi}'}\left[r(\boldsymbol{x}_0, \boldsymbol{\pi}'(\boldsymbol{x}_0)) + J_{r,1}(\boldsymbol{\pi}, \boldsymbol{x}_1') - J_r(\boldsymbol{\pi}, \boldsymbol{x}_0)\right]\\
&\quad + J_r(\boldsymbol{\pi}, \boldsymbol{x}_0) + \mathbb{E}_{\boldsymbol{\tau}\boldsymbol{\pi}'}\left[J_{r,1}(\boldsymbol{\pi}', \boldsymbol{x}_1') - J_{r,1}(\boldsymbol{\pi}, \boldsymbol{x}_1')\right]\\
&= \mathbb{E}_{\boldsymbol{\tau}\boldsymbol{\pi}'}\left[A_{r,0}(\boldsymbol{\pi}, \boldsymbol{x}_0, \boldsymbol{\pi}'(\boldsymbol{x}_0))\right] + J_r(\boldsymbol{\pi}, \boldsymbol{x}_0) + \mathbb{E}_{\boldsymbol{\tau}\boldsymbol{\pi}'}\left[J_{r,1}(\boldsymbol{\pi}', \boldsymbol{x}_1) - J_{r,1}(\boldsymbol{\pi}, \boldsymbol{x}_1)\right]
\end{aligned}
$$

Therefore we obtain

$$J_r(\boldsymbol{\pi}', \boldsymbol{x}_0) - J_r(\boldsymbol{\pi}, \boldsymbol{x}_0) = \mathbb{E}_{\boldsymbol{\tau}\boldsymbol{\pi}'}\left[A_0(\boldsymbol{\pi}, \boldsymbol{x}_0, \boldsymbol{\pi}'(\boldsymbol{x}_0))\right] + \mathbb{E}_{\boldsymbol{\tau}\boldsymbol{\pi}'}\left[J_{r,1}(\boldsymbol{\pi}', \boldsymbol{x}_1') - J_{r,1}(\boldsymbol{\pi}, \boldsymbol{x}_1')\right].$$

Using the same argument for $J_{r,1}$, $J_{r,2}$, …, $J_{r,T-1}$ and that $J_{r,T}(\boldsymbol{\pi}, \boldsymbol{x}) = 0$ for all $\boldsymbol{\pi} \in \Pi$ and $\boldsymbol{x} \in \mathcal{X}$ completes the proof. $\qquad\square$

Assume a policy $\boldsymbol{\pi}$ is fixed and dynamics are of the form:

$$\boldsymbol{x}' = \boldsymbol{\mu}_n(\boldsymbol{x}, \boldsymbol{\pi}(\boldsymbol{x})) + \beta_n(\delta)\boldsymbol{\sigma}(\boldsymbol{x}, \boldsymbol{\pi}(\boldsymbol{x}))\boldsymbol{u} + \boldsymbol{w}. \tag{14}$$

Here $\boldsymbol{u} \in [-1, 1]^{d_x}$. Furthermore, assume that the associated running rewards do not depend on $\boldsymbol{u}$, that is, $r(\boldsymbol{x}_t)$, and let $\boldsymbol{\eta} \in \Xi$ denote the policy, i.e., $\boldsymbol{\eta} : \mathcal{X} \to [-1, 1]^{d_x}$.

**Corollary 2.** *The following holds for all $\boldsymbol{x}_0 \in \mathcal{X}$ and policy $\boldsymbol{\pi}$:*

$$J_r(\boldsymbol{\pi}, \boldsymbol{\eta}', \boldsymbol{x}_0) - J_r(\boldsymbol{\pi}, \boldsymbol{\eta}, \boldsymbol{x}_0) = \mathbb{E}_{\boldsymbol{\tau}\boldsymbol{\eta}'}\left[\sum_{t=0}^{T-1} J_{r,t+1}(\boldsymbol{\pi}, \boldsymbol{\eta}, \boldsymbol{x}_{t+1}') - J_{r,t+1}(\boldsymbol{\pi}, \boldsymbol{\eta}, \boldsymbol{x}_{t+1})\right],$$

*with $\boldsymbol{x}_{t+1} = \boldsymbol{\mu}_n(\boldsymbol{x}_t', \boldsymbol{\pi}(\boldsymbol{x}_t')) + \beta_n(\delta)\boldsymbol{\sigma}(\boldsymbol{x}_t', \boldsymbol{\pi}(\boldsymbol{x}_t'))\boldsymbol{\eta}(\boldsymbol{x}_t') + \boldsymbol{w}_t$, and $\boldsymbol{x}_{t+1}' = \boldsymbol{\mu}_n(\boldsymbol{x}_t', \boldsymbol{\pi}(\boldsymbol{x}_t')) + \beta_n(\delta)\boldsymbol{\sigma}(\boldsymbol{x}_t', \boldsymbol{\pi}(\boldsymbol{x}_t'))\boldsymbol{\eta}'(\boldsymbol{x}_t') + \boldsymbol{w}_t$.*

*Proof.* From Lemma 5 we have

$$J_r(\boldsymbol{\pi}, \boldsymbol{\eta}', \boldsymbol{x}_0) - J_r(\boldsymbol{\pi}, \boldsymbol{\eta}, \boldsymbol{x}_0) = \mathbb{E}_{\boldsymbol{\tau}\boldsymbol{\eta}'}\left[\sum_{t=0}^{T-1} A_{r,t}(\boldsymbol{\eta}, \boldsymbol{x}_t', \boldsymbol{\eta}'(\boldsymbol{x}_t'))\right].$$

Furthermore,

$$
\begin{aligned}
\mathbb{E}_{\boldsymbol{\tau}\boldsymbol{\eta}'}\left[A_{r,t}(\boldsymbol{\eta}, \boldsymbol{x}_t', \boldsymbol{\eta}'(\boldsymbol{x}_t'))\right] &= \mathbb{E}_{\boldsymbol{\tau}\boldsymbol{\eta}'}\left[r(\boldsymbol{x}_t') + J_{r,t+1}(\boldsymbol{\pi}, \boldsymbol{\eta}, \boldsymbol{x}_{t+1}') - J_{r,t}(\boldsymbol{\pi}, \boldsymbol{\eta}, \boldsymbol{x}_t')\right]\\
&= \mathbb{E}_{\boldsymbol{\tau}\boldsymbol{\eta}'}\left[r(\boldsymbol{x}_t') + J_{r,t+1}(\boldsymbol{\pi}, \boldsymbol{\eta}, \boldsymbol{x}_{t+1}') - r(\boldsymbol{x}_t') - J_{r,t+1}(\boldsymbol{\pi}, \boldsymbol{\eta}, \boldsymbol{x}_{t+1})\right]\\
&= \mathbb{E}_{\boldsymbol{\tau}\boldsymbol{\eta}'}\left[J_{r,t+1}(\boldsymbol{\pi}, \boldsymbol{\eta}, \boldsymbol{x}_{t+1}') - J_{r,t+1}(\boldsymbol{\pi}, \boldsymbol{\eta}, \boldsymbol{x}_{t+1})\right].
\end{aligned}
$$

$\qquad\square$

*Proof of Lemma 2.* From Assumption 1 we know that with probability at least $1 - \delta$ there exists a $\bar{\boldsymbol{\eta}}$ such that $\boldsymbol{f}^*(\boldsymbol{z}) = \boldsymbol{\mu}_n(\boldsymbol{z}) + \beta_n(\delta)\boldsymbol{\sigma}(\boldsymbol{z})\bar{\boldsymbol{\eta}}(\boldsymbol{x})$ for all $\boldsymbol{z} \in \mathcal{Z}$.

$$
\begin{aligned}
J_n(\boldsymbol{\pi}_n^*) - J_n(\boldsymbol{\pi}_n) &\leq J_n(\boldsymbol{\pi}_n, \boldsymbol{\eta}_n) - J_n(\boldsymbol{\pi}_n) && \text{(Corollary 1)} \\
&= J_n(\boldsymbol{\pi}_n, \boldsymbol{\eta}_n) - J_n(\boldsymbol{\pi}_n, \bar{\boldsymbol{\eta}}) \\
&= \mathbb{E}_{\boldsymbol{\tau}^{\bar{\eta}}}\left[\sum_{t=0}^{T-1} J_{n,t+1}(\boldsymbol{\pi}_n, \boldsymbol{\eta}_n, \boldsymbol{x}'_{t+1}) - J_{n,t+1}(\boldsymbol{\pi}_n, \boldsymbol{\eta}_n, \boldsymbol{x}_{t+1})\right] && \text{(Corollary 2)} \\
&= \mathbb{E}_{\boldsymbol{\tau}^{\pi_n}}\left[\sum_{t=0}^{T-1} J_{n,t+1}(\boldsymbol{\pi}_n, \boldsymbol{\eta}_n, \boldsymbol{x}'_{t+1}) - J_{n,t+1}(\boldsymbol{\pi}_n, \boldsymbol{\eta}_n, \boldsymbol{x}_{t+1})\right], \\
&\qquad\qquad\qquad\qquad \text{(Expectation wrt } \boldsymbol{\pi}_n \text{ under true dynamics } \boldsymbol{f}^*)
\end{aligned}
$$

with $\boldsymbol{x}_{t+1} = \boldsymbol{f}^*(\boldsymbol{x}_t, \boldsymbol{\pi}_n(\boldsymbol{x}_t)) + \boldsymbol{w}_t$,

and $\boldsymbol{x}'_{t+1} = \boldsymbol{\mu}_{n-1}(\boldsymbol{x}_t, \boldsymbol{\pi}_n(\boldsymbol{x}_t)) + \beta_{n-1}(\delta)\boldsymbol{\sigma}_{n-1}(\boldsymbol{x}_t, \boldsymbol{\pi}_n(\boldsymbol{x}_t))\boldsymbol{\eta}_n(\boldsymbol{x}_t) + \boldsymbol{w}_t$.

$\square$

## A.2 Analyzing regret of optimistic planning

In the following, we analyze the regret of optimistic planning for both $\sigma$-Gaussian noise and $\sigma$-sub Gaussian noise case. We start with the Gaussian case.

**Lemma 6** (Absolute expectation Difference Under Two Gaussians (Lemma C.2. Kakade et al. (2020))). *For Gaussian distribution $\mathcal{N}(\mu_1, \sigma^2\mathbb{I})$ and $\mathcal{N}(\mu_2, \sigma^2\mathbb{I})$, and for any (appropriately measurable) positive function g, it holds that:*

$$
|\mathbb{E}_{z\sim\mathcal{N}_1}[g(z)] - \mathbb{E}_{z\sim\mathcal{N}_2}[g(z)]| \leq \min\left\{\frac{\|\mu_1 - \mu_2\|}{\sigma^2}, 1\right\}\sqrt{\mathbb{E}_{z\sim\mathcal{N}_1}[g^2(z)]}
$$

*Proof.*

$$
\begin{aligned}
|\mathbb{E}_{z\sim\mathcal{N}_1}[g(z)] - \mathbb{E}_{z\sim\mathcal{N}_2}[g(z)]| &= \left|\mathbb{E}_{z\sim\mathcal{N}_1}\left[g(z)\left(1 - \frac{\mathcal{N}_2}{\mathcal{N}_1}\right)\right]\right| \\
&\leq \left|\sqrt{\mathbb{E}_{z\sim\mathcal{N}_1}[g^2(z)]}\sqrt{\mathbb{E}_{z\sim\mathcal{N}_1}\left[\left(1 - \frac{\mathcal{N}_2}{\mathcal{N}_1}\right)^2\right]}\right| \\
&= \sqrt{\mathbb{E}_{z\sim\mathcal{N}_1}[g^2(z)]}\sqrt{\mathbb{E}_{z\sim\mathcal{N}_1}\left[\left(1 - \frac{\mathcal{N}_2}{\mathcal{N}_1}\right)^2\right]} \\
&\leq \sqrt{\mathbb{E}_{z\sim\mathcal{N}_1}[g^2(z)]}\min\left\{\frac{\|\mu_1 - \mu_2\|}{\sigma^2}, 1\right\} \\
&\qquad\qquad\qquad \text{(Lemma C.2. Kakade et al. (2020))}
\end{aligned}
$$

$\square$

**Corollary 3** (Regret of optimistic planning for Gaussian noise). *Let $\boldsymbol{\pi}_n^*$, $\boldsymbol{\pi}_n$ denote the solution to Equation (6) and Equation (7) respectively, and $\boldsymbol{z}_{n,t}^*$, $\boldsymbol{z}_{n,t}$ the corresponding state-action pairs visited during their respective trajectories. Furthermore, let Assumption 1 - 3 hold. Then, the following is true for all $n \geq 0$, $t \in [0, T-1]$, with probability at least $1 - \delta$*

$$
J_n(\boldsymbol{\pi}_n^*) - J_n(\boldsymbol{\pi}_n) \leq \mathcal{O}\left(T\mathbb{E}_{\boldsymbol{\tau}^{\pi_n}}\left[\sum_{t=0}^{T-1}\frac{(1 + \sqrt{d_x})\beta_{n-1}(\delta)\|\boldsymbol{\sigma}_{n-1}(\boldsymbol{z}_{n,t})\|}{\sigma^2}\right]\right)
$$

*Proof.* For simplicity, define $g_n(\boldsymbol{x}) = J_{n,t+1}(\boldsymbol{\pi}_n, \boldsymbol{\eta}_n, \boldsymbol{x})$. Note since $\boldsymbol{w}_t \sim \mathcal{N}(0, \sigma^2 \mathbb{I})$ (Assumption 3), we have that $\boldsymbol{x}'_{t+1}$ and $\boldsymbol{x}_{t+1}$ are also Gaussians. Therefore, we can leverage Lemma 6.

$$\mathbb{E}_{\boldsymbol{\tau}^{\boldsymbol{\pi}_n}} \left[ J_{n,t+1}(\boldsymbol{\pi}_n, \boldsymbol{\eta}_n, \boldsymbol{x}'_{t+1}) - J_{n,t+1}(\boldsymbol{\pi}_n, \boldsymbol{\eta}_n, \boldsymbol{x}_{t+1}) \right] = \mathbb{E} \left[ g_n(\boldsymbol{x}'_{t+1}) - g_n(\boldsymbol{x}_{t+1}) \right]$$

$$\leq \sqrt{\mathbb{E}[g_n^2(\boldsymbol{x}_{t+1})]} \min \left\{ \frac{\|\boldsymbol{x}'_{t+1} - \boldsymbol{x}_{t+1}\|}{\sigma^2}, 1 \right\} \qquad \text{(Lemma 6)}$$

$$\leq \sqrt{J_{\max}} \min \left\{ \frac{\|\boldsymbol{x}'_{t+1} - \boldsymbol{x}_{t+1}\|}{\sigma^2}, 1 \right\} . \qquad \text{(Lemma 4)}$$

Furthermore,

$$\begin{aligned}
\|\boldsymbol{x}'_{t+1} - \boldsymbol{x}_{t+1}\| &= \|\boldsymbol{\mu}_{n-1}(\boldsymbol{x}_t, \boldsymbol{\pi}_n(\boldsymbol{x}_t)) + \beta_{n-1}(\delta)\boldsymbol{\sigma}_{n-1}(\boldsymbol{x}_t, \boldsymbol{\pi}_n(\boldsymbol{x}_t))\boldsymbol{\eta}_n(\boldsymbol{x}_t) - \boldsymbol{f}^*(\boldsymbol{x}_t, \boldsymbol{\pi}_n(\boldsymbol{x}_t))\| \\
&\leq \|\boldsymbol{\mu}_{n-1}(\boldsymbol{x}_t, \boldsymbol{\pi}_n(\boldsymbol{x}_t)) - \boldsymbol{f}^*(\boldsymbol{x}_t, \boldsymbol{\pi}_n(\boldsymbol{x}_t))\| \\
&\quad + \beta_{n-1}(\delta) \|\boldsymbol{\sigma}_{n-1}(\boldsymbol{x}_t, \boldsymbol{\pi}_n(\boldsymbol{x}_t))\| \|\boldsymbol{\eta}_n(\boldsymbol{x}_t)\| \\
&\leq (1 + \sqrt{d_x})\beta_{n-1}(\delta) \|\boldsymbol{\sigma}_{n-1}(\boldsymbol{x}_t, \boldsymbol{\pi}_n(\boldsymbol{x}_t))\| . \qquad \text{(Assumption 1)}
\end{aligned}$$

Next, we use Lemma 2

$$\begin{aligned}
J_n(\boldsymbol{\pi}_n^*) - J_n(\boldsymbol{\pi}_n) &\leq \mathbb{E}_{\boldsymbol{\tau}^{\boldsymbol{\pi}_n}} \left[ \sum_{t=0}^{T-1} J_{n,t+1}(\boldsymbol{\pi}_n, \boldsymbol{\eta}_n, \boldsymbol{x}'_{t+1}) - J_{n,t+1}(\boldsymbol{\pi}_n, \boldsymbol{\eta}_n, \boldsymbol{x}_{t+1}) \right], \\
&\leq \mathbb{E}_{\boldsymbol{\tau}^{\boldsymbol{\pi}_n}} \left[ \sum_{t=0}^{T-1} \sqrt{J_{\max}} \min \left\{ \frac{(1 + \sqrt{d_x})\beta_{n-1}(\delta) \|\boldsymbol{\sigma}_{n-1}(\boldsymbol{x}_t, \boldsymbol{\pi}_n(\boldsymbol{x}_t))\|}{\sigma^2}, 1 \right\} \right], \\
&\leq \sqrt{J_{\max}} \mathbb{E}_{\boldsymbol{\tau}^{\boldsymbol{\pi}_n}} \left[ \sum_{t=0}^{T-1} \frac{(1 + \sqrt{d_x})\beta_{n-1}(\delta) \|\boldsymbol{\sigma}_{n-1}(\boldsymbol{x}_t, \boldsymbol{\pi}_n(\boldsymbol{x}_t))\|}{\sigma^2} \right], \\
&= \mathcal{O}\left( T \mathbb{E}_{\boldsymbol{\tau}^{\boldsymbol{\pi}_n}} \left[ \sum_{t=0}^{T-1} \frac{(1 + \sqrt{d_x})\beta_{n-1}(\delta) \|\boldsymbol{\sigma}_{n-1}(\boldsymbol{x}_t, \boldsymbol{\pi}_n(\boldsymbol{x}_t))\|}{\sigma^2} \right] \right).
\end{aligned}$$

$\square$

**Lemma 7** (Regret of planning optimistically for sub-Gaussian noise)**.** *Let $\boldsymbol{\pi}_n^*$, $\boldsymbol{\pi}_n$ denote the solution to Equation* (6) *and Equation* (7) *respectively, and $\boldsymbol{z}_{n,t}^*$, $\boldsymbol{z}_{n,t}$ the corresponding state-action pairs visited during their respective trajectories. Furthermore, let Assumption 1 and 2 hold, and relax Assumption 3 to $\sigma$-sub Gaussian noise. Then, the following is true for all $n \geq 0$ with probability at least $1 - \delta$*

$$J_n(\boldsymbol{\pi}_n^*) - J_n(\boldsymbol{\pi}_n) \leq \mathcal{O}\left( L_{\boldsymbol{\sigma}}^{T-1} \beta_{n-1}^T(\delta) T \mathbb{E}_{\boldsymbol{\tau}^{\boldsymbol{\pi}_n}} \left[ \sum_{t=0}^{T-1} \|\boldsymbol{\sigma}_{n-1,j}(\boldsymbol{z}_{n,t})\| \right] \right)$$

*Proof.* Curi et al. (2020, Lemma 5) bound the regret with the sum of epistemic uncertainties for Lipschitz continuous reward functions, under Assumption 1 and 2 for sub-Gaussian noise (c.f., Rothfuss et al. (2023, Theorem 3.5) for a more rigorous derivation). For the active exploration setting, the reward in episode $n + 1$ is

$$r(\boldsymbol{z}) = \frac{1}{2} \sum_{j=1}^{d_x} \log \left( 1 + \frac{\boldsymbol{\sigma}_{n-1,j}^2(\boldsymbol{z})}{\sigma^2} \right).$$

We show in Lemma 4 that our choice of exploration reward is Lipschitz continuous. Thus, can use the regret bound from Curi et al. (2020). $\square$

Compared to the Gaussian case, $\sigma$-sub Gaussian noise has the additional exponential dependence on the horizon $T$, i.e., the $\beta_n^T$ term. This follows from the analysis through Lipschitz continuity. Moreover, as we show in Lemma 2, the regret of planning optimistically is proportional to the change in value under the same optimistic dynamics and policy, but different initial states. The Lipschitz continuity property of our objective allows us to relate the difference in values to the discrepancy in the trajectories. Even for linear systems, trajectories under the same dynamics and policy but different initial states can deviate exponentially in the horizon.

### A.3 Proof for general Bayesian models

In this section, we analyze the information gain for general Bayesian models and prove Theorem 1.

**Theorem 3** (Entropy of a RV with finite second moment is upper bounded by Gaussian entropy (Theorem 8.6.5 Cover and Thomas (2006))). *Let the random vector $\boldsymbol{x} \in \mathbb{R}^n$ have covariance $\boldsymbol{K} = \mathbb{E}\left[\boldsymbol{x}\boldsymbol{x}^\top\right]$ (i.e., $\boldsymbol{K}_{ij} = E\left[\boldsymbol{x}_i\boldsymbol{x}_j\right], 1 \leq i, j \leq n$). Then*

$$H(X) \leq \frac{1}{2}\log((2\pi e)^n|\boldsymbol{K}|)$$

*with equality if and only if $\boldsymbol{x} \sim \mathcal{N}(\boldsymbol{\mu}, \boldsymbol{K})$ for $\boldsymbol{\mu} = E\left[\boldsymbol{x}\right]$.*

**Lemma 8** (Monotonocity of information gain). *Let $\boldsymbol{\tau^\pi}$ denote the trajectory induced by the policy $\boldsymbol{\pi}$. Then, the following is true for all $n \geq 0$, policies $\boldsymbol{\pi}$*

$$\mathbb{E}_{\mathcal{D}_{1:n}}\left[I\left(\boldsymbol{f}_{\boldsymbol{\tau^\pi}}^*; \boldsymbol{y}_{\boldsymbol{\tau^\pi}} \mid \mathcal{D}_{1:n}\right)\right] \leq \mathbb{E}_{\mathcal{D}_{1:n-1}}\left[I\left(\boldsymbol{f}_{\boldsymbol{\tau^\pi}}^*; \boldsymbol{y}_{\boldsymbol{\tau^\pi}} \mid \mathcal{D}_{1:n-1}\right)\right].$$

*Proof.*

$$
\begin{aligned}
\mathbb{E}_{\mathcal{D}_{1:n}} &\left[I\left(\boldsymbol{f}_{\boldsymbol{\tau^\pi}}^*; \boldsymbol{y}_{\boldsymbol{\tau^\pi}} \mid \mathcal{D}_{1:n-1}\right) - I\left(\boldsymbol{f}_{\boldsymbol{\tau^\pi}}^*; \boldsymbol{y}_{\boldsymbol{\tau^\pi}} \mid \mathcal{D}_{1:n}\right)\right] \\
&= \mathbb{E}_{\mathcal{D}_{1:n}}\left[H\left(\boldsymbol{y}_{\boldsymbol{\tau^\pi}} \mid \mathcal{D}_{1:n-1}\right) - H\left(\boldsymbol{y}_{\boldsymbol{\tau^\pi}} \mid \boldsymbol{f}_{\boldsymbol{\tau^\pi}}^*, \mathcal{D}_{1:n-1}\right)\right. \\
&\quad \left. - \left(H\left(\boldsymbol{y}_{\boldsymbol{\tau^\pi}} \mid \mathcal{D}_{1:n}\right) - H\left(\boldsymbol{y}_{\boldsymbol{\tau^\pi}} \mid \boldsymbol{f}_{\boldsymbol{\tau^\pi}}^*, \mathcal{D}_{1:n}\right)\right)\right] \\
&= \mathbb{E}_{\mathcal{D}_{1:n}}\left[H\left(\boldsymbol{y}_{\boldsymbol{\tau^\pi}} \mid \mathcal{D}_{1:n-1}\right) - H\left(\boldsymbol{y}_{\boldsymbol{\tau^\pi}} \mid \mathcal{D}_{1:n}\right)\right] \\
&\quad + \mathbb{E}_{\mathcal{D}_{1:n}}\left[H\left(\boldsymbol{y}_{\boldsymbol{\tau^\pi}} \mid \boldsymbol{f}_{\boldsymbol{\tau^\pi}}^*\right) - H\left(\boldsymbol{y}_{\boldsymbol{\tau^\pi}} \mid \boldsymbol{f}_{\boldsymbol{\tau^\pi}}^*\right)\right] \\
&\geq 0 \qquad\qquad\qquad\qquad\qquad\qquad\qquad\text{(information never hurts)}
\end{aligned}
$$

$\square$

A direct consequence of Lemma 8 is the following corollary.

**Corollary 4** (Information gain at $N$ is less than the average gain till $N$). *Let $\boldsymbol{\tau^\pi}$ denote the trajectory induced by the policy $\boldsymbol{\pi}$. Then, the following is true for all $N \geq 1$, policies $\boldsymbol{\pi}$,*

$$\mathbb{E}_{\mathcal{D}_{1:N-1}}\left[I\left(\boldsymbol{f}_{\boldsymbol{\tau^\pi}}^*; \boldsymbol{y}_{\boldsymbol{\tau^\pi}} \mid \mathcal{D}_{1:N-1}\right)\right] \leq \frac{1}{N}\sum_{n=1}^N \mathbb{E}_{\mathcal{D}_{1:n-1}}\left[I\left(\boldsymbol{f}_{\boldsymbol{\tau^\pi}}^*; \boldsymbol{y}_{\boldsymbol{\tau^\pi}} \mid \mathcal{D}_{1:n-1}\right)\right].$$

Next, we prove Lemma 1, which is central to our proposed active exploration objective in Equation (6).

*Proof of Lemma 1.* Let $\boldsymbol{y}_{\boldsymbol{\tau^\pi}} = \{\boldsymbol{y}_t\}_{t=0}^{T-1} = \{\boldsymbol{f}_t^* + \boldsymbol{w}_t\}_{t=0}^{T-1}$, where $\boldsymbol{f}_t^* = \boldsymbol{f}^*(\boldsymbol{z}_t)$. Furthermore, denote with $\Sigma_n(\boldsymbol{f}_{0:T-1}^*)$ the covariance of $\boldsymbol{f}_{0:T-1}^*$.

$$
\begin{aligned}
I\left(\boldsymbol{f}_{\boldsymbol{\tau^\pi}}^*; \boldsymbol{y}_{\boldsymbol{\tau^\pi}} \mid \mathcal{D}_{1:n}\right) &= I\left(\boldsymbol{f}_{0:T-1}^*; \boldsymbol{y}_{0:T-1} \mid \mathcal{D}_{1:n}\right) \\
&= H\left(\boldsymbol{y}_{0:T-1} \mid \mathcal{D}_{1:n}\right) - H\left(\boldsymbol{y}_{0:T-1} \mid \boldsymbol{f}_{0:T-1}^*, \mathcal{D}_{1:n}\right) \\
&\leq \frac{1}{2}\log\left(\left|\sigma^2\mathbb{I} + \Sigma_n(\boldsymbol{f}_{0:T-1}^*)\right|\right) - \frac{1}{2}\log\left(\left|\sigma^2\mathbb{I}\right|\right) \qquad \text{(Theorem 3)} \\
&\leq \frac{1}{2}\log\left(\left|\text{diag}\left(\mathbb{I} + \sigma^{-2}\Sigma_n(\boldsymbol{f}_{0:T-1}^*)\right)\right|\right) \qquad \text{(Hadamard's inequality)} \\
&= \frac{1}{2}\sum_{t=0}^{T-1}\sum_{j=1}^{d_x}\log\left(1 + \frac{\boldsymbol{\sigma}_{n,j}^2(\boldsymbol{z}_t)}{\sigma^2}\right).
\end{aligned}
$$

$\square$

We can leverage the result from Lemma 1 to bound the average mutual information with the sum of epistemic uncertainties.

**Lemma 9** (Average information gain is less than sum of average epistemic uncertainties). *Let Assumption 3 hold and denote with $\bar{\pi}_N$ be the solution of Equation* (4). *Then, for all $N \geq 1$ and dataset $\mathcal{D}_{1:N}$ the following is true*

$$\frac{1}{N} \sum_{n=1}^{N} \mathbb{E}_{\mathcal{D}_{1:n-1}, \boldsymbol{\tau}^{\bar{\pi}_n}} \left[ I \left( \boldsymbol{f}^*_{\boldsymbol{\tau}^{\bar{\pi}_N}} ; \boldsymbol{y}_{\boldsymbol{\tau}^{\bar{\pi}_N}} \mid \mathcal{D}_{1:n-1} \right) \right]$$

$$\leq \frac{1}{N} \sum_{n=1}^{N} \mathbb{E}_{\mathcal{D}_{1:n-1}, \boldsymbol{\tau}^{\pi^*_n}} \left[ \sum_{t=0}^{T-1} \sum_{j=1}^{d_x} \left( \frac{1}{2} \log \left( 1 + \frac{\boldsymbol{\sigma}^2_{n,j}(\boldsymbol{z}^*_{n,t})}{\sigma^2} \right) \right) \right],$$

*where $z^*_{n,t}$ are the state-action tuples visited by the solution of Equation* (6), *i.e., $\boldsymbol{\pi}^*_n$.*

*Proof.*

$$\frac{1}{N} \sum_{n=1}^{N} \mathbb{E}_{\mathcal{D}_{1:n-1}, \boldsymbol{\tau}^{\bar{\pi}_N}} \left[ I \left( \boldsymbol{f}^*_{\boldsymbol{\tau}^{\bar{\pi}_N}} ; \boldsymbol{y}_{\boldsymbol{\tau}^{\bar{\pi}_N}} \mid \mathcal{D}_{1:n-1} \right) \right]$$

$$\leq \frac{1}{N} \sum_{n=1}^{N} \mathbb{E}_{\mathcal{D}_{1:n-1}, \boldsymbol{\tau}^{\bar{\pi}_N}} \left[ \left( \sum_{\boldsymbol{z}_t \in \boldsymbol{\tau}^{\bar{\pi}_N}} \frac{1}{2} \sum_{j=1}^{d_x} \log \left( 1 + \frac{\boldsymbol{\sigma}^2_{n,j}(\boldsymbol{z}_t)}{\sigma^2} \right) \right) \right] \qquad \text{(Lemma 1)}$$

$$\leq \frac{1}{N} \sum_{n=1}^{N} \mathbb{E}_{\mathcal{D}_{1:n-1}} \left[ \max_{\boldsymbol{\pi} \in \Pi} \mathbb{E}_{\boldsymbol{\tau}^{\pi}} \left[ \sum_{\boldsymbol{z}_t \in \boldsymbol{\tau}_{\boldsymbol{\pi}}} \sum_{j=1}^{d_x} \left( \frac{1}{2} \log \left( 1 + \frac{\boldsymbol{\sigma}^2_{n,j}(\boldsymbol{z}_t)}{\sigma^2} \right) \right) \right] \right] \qquad (1)$$

$$= \frac{1}{N} \sum_{n=1}^{N} \mathbb{E}_{\mathcal{D}_{1:n-1}, \boldsymbol{\tau}^{\pi_n}} \left[ \sum_{t=0}^{T-1} \sum_{j=1}^{d_x} \left( \frac{1}{2} \log \left( 1 + \frac{\boldsymbol{\sigma}^2_{n,j}(\boldsymbol{z}^*_{n,t})}{\sigma^2} \right) \right) \right].$$

Here (1) follows from the tower property. Note that the second expectation in (1) is wrt $\boldsymbol{\tau}^{\pi}$ conditioned on a realization of $\mathcal{D}_{1:n-1}$, where the conditioning is captured in the epistemic uncertainty $\boldsymbol{\sigma}_n(\cdot)$. $\quad\square$

We use the results from above, to prove Theorem 1.

*Proof of theorem 1.* Let $\bar{\pi}_n$ denote the solution to Equation (4) at iteration $n \geq 1$. We first relate the information gain from OPAX to the information gain of $\bar{\pi}_n$.

$$\mathbb{E}_{\mathcal{D}_{1:N-1}, \boldsymbol{\tau}^{\bar{\pi}_N}} \left[ I \left( \boldsymbol{f}^*_{\boldsymbol{\tau}^{\bar{\pi}_N}} ; \boldsymbol{y}_{\boldsymbol{\tau}^{\bar{\pi}_N}} \mid \mathcal{D}_{1:N-1} \right) \right]$$

$$\leq \frac{1}{N} \sum_{n=1}^{N} \mathbb{E}_{\mathcal{D}_{1:n-1}, \boldsymbol{\tau}^{\bar{\pi}_n}} \left[ I \left( \boldsymbol{f}^*_{\boldsymbol{\tau}^{\bar{\pi}_n}} ; \boldsymbol{y}_{\boldsymbol{\tau}^{\bar{\pi}_n}} \mid \mathcal{D}_{1:n-1} \right) \right] \qquad \text{(Corollary 4)}$$

$$\leq \frac{1}{N} \sum_{n=1}^{N} \mathbb{E}_{\mathcal{D}_{1:n-1}, \boldsymbol{\tau}^{\pi_n}} \left[ \left( \sum_{t=0}^{T-1} \sum_{j=1}^{d_x} \frac{1}{2} \log \left( 1 + \frac{\boldsymbol{\sigma}^2_{n-1,j}(\boldsymbol{z}^*_{n,t})}{\sigma^2} \right) \right) \right] \qquad \text{(Lemma 9)}$$

$$= \frac{1}{N} \sum_{n=1}^{N} \left( \mathbb{E}_{\mathcal{D}_{1:n-1}, \boldsymbol{\pi}_n} \left[ \sum_{t=0}^{T-1} \frac{1}{2} \sum_{j=1}^{d_x} \log \left( 1 + \frac{\boldsymbol{\sigma}^2_{n-1,j}(\boldsymbol{z}_{n,t})}{\sigma^2} \right) \right] + J_n(\boldsymbol{\pi}^*_n) - J_n(\boldsymbol{\pi}_n) \right)$$

$$\leq \frac{1}{N} \sum_{n=1}^{N} \mathbb{E}_{\mathcal{D}_{1:n-1}} \left[ \mathbb{E}_{\boldsymbol{\tau}_{\boldsymbol{\pi}_n}} \left[ \sum_{t=0}^{T-1} \frac{1}{2} \sum_{j=1}^{d_x} \log \left( 1 + \frac{\boldsymbol{\sigma}^2_{n-1,j}(\boldsymbol{z}_{n,t})}{\sigma^2} \right) \right] \right.$$

$$\left. + \mathcal{O} \left( \beta_{n-1}(\delta) T \mathbb{E}_{\boldsymbol{\tau}_{\boldsymbol{\pi}_n}} \left[ \sum_{t=0}^{T-1} \| \boldsymbol{\sigma}_{n-1}(\boldsymbol{z}_{n,t}) \|_2 \right] \right) \right] \qquad \text{(Corollary 3)}$$

In summary, the maximum expected mutual information at episode $N$ is less than the mutual information of OPAX and the sum of model epistemic uncertainties. Crucial to the proof is the regret

bound for optimistic planning from Corollary 3.

$$\frac{1}{N}\sum_{n=1}^{N}\mathbb{E}_{\mathcal{D}_{1:n-1}}\left[\mathbb{E}_{\boldsymbol{\tau}_{\boldsymbol{\pi}_n}}\left[\sum_{t=0}^{T-1}\frac{1}{2}\sum_{j=1}^{d_x}\log\left(1+\frac{\boldsymbol{\sigma}_{n-1,j}^2(\boldsymbol{z}_{n,t})}{\sigma^2}\right)\right]\right.$$

$$\left.+\mathcal{O}\left(T\beta_{n-1}(\delta)\mathbb{E}_{\boldsymbol{\tau}_{\boldsymbol{\pi}_n}}\left[\sum_{t=0}^{T-1}\|\boldsymbol{\sigma}_{n-1}(\boldsymbol{z}_{n,t})\|_2\right]\right)\right]$$

$$=\frac{1}{N}\sum_{n=1}^{N}\mathbb{E}_{\mathcal{D}_{1:n-1}}\left[\mathbb{E}_{\boldsymbol{\tau}_{\boldsymbol{\pi}_n}}\left[\sum_{t=0}^{T-1}\sum_{j=1}^{d_x}\log\left(\sqrt{1+\frac{\boldsymbol{\sigma}_{n-1,j}^2(\boldsymbol{z}_{n,t})}{\sigma^2}}\right)\right]\right.$$

$$\left.+\mathcal{O}\left(T\beta_{n-1}(\delta)\mathbb{E}_{\boldsymbol{\tau}_{\boldsymbol{\pi}_n}}\left[\sum_{t=0}^{T-1}\|\boldsymbol{\sigma}_{n-1}(\boldsymbol{z}_{n,t})\|_2\right]\right)\right]$$

$$\leq\frac{1}{N}\sum_{n=1}^{N}\mathbb{E}_{\mathcal{D}_{1:n-1}}\left[\mathbb{E}_{\boldsymbol{\tau}_{\boldsymbol{\pi}_n}}\left[\sum_{t=0}^{T-1}\sum_{j=1}^{d_x}\log\left(1+\frac{\boldsymbol{\sigma}_{n-1,j}(\boldsymbol{z}_{n,t})}{\sigma}\right)\right]\right.$$

$$\left.+\mathcal{O}\left(T\beta_{n-1}(\delta)\mathbb{E}_{\boldsymbol{\tau}_{\boldsymbol{\pi}_n}}\left[\sum_{t=0}^{T-1}\|\boldsymbol{\sigma}_{n-1}(\boldsymbol{z}_{n,t})\|_2\right]\right)\right]$$

$$\leq\frac{1}{N}\sum_{n=1}^{N}\mathbb{E}_{\mathcal{D}_{1:n-1}}\left[\mathbb{E}_{\boldsymbol{\tau}_{\boldsymbol{\pi}_n}}\left[\sum_{t=0}^{T-1}\sum_{j=1}^{d_x}\frac{\boldsymbol{\sigma}_{n-1,j}(\boldsymbol{z}_{n,t})}{\sigma}\right]\right.$$

$$\left.+\mathcal{O}\left(T\beta_{n-1}(\delta)\mathbb{E}_{\boldsymbol{\tau}_{\boldsymbol{\pi}_n}}\left[\sum_{t=0}^{T-1}\|\boldsymbol{\sigma}_{n-1}(\boldsymbol{z}_{n,t})\|_2\right]\right)\right]\qquad(\log(1+x)\leq x\text{ for }x\geq 0.)$$

$$\leq\mathcal{O}\left(\frac{1}{N}\sum_{n=1}^{N}\mathbb{E}_{\mathcal{D}_{1:n-1}}\left[T\beta_{n-1}(\delta)\mathbb{E}_{\boldsymbol{\tau}_{\boldsymbol{\pi}_n}}\left[\sum_{t=0}^{T-1}\|\boldsymbol{\sigma}_{n-1}(\boldsymbol{z}_{n,t})\|_2\right]\right]\right)$$

Above, we show that the maximum expected mutual information can be upper bounded with the sum of epistemic uncertainties for the states OPAX visits during learning. Finally, we further upper bound this with the model complexity measure.

$$\mathcal{O}\left(\frac{1}{N}\sum_{n=1}^{N}\mathbb{E}_{\mathcal{D}_{1:n-1}}\left[\beta_{n-1}(\delta)T\mathbb{E}_{\boldsymbol{\tau}_{\boldsymbol{\pi}_n}}\left[\sum_{t=0}^{T-1}\|\boldsymbol{\sigma}_{n-1}(\boldsymbol{z}_{n,t})\|_2\right]\right]\right)$$

$$=\mathcal{O}\left(\frac{1}{N}\sqrt{\left(\mathbb{E}_{\mathcal{D}_{1:N}}\left[\sum_{n=1}^{N}(T\beta_{n-1}(\delta))\mathbb{E}_{\boldsymbol{\tau}_{\boldsymbol{\pi}_n}}\left[\sum_{t=0}^{T-1}\|\boldsymbol{\sigma}_{n-1}(\boldsymbol{z}_{n,t})\|_2\right]\right]\right)^2}\right)$$

$$\leq\mathcal{O}\left(\frac{1}{N}T\beta_N(\delta))\sqrt{TN\mathbb{E}_{\mathcal{D}_{1:N}}\left[\sum_{n=1}^{N}\mathbb{E}_{\boldsymbol{\tau}_{\boldsymbol{\pi}_n}}\left[\sum_{t=0}^{T-1}\|\boldsymbol{\sigma}_{n}(\boldsymbol{z}_{n,t})\|_2^2\right]\right]}\right)$$

$$\leq\mathcal{O}\left(\beta_N(\delta)T^{3/2}\sqrt{\frac{\mathcal{MC}_N(\boldsymbol{f}^*)}{N}}\right)$$

□

Theorem 1 gives a bound on the maximum expected mutual information w.r.t. the model complexity. We can use concentration inequalities such as Markov, to give a high probability bound on the

information gain. In particular, we have for all $\epsilon > 0$

$$\Pr\left(I\left(\boldsymbol{f}^*_{\boldsymbol{\tau}^{\bar{\pi}_N}}; \boldsymbol{y}_{\boldsymbol{\tau}^{\bar{\pi}_N}} \mid \mathcal{D}_{1:N-1}\right) \geq \epsilon\right) \leq \frac{\mathbb{E}_{\mathcal{D}_{1:N-1}, \boldsymbol{\tau}^{\bar{\pi}_N}}\left[I\left(\boldsymbol{f}^*_{\boldsymbol{\tau}^{\bar{\pi}_N}}; \boldsymbol{y}_{\boldsymbol{\tau}^{\bar{\pi}_N}} \mid \mathcal{D}_{1:N-1}\right)\right]}{\epsilon}$$

$$\leq \mathcal{O}\left(T^{3/2} \beta_N(\delta) \sqrt{\frac{\mathcal{MC}_N(\boldsymbol{f}^*)}{N\epsilon^2}}\right).$$

### A.4 Proof of GP results

This section presents our results for the frequentist setting where the dynamics are modeled using GPs. Since the information gain has no meaning in the frequentist setting, we study the epistemic uncertainty of the GP models.

**Corollary 5** (Monotonicity of the variance). *For all $n \geq 0$, and policies $\boldsymbol{\pi}$ the following is true.*

$$\sum_{t=0}^{T-1} \sum_{j=1}^{d_x} \frac{1}{2} \log\left(1 + \frac{\sigma^2_{N-1,j}(\boldsymbol{z}_t)}{\sigma^2}\right) \leq \frac{1}{N} \sum_{n=1}^{N} \sum_{t=0}^{T-1} \sum_{j=1}^{d_x} \frac{1}{2} \log\left(1 + \frac{\boldsymbol{\sigma}^2_{n-1,j}(\boldsymbol{z}_t)}{\sigma^2}\right)$$

*Proof.* Follows directly due to the monotonicity of GP posterior variance. $\square$

Next, we prove that the trajectory of Equation (6) at iteration $n$ is upper-bounded with the maximum information gain.

**Lemma 10.** *Let Assumption 2 - 4 hold Then, for all $N \geq 1$, with probability at least $1 - \delta$, we have*

$$\max_{\boldsymbol{\pi} \in \Pi} \mathbb{E}_{\boldsymbol{\tau}_{\boldsymbol{\pi}}}\left[\left(\sum_{t=0}^{T-1} \sum_{j=1}^{d_x} \frac{1}{2} \log\left(1 + \frac{\sigma^2_{N,j}(\boldsymbol{z}_t)}{\sigma^2}\right)\right)\right] \leq \mathcal{O}\left(\beta_N(\delta) T^{3/2} \sqrt{\frac{\gamma_N}{N}}\right).$$

*Moreover, relax noise Assumption 3 to $\sigma$-sub Gaussian. Then, for all $N \geq 1$, with probability at least $1 - \delta$, we have*

$$\max_{\boldsymbol{\pi} \in \Pi} \mathbb{E}_{\boldsymbol{\tau}_{\boldsymbol{\pi}}}\left[\left(\sum_{t=0}^{T-1} \sum_{j=1}^{d_x} \frac{1}{2} \log\left(1 + \frac{\sigma^2_{N,j}(\boldsymbol{z}_t)}{\sigma^2}\right)\right)\right] \leq \mathcal{O}\left(L_{\boldsymbol{\sigma}}^T \beta_N^T(\delta) T^{3/2} \sqrt{\frac{\gamma_N}{N}}\right)$$

*Proof.* **Gaussian noise case**: Let $\boldsymbol{z}^*_{n,t}$ denote the state-action pair at time $t$ for the trajectory of Equation (6) at iteration $n \geq 1$ and $\boldsymbol{\pi}^*_n$ the corresponding policy.

$$\mathbb{E}_{\boldsymbol{\tau}_{\boldsymbol{\pi}^*_n}}\left[\left(\sum_{t=0}^{T-1} \sum_{j=1}^{d_x} \frac{1}{2} \log\left(1 + \frac{\sigma^2_{N,j}(\boldsymbol{z}^*_{N,t})}{\sigma^2}\right)\right)\right]$$

$$\leq \frac{1}{N} \sum_{n=1}^{N} \mathbb{E}_{\boldsymbol{\tau}_{\boldsymbol{\pi}^*_n}}\left[\sum_{t=0}^{T-1} \sum_{j=1}^{d_x} \frac{1}{2} \log\left(1 + \frac{\sigma^2_{n,j}(\boldsymbol{z}^*_{N,t})}{\sigma^2}\right)\right] \qquad \text{(Corollary 5)}$$

$$\leq \frac{1}{N} \sum_{n=1}^{N} \mathbb{E}_{\boldsymbol{\tau}_{\boldsymbol{\pi}^*_n}}\left[\sum_{t=0}^{T-1} \sum_{j=1}^{d_x} \frac{1}{2} \log\left(1 + \frac{\sigma^2_{n,j}(\boldsymbol{z}^*_{n,t})}{\sigma^2}\right)\right] \qquad \text{(By definition of } \boldsymbol{\pi}^*_n\text{)}$$

$$\leq \mathcal{O}\left(\beta_N(\delta) T^{3/2} \sqrt{\frac{\mathcal{MC}_N(\boldsymbol{f}^*)}{N}}\right) \qquad \text{(See proof of Theorem 1)}$$

$$\leq \mathcal{O}\left(\beta_N(\delta) T^{3/2} \sqrt{\frac{\gamma_N}{N}}\right) \qquad \text{(Curi et al., 2020, Lemma 17)}$$

**Sub-Gaussian noise case**: The only difference between the Gaussian and sub-Gaussian case is the regret term (c.f., Corollary 3 and Lemma 7). In particular, the regret for the sub-Gaussian case leverages the Lipschitz continuity properties of the system (Assumption 2). This results in an exponential dependence on the horizon for our bound. We refer the reader to Curi et al. (2020); Rothfuss et al. (2023) for a more detailed derivation. $\square$

Lemma 10 gives a sample complexity bound that holds for a richer class of kernels. Moreover, for GP models, $\beta_N \propto \sqrt{\gamma_N}$ (Chowdhury and Gopalan, 2017). Therefore, for kernels, where $\lim_{N\to\infty} \gamma_N^2/N \to 0$, we can show convergence (for the Gaussian case). We summarize bounds on $\gamma_N$ from Vakili et al. (2021) in Table 1.

Table 1: Maximum information gain bounds for common choice of kernels.

| Kernel | $k(\boldsymbol{x}, \boldsymbol{x}')$ | $\gamma_n$ |
|---|---|---|
| Linear | $\boldsymbol{x}^\top \boldsymbol{x}'$ | $\mathcal{O}\left(d\log(n)\right)$ |
| RBF | $e^{-\frac{\|\boldsymbol{x}-\boldsymbol{x}'\|^2}{2l^2}}$ | $\mathcal{O}\left(\log^{d+1}(n)\right)$ |
| Matèrn | $\frac{1}{\Gamma(\nu)2^{\nu-1}}\left(\frac{\sqrt{2\nu}\|\boldsymbol{x}-\boldsymbol{x}'\|}{l}\right)^\nu B_\nu\left(\frac{\sqrt{2\nu}\|\boldsymbol{x}-\boldsymbol{x}'\|}{l}\right)$ | $\mathcal{O}\left(n^{\frac{d}{2\nu+d}}\log^{\frac{2\nu}{2\nu+d}}(n)\right)$ |

From hereon, we focus on deriving the results for the case of Gaussian noise case. All our results can be easily extended for the sub-Gaussian setting by considering its corresponding bound.

**Lemma 11.** *The following is true for all $N \geq 0$ and policies $\boldsymbol{\pi} \in \Pi$,*

$$\mathbb{E}_{\boldsymbol{\tau}_{\boldsymbol{\pi}}}\left[\max_{\boldsymbol{z}\in\boldsymbol{\tau}^{\boldsymbol{\pi}}}\sum_{j=1}^{d_x}\frac{1}{2}\boldsymbol{\sigma}_{N,j}^2(\boldsymbol{z})\right] \leq C_{\boldsymbol{\sigma}}\mathbb{E}_{\boldsymbol{\tau}_{\boldsymbol{\pi}}}\left[\sum_{t=0}^{T-1}\sum_{j=1}^{d_x}\frac{1}{2}\log\left(1+\frac{\boldsymbol{\sigma}_{N,j}^2(\boldsymbol{z}_t)}{\sigma^2}\right)\right],$$

*with $C_{\boldsymbol{\sigma}} = \frac{\boldsymbol{\sigma}_{\max}}{\log(1+\sigma^{-2}\boldsymbol{\sigma}_{\max})}$.*

*Proof.*

$$C_{\boldsymbol{\sigma}}\mathbb{E}_{\boldsymbol{\tau}_{\boldsymbol{\pi}}}\left[\sum_{t=0}^{T-1}\sum_{j=1}^{d_x}\frac{1}{2}\log\left(1+\frac{\boldsymbol{\sigma}_{N,j}^2(\boldsymbol{z}_t)}{\sigma^2}\right)\right]$$

$$\geq \mathbb{E}_{\boldsymbol{\tau}_{\boldsymbol{\pi}}}\left[\sum_{t=0}^{T-1}\sum_{j=1}^{d_x}\frac{1}{2}\boldsymbol{\sigma}_{N,j}^2(\boldsymbol{z}_t)\right], \qquad \text{(Curi et al., 2020, Lemma. 15)}$$

$$\geq \mathbb{E}_{\boldsymbol{\tau}_{\boldsymbol{\pi}}}\left[\max_{\boldsymbol{z}\in\boldsymbol{\tau}^{\boldsymbol{\pi}}}\sum_{j=1}^{d_x}\frac{1}{2}\boldsymbol{\sigma}_{N,j}^2(\boldsymbol{z})\right].$$

$\square$

**Corollary 6.** *Let Assumption 2 and 4 hold, and relax noise Assumption 3 to $\sigma$-sub Gaussian. Then, for all $N \geq 1$, with probability at least $1 - \delta$, we have*

$$\max_{\boldsymbol{\pi}\in\Pi}\mathbb{E}_{\boldsymbol{\tau}_{\boldsymbol{\pi}}}\left[\max_{\boldsymbol{z}\in\boldsymbol{\tau}^{\boldsymbol{\pi}}}\sum_{j=1}^{d_x}\frac{1}{2}\boldsymbol{\sigma}_{N,j}^2(\boldsymbol{z})\right] \leq \mathcal{O}\left(\beta_N(\delta)T^{3/2}\sqrt{\frac{\gamma_N}{N}}\right).$$

**Lemma 12.** *Let Assumption 2 and 4 hold, and relax noise Assumption 3 to $\sigma$-sub Gaussian. Furthermore, assume $\lim_{N\to\infty}\beta_N^2(\delta)\gamma_N(k)/N \to 0$. Then for all $N \geq 1$, $\boldsymbol{z} \in \mathcal{R}$, and $1 \leq j \leq d_x$, with probability at least $1 - \delta$, we have*

$$\boldsymbol{\sigma}_{n,j}(\boldsymbol{z}) \xrightarrow{\text{a.s.}} 0 \text{ for } n \to \infty.$$

*Proof.* We first show that the expected epistemic uncertainty along a trajectory converges to zero almost surely. Then we leverage this result to show almost sure convergence of all trajectories induced

by $\boldsymbol{\pi} \in \Pi$. To this end, let $S_n = \mathbb{E}_{\boldsymbol{\tau}_{\boldsymbol{\pi}_n^*}} \left[ \left( \sum_{t=0}^{T-1} \sum_{j=1}^{d_x} \frac{1}{2} \log \left( 1 + \frac{\sigma_{n,j}^2(\boldsymbol{z}_{n,t}^*)}{\sigma^2} \right) \right) \right]$ for all $n \geq 0$. So far we have,

$$\Pr \left( S_n \leq \mathcal{O} \left( \beta_N(\delta) T^{3/2} \sqrt{\frac{\gamma_n}{n}} \right) \right) \geq 1 - \delta$$

Consider a sequence $\{\delta_n\}_{n \geq 0}$ such that $\lim_{n \to \infty} \delta_n = 0$, and $\lim_{n \to \infty} \beta_n(\delta_n) T^{3/2} \sqrt{\frac{\gamma_n}{n}} \to 0$. Note, for GP models with $\lim_{N \to \infty} \beta_N^2(\delta) \gamma_N(k)/N \to 0$, such a sequence of $\delta_n$ exists (Chowdhury and Gopalan, 2017). Consider any $\epsilon > 0$ and let $N^*(\epsilon)$ be the smallest integer such that

$$\mathcal{O} \left( \beta_{N^*(\epsilon)}(\delta) T^{3/2} \sqrt{\frac{\gamma_{N^*(\epsilon)}}{N^*(\epsilon)}} \right) < \epsilon.$$

Then, we have

$$\sum_{n=0}^{\infty} \Pr(S_n > \epsilon) = \sum_{n=0}^{N^*(\epsilon)-1} \Pr(S_n > \epsilon) + \sum_{n=N^*(\epsilon)}^{\infty} \Pr(S_n > \epsilon)$$

$$= \sum_{n=0}^{N^*(\epsilon)-1} \Pr(S_n > \epsilon) + \sum_{n=N^*(\epsilon)}^{\infty} \delta_n$$

$$\leq N^*(\epsilon) + \sum_{n=N^*(\epsilon)}^{\infty} \delta_n.$$

Note, since $\lim_{n \to \infty} \delta_n = 0$, we have

$$\sum_{n=N^*(\epsilon)}^{\infty} \delta_n < \infty.$$

In particular, $\sum_{n=0}^{\infty} \Pr(S_n > \epsilon) < \infty$ for all $\epsilon > 0$. Therefore, we obtain

$$S_n \xrightarrow{\text{a.s.}} 0 \text{ for } n \to \infty.$$

Define the random variable $V = \lim_{n \to \infty} \left( \sum_{t=0}^{T-1} \sum_{j=1}^{d_x} \frac{1}{2} \log \left( 1 + \frac{\sigma_{n,j}^2(\boldsymbol{z}_{n,t}^*)}{\sigma^2} \right) \right)$, with $\boldsymbol{z}_{n,t}^* \in \boldsymbol{\tau}$ and $\boldsymbol{\tau} \sim \boldsymbol{\tau}^{\boldsymbol{\pi}_n^*}$. $V$ represents the sum of epistemic uncertainties of a random trajectory induced by the sequence of policies $\{\boldsymbol{\pi}_n\}_{n \geq 0}$. Note $V \geq 0$, therefore we apply Markov's inequality. Moreover, for all $\epsilon > 0$, we have

$$\Pr(V > \epsilon) \leq \frac{\mathbb{E}[V]}{\epsilon} = 0.$$

Hence, we have

$$\Pr(V = 0) = 1 \implies V \xrightarrow{\text{a.s.}} 0 \text{ for } n \to \infty$$

Accordingly, we get for all $\boldsymbol{\pi} \in \Pi$.

$$\Pr \left( \lim_{n \to \infty} \sum_{\boldsymbol{z}_t \in \boldsymbol{\tau}_{\boldsymbol{\pi}}} \sum_{j=1}^{d_x} \frac{1}{2} \log \left( 1 + \frac{\sigma_{n,j}^2(\boldsymbol{z}_t)}{\sigma^2} \right) \to 0 \right) = 1. \tag{15}$$

Assume there exists a $\boldsymbol{z} \in \mathcal{R}$, such that for some $\epsilon$, $\sigma_{n,j}^2(\boldsymbol{z}) > \epsilon$ for all $n \geq 0$. Since, $\boldsymbol{z} \in \mathcal{R}$, there exists a $t$ and $\boldsymbol{\pi} \in \Pi$ such that $p(\boldsymbol{z}_t = \boldsymbol{z} | \pi, \boldsymbol{f}^*) > 0$. This implies that $\Pr(\boldsymbol{z} \in \boldsymbol{\tau}_{\boldsymbol{\pi}}) > 0$. However, from Equation (15), we have that $\sigma_{n,j}^2(\boldsymbol{z}) \to 0$ for $N \to \infty$ almost surely, which is a contradiction. $\quad \square$

Finally, we leverage the results from above to prove Theorem 2.

*Proof of Theorem 2.* The proof follows directly from Corollary 6 and Lemma 12. $\quad \square$

## A.5 Zero-shot guarantees

In this section, we give guarantees on the zero-shot performance of OPAX for a bounded cost function. We focus this section on the case of Gaussian noise. However, a similar analysis can be performed for the sub-Gaussian case and Lipschitz continuous costs. Since the analysis for both cases is similar, we only present the Gaussian case with bounded costs here.

**Corollary 7.** *Consider the following optimal control problem*

$$\underset{\boldsymbol{\pi} \in \Pi}{\arg\min}\, J_c(\boldsymbol{\pi}, \boldsymbol{f}^*) = \underset{\boldsymbol{\pi} \in \Pi}{\arg\min}\, \mathbb{E}_{\boldsymbol{\tau}^{\boldsymbol{\pi}}} \left[ \sum_{t=0}^{T-1} c(\boldsymbol{x}_t, \boldsymbol{\pi}(\boldsymbol{x}_t)) \right], \tag{16}$$
$$\boldsymbol{x}_{t+1} = \boldsymbol{f}^*(\boldsymbol{x}_t, \boldsymbol{\pi}(\boldsymbol{x}_t)) + \boldsymbol{w}_t \quad \forall 0 \le t \le T,$$

*with bounded and positive costs. Then we have for all policies $\boldsymbol{\pi}$ with probability at least $1 - \delta$*

$$J_c(\boldsymbol{\pi}, \boldsymbol{\eta}^P) - J_c(\boldsymbol{\pi}) \le \mathcal{O}\left( T \mathbb{E}_{\boldsymbol{\tau}^{\boldsymbol{\pi}_n}} \left[ \sum_{t=0}^{T-1} \frac{(1 + \sqrt{d_x}) \beta_{n-1}(\delta) \|\boldsymbol{\sigma}_{n-1}(\boldsymbol{z}_t)\|}{\sigma^2} \right] \right),$$

*where $J_c(\boldsymbol{\pi}, \boldsymbol{\eta}^P) = \max_{\boldsymbol{\eta} \in \Xi} J_c(\boldsymbol{\pi}, \boldsymbol{\eta})$.*

*Proof.* From Corollary 2 we get

$$J_c(\boldsymbol{\pi}, \boldsymbol{\eta}^P) - J_c(\boldsymbol{\pi}) = \mathbb{E}_{\boldsymbol{\tau}^{\boldsymbol{\pi}}} \left[ \sum_{t=0}^{T-1} J_{r,t+1}(\boldsymbol{\pi}, \boldsymbol{\eta}^P, \boldsymbol{x}_{t+1}^P) - J_{r,t+1}(\boldsymbol{\pi}, \boldsymbol{\eta}^P, \boldsymbol{x}_{t+1}) \right],$$

with $\boldsymbol{x}_{t+1} = \boldsymbol{f}^*(\boldsymbol{x}_t, \boldsymbol{\pi}(\boldsymbol{x}_t)) + \boldsymbol{w}_t$, and $\boldsymbol{x}_{t+1}^P = \boldsymbol{\mu}_n(\boldsymbol{x}_t, \boldsymbol{\pi}(\boldsymbol{x}_t)) + \beta_n(\delta)\boldsymbol{\sigma}(\boldsymbol{x}_t, \boldsymbol{\pi}(\boldsymbol{x}_t))\boldsymbol{\eta}^P(\boldsymbol{x}_t) + \boldsymbol{w}_t$. Furthermore, the cost is positive and bounded. Therefore,

$$J_c^2(\boldsymbol{\pi}, \boldsymbol{\eta}, \boldsymbol{x}) \le T^2 c_{\max}^2,$$

for all $\boldsymbol{x}, \boldsymbol{\eta}, \boldsymbol{\pi}$. Accordingly, we can now use the same analysis as in Lemma 2 and get

$$J_c(\boldsymbol{\pi}, \boldsymbol{\eta}^P) - J_c(\boldsymbol{\pi}) \le \mathcal{O}\left( T \mathbb{E}_{\boldsymbol{\tau}^{\boldsymbol{\pi}_n}} \left[ \sum_{t=0}^{T-1} \frac{(1 + \sqrt{d_x}) \beta_{n-1}(\delta) \|\boldsymbol{\sigma}_{n-1}(\boldsymbol{z}_t)\|}{\sigma^2} \right] \right),$$

$\square$

**Lemma 13.** *Consider the control problem in Equation (16) and let Assumption 3 and 4 hold. Furthermore, assume for every $\epsilon > 0$, there exists a finite integer $n^*$ such that*

$$\forall n \ge n^*;\, \beta_n^{3/2}(\delta) T^{11/4} \left( \frac{\gamma_n(k)}{n} \right)^{\frac{1}{4}} \le \epsilon, \tag{17}$$

*and denote with $\hat{\boldsymbol{\pi}}_n$ the minimax optimal policy, i.e., the solution to $\min_{\boldsymbol{\pi} \in \Pi} \max_{\boldsymbol{\eta} \in \Xi} J_c(\boldsymbol{\pi}, \boldsymbol{\eta})$. Then for all $n \ge n^*$, we have probability at least $1 - \delta$, $J_c(\hat{\boldsymbol{\pi}}_N) - J_c(\boldsymbol{\pi}^*) \le \mathcal{O}(\epsilon)$.*

*Proof of Zero-shot performance.* In Theorem 2 we give a rate at which the maximum uncertainty along a trajectory decreases:

$$\max_{\boldsymbol{\pi} \in \Pi} \mathbb{E}_{\boldsymbol{\tau}^{\boldsymbol{\pi}}} \left[ \max_{\boldsymbol{z} \in \boldsymbol{\tau}^{\boldsymbol{\pi}}} \frac{1}{2} \|\sigma_{N,j}(\boldsymbol{z})\|^2 \right] \le \mathcal{O}\left( \beta_N T^{3/2} \sqrt{\frac{\gamma_N(k)}{N}} \right).$$

Combining this with Corollary 7 we get

$$
J_c(\boldsymbol{\pi}, \boldsymbol{\eta}^P) - J_c(\boldsymbol{\pi}) \le \mathcal{O}\left( T\mathbb{E}_{\boldsymbol{\tau}^{\boldsymbol{\pi}_n}}\left[ \sum_{t=0}^{T-1} \frac{(1+\sqrt{d_x})\beta_{n-1}(\delta)\,\|\boldsymbol{\sigma}_{n-1}(\boldsymbol{z}_t)\|}{\sigma^2} \right] \right)
$$

$$
\le \mathcal{O}\left( T^2\beta_{n-1}(\delta)\mathbb{E}_{\boldsymbol{\tau}^{\boldsymbol{\pi}_n}}\left[ \max_{\boldsymbol{z}\in\boldsymbol{\tau}^{\boldsymbol{\pi}_n}} \|\boldsymbol{\sigma}_{n-1}(\boldsymbol{z})\| \right] \right)
$$

$$
\le \mathcal{O}\left( \sqrt{\mathbb{E}_{\boldsymbol{\tau}^{\boldsymbol{\pi}_n}}\left[ \max_{\boldsymbol{z}\in\boldsymbol{\tau}^{\boldsymbol{\pi}_n}} T^4\beta_{n-1}^2(\delta)\,\|\boldsymbol{\sigma}_{n-1}(\boldsymbol{z})\|^2 \right]} \right)
$$

$$
\le \mathcal{O}\left( \sqrt{\beta_n^3 T^{11/2}\sqrt{\frac{\gamma_n(k)}{n}}} \right)
$$

$$
= \mathcal{O}\left( \beta_n^{3/2}(\delta)T^{11/4}\left(\frac{\gamma_n(k)}{n}\right)^{\frac{1}{4}} \right)
$$

$$
= \mathcal{O}(\epsilon). \qquad\qquad (\forall n \ge n^*)
$$

Hence, we have that for each policy $\boldsymbol{\pi}$, our upper bound $\max_{\boldsymbol{\eta}} J_c(\boldsymbol{\pi}, \boldsymbol{\eta}^P)$ is $\epsilon$ precise, i.e.,

$$
\max_{\boldsymbol{\eta}\in\Xi} J_c(\boldsymbol{\pi}, \boldsymbol{\eta}) - J_c(\boldsymbol{\pi}) \le \mathcal{O}(\epsilon), \forall \boldsymbol{\pi} \in \Pi. \qquad (18)
$$

We leverage this to prove optimality for the minimax solution. For the sake of contradiction, assume that

$$
J_c(\hat{\boldsymbol{\pi}}_n) > J_c(\boldsymbol{\pi}^*) + \mathcal{O}(\epsilon). \qquad (19)
$$

Then we have,

$$
\max_{\boldsymbol{\eta}\in\Xi} J_c(\boldsymbol{\pi}^*, \boldsymbol{\eta}) \ge \min_{\boldsymbol{\pi}\in\Pi} \max_{\boldsymbol{\eta}\in\Xi} J_c(\boldsymbol{\pi}, \boldsymbol{\eta})
$$

$$
= J_c(\hat{\boldsymbol{\pi}}_n, \hat{\boldsymbol{\eta}}^P)
$$

$$
= \max_{\boldsymbol{\eta}\in\Xi} J_c(\hat{\boldsymbol{\pi}}_n, \boldsymbol{\eta})
$$

$$
\ge J_c(\hat{\boldsymbol{\pi}}_n)
$$

$$
> J_c(\boldsymbol{\pi}^*) + \mathcal{O}(\epsilon) \qquad \text{(Equation (19))}
$$

$$
\ge J_c(\boldsymbol{\pi}^*, \boldsymbol{\eta}^{*,P}) \qquad \text{(Equation (18))}
$$

$$
= \max_{\boldsymbol{\eta}\in\Xi} J_c(\boldsymbol{\pi}^*, \boldsymbol{\eta}).
$$

This is a contradiction, which completes the proof. $\qquad\square$

Lemma 13 shows that OPAX also results in nearly-optimal zero-shot performance. The convergence criteria in Equation (17) is satisfied for kernels $k$ that induce a very rich class of RKHS (c.f., Table 1).

## B  Experiment Details

The environment details and hyperparameters used for our experiments are presented in this section. In Appendix B.2 we discuss the experimental setup of the Fetch Pick & Construction environment (Li et al., 2020) in more detail.

### B.1  Environment Details

Table 2 lists the rewards used for the different environments. We train the dynamics model after each episode of data collection. For training, we fix the number of epochs to determine the number of gradient steps. Furthermore, for computational reasons, we upper bound the number of gradient steps by "maximum number of gradient steps". The hyperparameters for the model-based agent are presented in Table 3. Furthermore, we present the iCEM hyperparameters in Table 4. For more detail on the iCEM hyperparameters see Pinneri et al. (2021).

Table 2: Downstream task rewards for the environments presented in Section 5.

| Environment | Reward $r_t$ |
|---|---|
| Pendulum-v1 - swing up | $\theta_t^2 + 0.1\dot{\theta}_t + 0.001u_t^2$ |
| Pendulum-v1 - keep down | $(\theta_t - \pi)^2 + 0.1\dot{\theta}_t + 0.001u_t^2$ |
| MountainCar | $-0.1u_t^2 + 100(1\{\boldsymbol{x}_t \in \boldsymbol{x}_{\text{goal}}\})$ |
| Reacher - go to target | $100(1\{\|\boldsymbol{x}_t - \boldsymbol{x}_{\text{target}}\|_2 \leq 0.05\}$ |
| Swimmer - go to target | $-\|\boldsymbol{x}_t - \boldsymbol{x}_{\text{target}}\|_2$ |
| Swimmer - go away from target | $\|\boldsymbol{x}_t - \boldsymbol{x}_{\text{target}}\|_2$ |
| Cheetah - run forward | $v_{\text{forward},t}$ |
| Cheetah - run backward | $-v_{\text{forward},t}$ |

Table 3: Hyperparameters for results in Section 5.

| Hyperparameters | Pendulum-v1 - GP | Pendulum-v1 | MountainCar | Reacher | Swimmer | Cheetah |
|---|---|---|---|---|---|---|
| Action repeat | N/A | N/A | N/A | N/A | 4 | 4 |
| Exploration horizon | 100 | 200 | 200 | 50 | 1000 | 1000 |
| Downstream task horizon | 200 | 200 | 200 | 50 | 1000 | 1000 |
| Hidden layers | N/A | 2 | 2 | 2 | 4 | 4 |
| Neurons per layers | N/A | 256 | 128 | 256 | 256 | 256 |
| Number of ensembles | N/A | 7 | 5 | 5 | 5 | 5 |
| Batch size | N/A | 64 | 64 | 64 | 64 | 64 |
| Learning rate | 0.1 | $5 \times 10^{-4}$ | $5 \times 10^{-4}$ | $10^{-3}$ | $5 \times 10^{-4}$ | $5 \times 10^{-5}$ |
| Number of epochs | 50 | 50 | 50 | 50 | 50 | 50 |
| Maximum number of gradient steps | N/A | 5000 | 5000 | 6000 | 7500 | 7500 |
| $\beta_n$ | 2.0 | 2.0 | 1.0 | 1.0 | 2.0 | 2.0 |

**Model-based SAC optimizer**   For the reacher and MountainCar environment we use scarce rewards (c.f., Table 2), which require long-term planning. Therefore, a receding horizon MPC approach is less suitable for these tasks. Accordingly, we use a policy network which we train using SAC (Haarnoja et al., 2018). Moreover, our model-based SAC uses transitions simulated through the learned model to train a policy. Accordingly, given a dataset of transitions from the true environment, we sample $P$ initial states from the dataset. For each of the states, we simulate a trajectory of $H$ steps using our learned model and the SAC policy. We collect the simulated transitions into a simulation data buffer $\mathcal{D}_{\text{SIM}}$, which we then use to perform $G$ gradient steps as suggested by Haarnoja et al. (2018) to train the policy. The algorithm is summarized below, and we provide the SAC hyperparameters in Table 5.

Table 4: Parameters of iCEM optimizer for experiments in Section 5.

| Hyperparameters | Pendulum-v1 - GP | Pendulum-v1 | Swimmer | Cheetah |
|---|---|---|---|---|
| Number of samples $P$ | 500 | 500 | 250 | 200 |
| Horizon $H$ | 20 | 20 | 15 | 10 |
| Size of elite-set $K$ | 50 | 50 | 25 | 20 |
| Colored-noise exponent $\beta$ | 0.25 | 0.25 | 0.25 | 0.25 |
| Number of particles | 10 | 10 | 10 | 10 |
| *CEM-iterations* | 10 | 10 | 5 | 5 |
| Fraction of elites reused $\xi$ | 0.3 | 0.3 | 0.3 | 0.3 |

**Model-based SAC**

---

**Init:** Stastistical Model $M = (\boldsymbol{\mu}_n, \boldsymbol{\sigma}_n, \beta_n(\delta))$, Dataset of transitions $\mathcal{D}_n$, initial policy $\pi_0$, Model Rollout steps $H$

$\mathcal{D}_{\text{SIM}} \leftarrow \emptyset$  ➤ Initialize simulated transitions buffer

**for** Training steps $k = 1, \ldots, K$ **do**

$\qquad \boldsymbol{x}_{0,1:P} \sim \mathcal{D}_n$  ➤ Sample $P$ initial states from buffer

$\quad$ **for** Initial state $\boldsymbol{x}_0 \in \boldsymbol{x}_{0,1:P}$ **do**

$\qquad \mathcal{D}_{\text{SIM}} \leftarrow \mathcal{D}_{\text{SIM}} \cup \text{MODELROLLOUT}(\boldsymbol{\pi}_k, \boldsymbol{x}_0, H, M)$  ➤ Collect simulated transitions

$\qquad$ Perform $G$ gradient updates on $\boldsymbol{\pi}_k$ as proposed in Haarnoja et al. (2018) using $\mathcal{D}_{\text{SIM}}$.

---

Table 5: Parameters of model-based SAC optimizer for experiments in Section 5.

| Parameters | MountainCar | Reacher |
|---|---|---|
| Discount factor | 0.99 | 0.99 |
| Learning rate actor | $5 \times 10^{-4}$ | $5 \times 10^{-4}$ |
| Learning rate critic | $5 \times 10^{-4}$ | $5 \times 10^{-4}$ |
| Learning rate entropy coefficient | $5 \times 10^{-4}$ | $5 \times 10^{-4}$ |
| Actor architecture | $[64, 64]$ | $[250, 250]$ |
| Critic architecture | $[256, 256]$ | $[250, 250]$ |
| Batch size | 32 | 64 |
| Gradient steps $G$ | 350 | 500 |
| Simulation horizon $H$ | 4 | 10 |
| Initial state sample size $P$ | 500 | 2000 |
| Total SAC training steps $K$ | 35 | 350 |

## B.2 OPAX in the High-dimensional Fetch Pick & Place Environment

### B.2.1 Environment and Model Details

In our experiments, we use an extension of the Fetch Pick & Place environment to multiple objects as proposed in Li et al. (2020) and further modified in Sancaktar et al. (2022) with the addition of a large table. This is a compositional object manipulation environment with an end-effector-controlled robot arm. The robot actions $\boldsymbol{u} \in \mathbb{R}^4$ correspond to the gripper movement in Cartesian coordinates and the gripper opening/closing. The robot state $\boldsymbol{x}_{\text{robot}} \in \mathbb{R}^{10}$ contains positions and velocities of the end-effector as well as the gripper-state (open/close) and gripper-velocity. Each object's state $\boldsymbol{x}_{\text{obj}} \in \mathbb{R}^{12}$ is given by its position, orientation (in Euler angles), and linear and angular velocities.

We follow the free-play paradigm as used in CEE-US (Sancaktar et al., 2022). At the beginning of free play, an ensemble of world models is randomly initialized with an empty replay buffer. At each iteration of free play, a certain number of rollouts (here: 20 rollouts with 100 timesteps each) are collected and then added to the replay buffer. Afterwards, the models in the ensemble are trained for a certain number of epochs on the collected data so far. Note that unlike in the original proposal by Sancaktar et al. (2022), we use Multilayer Perceptrons (MLP) as world models instead of Graph Neural Networks (GNN), for the sake of reducing run-time. This corresponds to the ablation MLP + iCEM presented in Sancaktar et al. (2022), which was shown to outperform all the baselines other than CEE-US with GNNs. As we are interested in exploring the difference in performance via injection of optimism into active exploration, we use the computationally less heavy MLPs in our work. This is reflected in the downstream task performance we report compared to the original CEE-US with GNNs. Details for the environment and models are summarized in Table 6.

After the free-play phase, we use the trained models to solve downstream tasks zero-shot via model-based planning with iCEM. We test for the tasks pick & place, throwing and flipping with 4 objects. The rewards used for these tasks are the same as presented in Sancaktar et al. (2022).

Table 6: Environment and model settings used for the experiment results shown in Figure 4.

(a) Fetch Pick & Place settings.

| Parameter | Value |
|---|---|
| Episode Length | 100 |
| Train Model Every | 20 Ep. |
| Action Dim. | 4 |
| Robot/Agent State Dim. | 10 |
| Object Dynamic State Dim. | 12 |
| Number of Objects | 4 |

(b) MLP model settings.

| Parameter | Value |
|---|---|
| Network Size | $3 \times 256$ |
| Activation function | SiLU |
| Ensemble Size | 5 |
| Optimizer | ADAM |
| Batch Size | 256 |
| Epochs | 50 |
| Learning Rate | 0.0001 |
| Weight decay | $5 \cdot 10^{-5}$ |
| Weight Initialization | Truncated Normal |
| Normalize Input | Yes |
| Normalize Output | Yes |
| Predict Delta | Yes |

### B.2.2 OPAX Heuristic Variant

In the case of Fetch Pick & Place with an high-dimensional observation space, we implement a heuristic of OPAX. Note that as Fetch Pick & Place is a deterministic environment without noise, we only model epistemic uncertainty.

---

**OPAX (Heuristic Variant)**

---

**Input:** Ensemble $\{\boldsymbol{f}_i\}_{k=1}^K, \epsilon \ll 1$
**for** $n \in 1, \ldots, N_{\max}$ **do**
Solve optimal control problem till convergence for the system: $\boldsymbol{x}_{t+1} = \boldsymbol{f}_{j_t}(\boldsymbol{x}_t, \boldsymbol{u}_t)$.

$$\boldsymbol{u}_{0:T-1}^\star, j_{0:T-1}^\star = \underset{\boldsymbol{u}_{0:T-1}, j_{0:T-1}}{\mathrm{argmax}} \sum_{t=0}^{T-1} \sum_{i=1}^{d_x} \log \left( \epsilon^2 + \sigma_{n,i}^2(\boldsymbol{x}_t, \boldsymbol{u}_t) \right) \qquad \blacktriangleright \text{Estimate}$$

$\sigma_{n,i}(\boldsymbol{x}_t, \boldsymbol{u}_t)$ with ensemble disagreement.

Rollout $\boldsymbol{u}_{0:T_1}^\star$ on the system and collect data $\mathcal{D}_n$.
Update models $\{\boldsymbol{f}_i\}_{k=1}^K$.

---

### B.2.3 Controller Parameters

The set of default hyperparameters used for the iCEM controller are presented in Table 7, as they are used during the intrinsic phase for OPAX, CEE-US, and other baselines. The controller settings used for the extrinsic phase are given in Table 8. For more information on the hyperparameters and the iCEM algorithm, we refer the reader to Pinneri et al. (2021).

Table 7: Base settings for iCEM as they are used in the intrinsic phase. Same settings are used for all methods.

| Parameter | Value |
|---|---|
| Number of samples $P$ | 128 |
| Horizon $H$ | 30 |
| Size of elite-set $K$ | 10 |
| Colored-noise exponent $\beta$ | 3.5 |
| *CEM-iterations* | 3 |
| Noise strength $\sigma_{\text{init}}$ | 0.5 |
| Momentum $\alpha$ | 0.1 |
| `use_mean_actions` | Yes |
| `shift_elites` | Yes |
| `keep_elites` | Yes |
| Fraction of elites reused $\xi$ | 0.3 |
| Cost along trajectory | `sum` |

Table 8: iCEM hyperparameters used for zero-shot generalization in the extrinsic phase. Any settings not specified here are the same as the general settings given in Table 7.

| Task | Controller Parameters | | | | |
|---|---|---|---|---|---|
| | Horizon $h$ | Colored-noise exponent $\beta$ | `use_mean_actions` | Noise strength $\sigma_{\text{init}}$ | Cost Along Trajectory |
| Pick & Place | 30 | 3.5 | Yes | 0.5 | `best` |
| Throwing | 35 | 2.0 | Yes | 0.5 | `sum` |
| Flipping | 30 | 3.5 | No | 0.5 | `sum` |

# C  Study of exploration intrinsic rewards

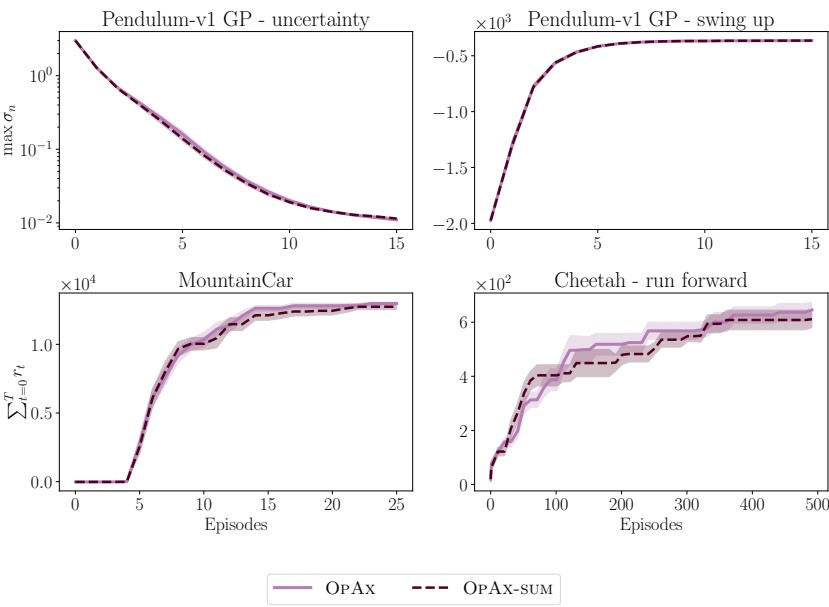

Figure 5: Comparison of OPAX between intrinsic reward proposed in Equation (6) and the sum of epistemic uncertainties proposed by (Pathak et al., 2019), i.e., OPAX-SUM. For both choices of intrinsic rewards, OPAX reduces the epistemic uncertainty and performs well on downstream tasks.

The intrinsic reward suggested in Equation (6) takes the log of the model epistemic uncertainty. Another common choice for the intrinsic reward is the epistemic uncertainty or model disagreement without the log (Pathak et al., 2019; Sekar et al., 2020). In the following Lemma, we show that these rewards are closely related.

**Lemma 14.** *Let* $\sigma_{\max} = \sup_{\boldsymbol{z} \in \mathcal{Z}; i \geq 0; 1 \leq j \leq d_x} \sigma_{i,j}(\boldsymbol{z})$ *and* $\sigma > 0$. *Then for all* $i \geq 0$ *and* $j \in \{1, \ldots, d_x\}$

$$\sigma_{i,j}^2(\boldsymbol{z}) \leq \frac{\sigma_{\max}}{\log(1 + \sigma^{-2}\sigma_{\max})} \log(1 + \sigma^{-2}\sigma_{i,j}^2(\boldsymbol{z})) \leq \frac{\sigma^{-2}\sigma_{\max}}{\log(1 + \sigma^{-2}\sigma_{\max})} \sigma_{i,j}^2(\boldsymbol{z}).$$

*Proof.* Curi et al. (2020) derive the first inequality on the left. The second inequality follows since $\log(1 + x) \leq x$ for all $x \geq 0$. $\qquad\square$

Due to this close relation between the two objectives, our theoretical findings also apply to the intrinsic reward proposed by (Pathak et al., 2019). Moreover, empirically we notice in Figure 5 that OPAX performs similarly when the sum of epistemic uncertainties is used instead of the objective in Equation (6). However, our objective can naturally be extended to the case of heteroscedastic aleatoric noise, since it trades off the ratio of epistemic and aleatoric uncertainty.

