# OpenReview forum: "Optimistic Active Exploration of Dynamical Systems"
_NeurIPS.cc/2023/Conference — NeurIPS 2023 poster_

### Official Review · Reviewer_Y1JX · 2023-06-28

**Soundness:** 3 good
**Presentation:** 3 good
**Contribution:** 2 fair
**Rating:** 5
**Confidence:** 3

**Summary:**

This paper addresses the problem of active exploration of a (Markovian) dynamical system with continuous states and actions. In the absence of any cost function, the proposed objective is the maximization of the one-step information gain on the dynamics model. The paper presents an algorithm, called OpAx, to maximize the introduced objective by alternating optimistic planning with the current estimate of the dynamics model and data collection with the resulting policy to improve the model estimate. Then, the paper provides a convergence analysis of OpAx, which provably converge to the true model asymptotically under various assumptions. Finally, the OpAx algorithm is evaluated against some relevant baselines in continuous control tasks.

**Strengths:**

- The paper tackles an active exploration problem for system identification that is of general interest to both the reinforcement learning and optimal control communities;
- The algorithm presented in the paper is simple and intuitively sound;
- The paper combines theoretical justification with empirical validation in challenging domains;
- The paper is well-presented and looks rigorous in the theoretical statements.

**Weaknesses:**

- The paper is mainly motivated as a tool for system identification that can be used in preparation to solve RL or control tasks, but this motivation looks somewhat weak for the RL side;
- The experimental results are not terrible, but I can hardly see improvements over simpler baselines, such as one using planning with the average model instead of optimism;
- The paper neglects several related works in RL, especially reward-free RL and active model estimation.

**Questions:**

This paper looks like a sound and neat work in system identification through active exploration and optimism. I do not have major concerns about the quality of the paper per se. However, judging from an RL perspective, this contribution looks two/three years late. Several recent works have demonstrated how to learn a model of the transition dynamics that is just good enough for planning by taking a polynomial number of samples from the process, which goes under the name of reward-free exploration. Instead, this paper takes the hardest route of trying to reduce the epistemic uncertainty "uniformly" over the dynamics model, which is arguably far-fetched without strong reachability assumptions.

For this reason, I struggle to see how this paper contributes to the advancement of RL research, as it does not compare well with reward-free exploration in terms of theoretical guarantees, neither significantly improves over previous practical methods empirically.
However, this might be a useful contribution for the control community, and I am open to raise my score if the authors could show me that.

I report below some detailed comments.

**Dynamical Systems and MDPs**

The paper motivates the approach as a tool for learning the dynamics model to then solve RL tasks. However, the standard model for RL is the MDP, while the paper focuses on a specific type of Markovian processes in which the transitions are given by a deterministic function plus a Gaussian noise. Can the authors explain how this model relates with MDPs, and the relative expressive power of the two?

**Well Calibration Assumption**

Whereas Ass. 1 might be standard in the control literature, which I am not familiar with, it is kind of black-box from an RL perspective. It looks like it is saying that the problem is learnable, but I would like to know under which conditions on $f^*$ this is the case.
When is the Ass. 1 violated in practice? Does this implicitly assume strong reachability? Can the authors relate the Ass. 1,2,3 to the common structural assumptions on the MDP model?

**Convergence Results**

The theoretical results focus on showing that the maximum information gain shrinks with $N$, but the rate is actually exponential in $T$. This means that the sample complexity of learning an approximately correct model is also exponential?

**Experiments**

The OpAx algorithm does not seem to improve significantly over the Mean-AE baseline in the considered experiments. While OpAx arguably comes with a stronger theoretical justification, I would like to see some benefit in the experiments as well, or at least a theoretical result showing that planning with the mean model cannot be provably efficient under the considered assumptions.

**Related Works**

The paper neglects a recent yet considerable stream of works in reward-free RL, which seems to have overlapping motivation with their active exploration problem. This stream of works started with (Jin et al., Reward-free exploration for reinforcement learning, 2020) and counts dozens of subsequent results for tabular models, linear models, and general function approximation. Can the authors relates their contribution with reward-free RL literature?

Another important work that seems to be missing is (Tarbouriech et al., Active model estimation in Markov decision processes, 2020), which looks also very close to the problem formulation of this paper, although it focuses on MDPs rather than dynamical systems with Gaussian noise.

**Minor**

- The paper refers to the Appendix A for the proof of Lemma 1, but I cannot find it;
- I would report confidence intervals instead of standard error in the plots.

**Limitations:**

The paper does not explicitly discuss the limitations of this work in terms of empirical results or theoretical guarantees.

---

> ### Author Rebuttal · Authors · 2023-08-07
>
> We thank the reviewer for their feedback and for sharing other relevant work. Based on their summary, we believe that the reviewer misunderstood our objective. In particular, our objective is *not the one step but $T$-step information gain on the model*. This is a crucial difference since it allows us to quantify the information gained over a whole rollout instead of just one transition.
> ## Weaknesses:
> *W1 Motivation from RL side*: There is a plethora of work on unsupervised RL that studies our problem setting [7-12]. Generally, systems such as robots are used to perform multiple tasks that are often not known apriori. Prior works such as [5] motivate the problem in the context of reward-free RL but with rewards coming from a known reward class. This setting, however, needs a finite covering number of the reward class and is not practical for our general problem.
> Traditionally, for a given dynamics model, well-established model-based controllers are commonly used to optimize for any task. Therefore, having a generalist approach to efficient exploration and identification of the dynamics *uniformly* is typically preferred.
>
> ## Additional Questions:
> *Q1 Dynamical Systems and MDPs*: We consider an MDP model with an infinite number of states and actions, unlike in [13, 14]. We use a classical textbook definition of a dynamical system [6], which includes a large class of systems in RL (MUJOCO physics simulator, etc., ).
>
> *Q2 Well Calibrate Assumption*: Well calibration is a standard assumption in model-based RL with continuous states and action spaces. It has been used for both linear and nonlinear systems [1-4].  For the RKHS settings, this assumption is satisfied (cf., Lemma 2). For BNNs, proving well-calibration is an open problem (cf, line 108-112). The well-calibrated assumption has no connection to strong reachability. This assumption intuitively tells that the algorithm is able to learn a mean as well as the confidence region around the mean function.
>
> *Q3 Intuition of Assumptions 2 and 3*: Assumption 3 is a common assumption on the transition noise, which can further be relaxed to the even more general heteroscedastic noise case (cf., line 123).  Finally, Assumption 2, makes smoothness assumptions on the dynamics that are common in nonlinear systems literature [6].
>
> *Q4 Convergence Results*: The rate of convergence depends on $N$ so the sample complexity (as a rate) for kernels such as RBF, linear, etc is of polynomial order (cf.Theorem 2). Under our assumptions, the dependence on the horizon T is exponential, similar to [3]. However, that doesn’t influence the rate. As also argued in [3], to obtain better dependence on the horizon T, we need stronger assumptions (one such assumption is shown in Appendix B where we obtain polynomial dependence on T).
>
> *Q4 Experiments*: The main contribution of our work is theoretical. In [3] the benefits of optimism are discussed in further detail. Here it is shown that optimistic planning performs similarly to mean or PETS planning when the reward landscape is not scarce. Since this is the case for our intrinsic rewards, we empirically observe similar performance. Nonetheless, Mean and PETS, until now, have no theoretical guarantees for the general class of systems we consider. Deriving such guarantees for them is an open research problem.
>
> *Q5 Related Works*: We thank the reviewer for sharing additional related works. We have included them and more works from reward-free RL in the revised paper. In our work, we consider a general class of dynamical systems for which we make very common/practical assumptions.  As we mention in the general comment, theoretical results of our kind, do not exist.
>
> *Q6 Proof of Lemma 1*: Cf., line 516.
>
> Having addressed all of the questions provided by the reviewer, and given the contributions of this paper, we kindly ask the reviewer to increase their evaluated score for our paper. We would be happy to answer any remaining questions or concerns.
>
> ## References:
> [1] Chowdhury, Sayak Ray, and Aditya Gopalan. "On kernelized multi-armed bandits." ICML, 2017.
>
> [2] Yasin Abbasi-Yadkori et al (2011). Regret bounds for the adaptive control of linear quadratic systems. COLT.
>
> [3] Curi, S., et al (2020). "Efficient model-based reinforcement learning through optimistic policy search and planning." NeurIPS.
>
> [4] Abeille, M., et al (2017). "Thompson sampling for linear-quadratic control problems." AISTATS.
>
> [5] Chen, J. et al (2023). On the statistical efficiency of reward-free exploration in non-linear rl. NeurIPS.
>
> [6] Khalil, H. K. (2015). Nonlinear control, volume 406. Pearson New York.
>
> [7] Schmidhuber, J. (1991). A possibility for implementing curiosity and boredom in model-building neural controllers. In Proc. of the international conference on simulation of adaptive behavior: From animals to animats.
>
> [8] Wagenmaker, A. et al (2020). Active learning for identification of linear dynamical
> systems. COLT.
>
> [9] Mania, H. et al (2022). Active learning for nonlinear system identification with guarantees. JMLR.
>
> [10] Pathak et al (2019). Self-supervised exploration via disagreement. ICML.
>
> [11] Sekar, R. et al (2020). Planning to explore via self-supervised world models.  ICML.
>
> [12] Sancaktar, C., et al (2022). Curious exploration via structured world models yields zero-shot object manipulation. NeurIPS.
>
> [13] Jin, C. et al (2020). Reward-free exploration for reinforcement learning. ICML.
>
> [14] Tarbouriech, J. et al (2020). Active model estimation in markov decision processes. UAI.

---

> > ### Author Response · Authors · 2023-08-17
> > **Follow up on rebuttal**
> >
> > We hope we could address your concerns adeptly. We would further like to emphasize how our work differs from reward-free exploration and the practical methods we consider for our evaluation.
> >
> > 1. Our work considers a very general class of dynamical systems in continuous state and action spaces. As we highlight in our rebuttal, our assumptions are common in both control and learning literature. Compared to reward-free RL (referring to the papers we discuss in the general comment and [3]), our assumptions are more practical and general.
> >
> > 2. We propose *a practical algorithm with theoretical guarantees* for this very general setting. We show that our algorithm works on  several real-world systems of varying state and action space dimensions. Compared to other practical methods, our algorithm comes with first-of-its-kind theoretical guarantees. Such guarantees did not exist before for our setting. This is also discussed by other concurrent works ([1, Paragraph 1 in the Introduction] [2, Paragraph 2 in the Related Work]).
> >
> > We have updated the paper to further highlight our contribution. It is much appreciated if you could please reconsider your assessment, or respond with questions/suggestions so that we can improve the paper in this regard.
> >
> > [1] Chakraborty, S. et al (2023). STEERING: Stein Information Directed Exploration for Model-Based Reinforcement Learning. arXiv preprint arXiv:2301.12038.
> >
> > [2] Wagenmaker, A. et al (2023). Optimal Exploration for Model-Based RL in Nonlinear Systems." arXiv preprint arXiv:2306.09210.
> >
> > [3] Chen, J. et al (2023). On the statistical efficiency of reward-free exploration in non-linear rl. NeurIPS.

---

> > ### Comment · Reviewer_Y1JX · 2023-08-17
> > **Follow-up comments**
> >
> > I want to thank the authors for their thorough rebuttal.
> >
> > While I have got clarifications on various aspects, I feel like my main concern on the motivation of the work to be somewhat unresolved. I would be happy to raise my score if the authors can convince me on that.
> >
> > **1. Learning the model**
> > The rebuttal says *"traditionally, for a given dynamics model, well-established model-based controllers are commonly used to optimize for any task. Therefore, having a generalist approach to efficient exploration and identification of the dynamics uniformly is typically preferred."* If the goal is to solve RL tasks with the learned model, I struggle to understand why one should care about reducing the epistemic uncertainty uniformly over the state/action space. I think it is clear that reducing the epistemic uncertainty in a state that cannot be reached with meaningful probability by any policy is less important than reducing the epistemic uncertainty in a state that is easier to reach. Can the authors tell me what is the flaw in this thought? It is great to provide the first *"theoretical results of this kind"*, but I would like to understand why they matters for RL.
> >
> > **2. Reward-free RL**
> > To develop on the previous comment, reward-free RL objective instead focus on learning the model "where it matters". In the global response, the authors are saying that their model assumption is strictly more general than what has been considered in reward-free RL. If this is the case, can they specialize their sample complexity result to have a direct comparison with those works (e.g., in linear MDPs, low-rank MDPs...)? From my understanding, the reward-free literature also considered very general MDP classes, such as in (Qiu et al., On Reward-Free RL with Kernel and Neural Function Approximations: Single-Agent MDP and Markov Game, 2021) and (Chen et al., On the Statistical Efficiency of Reward-Free Exploration in Non-Linear RL, 2022) also mentioned by the authors. How does their model assumption generalize over them?
> >
> > **3. Active model estimation**
> > Can the authors discuss how their objective relates with the one in (Tarbouriech et al., Active model estimation in Markov decision processes, 2020)? Do they consider a generalization of the latter objective for a larger class of MDPs or there are other menanigful differences? (Note that the reviewers cannot see the revised version of the paper).
> >
> > **4. Theoretical ground for heuristic methods** In their rebuttal, the authors are providing a compelling argument for the motivation of this paper as a way to provide theoretical ground for heuristic methods that have been extensively used in practical RL (e.g., curiosity-driven approaches). Can the authors develop on this?. To me, this looks a stronger motivation, but it is also sidelined in the presentation.
> >
> > I am sorry for the late reply, but hopefully there will still be sufficient time to conclude the discussion.

---

> > > ### Author Response · Authors · 2023-08-18
> > > **Response on the comment**
> > >
> > > We thank the reviewer for the discussion.
> > >
> > > **1. Learning the model** 1) The main benefit of learning a general model is that once we have an accurate model of the system, we can use it with any model-based or model-free method to quickly obtain a policy for any new reward/downstream task interacting with the system.  For example, consider a home robot that is asked to solve unknown tasks specified by the user (human). With OpAx the robot can independently explore and learn the environment and quickly solve any tasks when dictated by the user without further exploration. 2) We would also like to emphasize that our algorithm **explores only the state action pairs that are reachable** under the policy class $\Pi$.  Accordingly, our algorithm only reduces uncertainty in the reachability set (cf., Theorem 2) and does not explore unreachable state action pairs. We also make no assumptions about the downstream tasks and their policy classes. Nonetheless, if additional information on the tasks and their policy class is available, we can also restrict the policy class (and exploration) for OpAx to the set of policies (state-action pairs) that are relevant to the downstream tasks.
> > >
> > > **2. Reward-free RL** 1) We assume dynamics of the form $x_{t+1} = f(x_t, u_t) + \epsilon_t$. For the case of linear MDPs or low-rank MDPs, a special structure on the function $f(x_t, u_t)$ is assumed. In the case of linear MDPs ($f(x_t, u_t) = \phi^T(x_t, u_t) \mu(x_t)$), if the unknown embedding $\mu$ satisfies Assumption 4 (RKHS setting), our sample complexity bound remains of the same order with differences arising only in constant factors. 2) Indeed, Qiu et al., 2021 consider nonlinear approximations. However, they make a very different set of assumptions. Particularly, they make continuity assumptions on the value function of all possible rewards (Assumption 3.2). This inherently implies a restricted reward class. Furthermore, the resulting algorithm requires a covering number of the resulting Q-function class (section 3.3). Jin et al., 2022 also take a similar route with the covering number and making assumptions for all possible rewards (cf., Assumption 1, 2 and Definition 1). Our set of assumptions are very different. 1) We make **no assumptions about the rewards and their class**. For our setting, we typically have no knowledge of the underlying downstream tasks and their rewards. 2) Our assumptions are made only on the dynamics model. Dynamical systems such as robots are driven by physical principles for which we typically have a better understanding. Thus, we believe making assumptions only on the dynamics is more natural and is often also done in the learning-based control community [1]. We added this discussion in the revised version.
> > >
> > > **3. Active model estimation** Tarbouriech et al., 2020 design an algorithm that minimizes model estimation error (cf., Section 3.1). They operate in finite state action MDPs. In this setting, the model estimation error is inversely proportional to the state action visitation count (Proposition 2). **We are in a continuous state-action space**. In this setting, there are infinitely many states and actions and we cannot use the state action visitation count. For the continuous state and actions, the equivalent of model error is the model epistemic uncertainty. We use this as our objective to guide the exploration. In addition to the objective, we also plan in continuous spaces under unknown dynamics. We relate the regret of planning to the decay of the model epistemic uncertainty and give the sample complexity bound (proof of Theorem 1). Compared to the finite state action MDP case, this requires a completely different set of tools. In summary, Tarbouriech et al., 2020 consider a discrete state action space and we consider a continuous one.  Therefore, the tools for both are different and neither generalizes the other.
> > >
> > > **4. Theoretical ground for heuristic methods** We thank the reviewer for acknowledging that our argument is compelling. Indeed, a key contribution of our work is using ideas from experiment design and information theory to derive our intrinsic/exploration reward which is the epistemic uncertainty. This reward is commonly used by practical curiosity-driven RL methods and through our derivation we give theoretical grounds for such approaches. In addition to this, we show that combining this objective with optimism gives theoretical guarantees. We are the first to give such a result. Following the reviews, we updated the paper to emphasize more on this aspect of our contribution.
> > >
> > > We hope our response addresses the reviewer's concerns and are happy to answer further questions.
> > >
> > > [1] Hewing, Lukas, et al. “Learning-based model predictive control: Toward safe learning in control.” Annual Review of Control, Robotics, and Autonomous Systems 3 (2020).

---

> > > > ### Comment · Reviewer_Y1JX · 2023-08-18
> > > > **Re: Response on the comment**
> > > >
> > > > I thank the authors for the quick reply. Let me rephrase some of my previous comments, as I guess they were not clear enough. I kindly ask the authors to answer those questions pointedly if they can.
> > > >
> > > > **1. Learning the model** The benefit of having an exact model of the transitions was clear already. However, if the goal is to solve RL tasks with the learned model, we do not really need to model the transitions everywhere, but just where it is useful to compute the optimal policy. The latter is true also when the reward function is unknown while learning the model, and this is the crucial insight from reward-free RL literature. Does the proposed exploration method include a way to weight the epistemic uncertainty in a state/action pair for the "importance" of the state/action pair in the computation of the optimal policy (for any task)?
> > > >
> > > > **2. Reward-free RL** I think the authors are saying that those reward-free results (e.g., linear MDPs) are special cases of their. Thus, it should be possible to compare apples to apples. Can they compare their sample complexity rate with those from reward-free exploration?
> > > >
> > > > **3. Active model estimation** So the short answer is "Yes", the proposed objective is a version of the active model estimation objective for continuous MDPs.
> > > >
> > > > **4. Theoretical ground for heuristic methods** I currently believe this should be made the central motivation for the work rather than a by-product.

---

> > > > > ### Author Response · Authors · 2023-08-18
> > > > > **Response to the reviewer's comments**
> > > > >
> > > > > We would also like to thank the reviewer for their active engagement in the review process. Prior to our response, we would like to highlight a key difference between reward-free RL methods and ours. The reward-free RL methods we have discussed this far make assumptions on the reward structure, (cf, our previous comment for instance where assumptions on the value functions are made, or linear MDPs). In our setting, we only make assumptions about dynamics and none about the rewards. We acknowledge that if more knowledge about the reward’s structure is available this can be leveraged for better guided exploration. However, this is not the problem setting we investigate.
> > > > >
> > > > > **1. Learning the model**: In our setting, because we have no assumptions about the reward function and its structure, we cannot determine the regions that are useful to compute the optimal policy (think of a downstream task that requires the system to go to any arbitrary reachable state). This is why our method explores the whole reachable state-action space. As the reviewer indicated, one could in principle weight our intrinsic reward with additional “importance” weights. However, we believe incorporating additional knowledge on the reward structure, and thus knowledge of the optimal policy, into our method would result in a different algorithm that tackles a different problem setting.
> > > > >
> > > > > **2. Reward-free RL**: We apologize if there has been a misunderstanding. We do not imply that our result is a general case of the linear MDP results for reward-free exploration. We believe these works tackle a different problem i.e., we are not comparing apples and apples. With our response, we meant that the transition dynamics considered in the linear-MDP case are a special case of the dynamics model we study. However, our algorithm is designed for the case where the reward can have any structure. On the contrary, the reward has a linear structure for the linear-MDP case. Similarly, other works in reward-free RL that we have discussed make some assumptions about the reward. If this knowledge of the reward is available, these algorithms might be preferred. However, since we assume no such knowledge, we make a completely different set of assumptions and design an algorithm for a different setting. Therefore, we cannot compare the sample complexity of the two methods since 1) they are designed for different settings and investigate different quantifies, and 2) they make a different set of assumptions.
> > > > >
> > > > > **3. Active model estimation**: Yes, we believe this is one plausible interpretation of our result.
> > > > >
> > > > > **4. Theoretical ground for heuristic methods**: Thank you for the feedback. As we highlighted in the previous comment, we have indeed adapted the paper and in the new version, we emphasize this aspect of our contribution very clearly.
> > > > >
> > > > > We hope we could address the reviewer’s questions.

---

> > > > > > ### Comment · Reviewer_Y1JX · 2023-08-18
> > > > > > **Re: Response to the reviewer's comments**
> > > > > >
> > > > > > Thank you for the further reply, we are hopefully clarifying each other opinion. Let me provide some additional thoughts.
> > > > > >
> > > > > > *"In our setting, because we have no assumptions about the reward function and its structure, we cannot determine the regions that are useful to compute the optimal policy"*
> > > > > >
> > > > > > I disagree with this statement, which I think is the essence of the problem. The structure of the transition model alone still gives an indication on which are the regions that are useful to compute optimal policies. For instance, let us think about an MDP having a state that is reachable with some probability $\epsilon > 0$. Even for a reward function that places a reward of $1$ in this state and $0$ everywhere else, it is clear that any policy is $\epsilon$-optimal for this task. This means that I can forget about reducing the epistemic uncertainty on the transitions that lead the agent to that state, because it is irrelevant for **every** reward function. This is exactly the premise of reward-free exploration.
> > > > > >
> > > > > > *"The reward-free RL methods we have discussed this far make assumptions on the reward structure. In our setting, we only make assumptions about dynamics and none about the rewards."*
> > > > > >
> > > > > > It is true that linear MDP assumption affects also the structure of the reward, but in general reward-free RL does not make any assumption on the reward beyond boundedness (see the original paper by Jin et al. in tabular settings). I guess boundedness would be needed in your formulation as well, otherwise I cannot see how any zero-shot guarantee can be provided.
> > > > > >
> > > > > > *"Therefore, we cannot compare the sample complexity of the two methods since 1) they are designed for different settings and investigate different quantifies, and 2) they make a different set of assumptions."*
> > > > > >
> > > > > > On this one I can agree. My concern is that reward-free exploration is the "right" objective if one cares about learning transitions for zero-shot generalization, and that the motivation for your active exploration objective shall be found elsewhere.

---

> > > > > > > ### Author Response · Authors · 2023-08-18
> > > > > > > **Follow up on the comment**
> > > > > > >
> > > > > > > Indeed, we believe we disagree on the approach. Here is our response below.
> > > > > > >
> > > > > > > 1. **On your example**: First we would like to clarify that in continuous state-action spaces, the probability of exactly reaching any state is 0. Thus, from hereon on we can talk about a desired region of the state-action space (e.g., a ball around the desired state). Assume that the reward at this region which is reachable with probability $\epsilon$,  is $1/\epsilon$  (or any scaling) and 0 everywhere else. If the model we learn cannot accurately drive us to this region, the regret will be $\mathcal{O}(1)$.
> > > > > > > In general, we construct our method for any possible reward (including the worse case).
> > > > > > > Our algorithm also trades off the epistemic uncertainty and the aleatoric uncertainty. The aleatoric uncertainty corresponds to the inherent uncertainty of the true dynamical system. Our reward is proportional to $\sigma_n(x_t, u_t) / \sigma $, where $\sigma$ is the aleatoric uncertainty. As we highlight in lines 123-124, the aleatoric uncertainty can be heteroscedastic, i.e., $\sigma$ depends on the $(x_t, u_t)$ without loss of generality. Therefore, our algorithm naturally prefers visiting states that we can plausibly reach under some policy with true dynamics. Nonetheless, if you run our algorithm for infinitely long, eventually the uncertainty will reduce in the whole space.
> > > > > > >
> > > > > > > 2. **Structural assumptions on MDP**: Again we would like to highlight that we are in the setting of **continuous state-action spaces**. This is different from the finite setting. Hence, we would appreciate focusing the discussion on this setting. In this setting, the works we have discussed thus far (including the ones shared by the reviewer), make structural assumptions on the reward (or its corresponding value) functions. We believe this is itself an essential argument for our approach since the works we have referred to thus far require a specific class of rewards and we do not.
> > > > > > >
> > > > > > > We believe our approach is motivated from a practical standpoint and therefore also generally pursued in the control/system identification, and active/curious exploration community. Guarantees of our sort do not exist for active exploration. On top, it is a practical method as we also show in our experiments. Even though we don't have the same opinion on what is the best approach for zero-shot generalization under no structural assumptions on the reward class, we solve an open problem that we think is an important contribution to the community.

---

> > > > > > > > ### Comment · Reviewer_Y1JX · 2023-08-18
> > > > > > > > **Re: Follow up on the comment**
> > > > > > > >
> > > > > > > > Before following up below, I want to emphasize that my comments are only driven by the pursuit of my best evaluation of this paper. I hope the authors will not be offended if I share my view bluntly.
> > > > > > > >
> > > > > > > > **1. Example** I think you mean sub-optimality, as there is no regret in zero-shot RL. In that case, this is why you need boundedness of the reward function. I am very skeptical you can derive any zero-shot guarantee without boundedness, otherwise there will always be a reward function that gives infinite reward to a state in which you are making some error in the transition model estimation, and the sub-optimality will be infinite. With boundedness, I think my example stands (as corrected by you for continuous settings).
> > > > > > > >
> > > > > > > > **2. Structural assumptions** I get the difference between finite and continuous state-action spaces, but I think it is beside the point. Reward-free RL papers make structural assumption on the transitions that are reflected by the reward function, and those are arguably essential to provide zero-shot guarantees. Can you provide zero-shot guarantees without any structural assumption on the rewards or transitions?
> > > > > > > >
> > > > > > > > **Even though we don't have the same opinion on what is the best approach for zero-shot generalization...** Unfortunately, this one does not look like a matter of opinion. Can you provide a zero-shot guarantee that outperforms reward-free exploration in any setting (with assumptions of your choice)?
> > > > > > > >
> > > > > > > > **We believe our approach is motivated from a practical standpoint** This is a very good point. Indeed, reward-free framework did not lead to any practical algorithm. This is arguably the right dimension to motivate the work.

---

> > > > > > > > > ### Author Response · Authors · 2023-08-18
> > > > > > > > > **Response to the reviewer**
> > > > > > > > >
> > > > > > > > > We thank the reviewer for their feedback and appreciate their effort in evaluating our work.
> > > > > > > > > When talking about regret, we mean simple regret, which is a different term for the sub-optimality gap.
> > > > > > > > >
> > > > > > > > > *1. Structural assumptions and 2. Provide a zero-shot guarantee*: The works from Jin et al., 2022 and Qiu et al., 2021 make assumptions on the value function, which depends both on the transition dynamics and the reward. Furthermore, we would like to clarify that *$\epsilon$ in the example above corresponds to the probability of reaching a particular region $A$ under the true dynamics*. All subsets $A$ of the reachability set $R$, have a probability $p(A) >0$. Therefore, for any subset $A$ of $R$, the reward $r= 1_{(x, u) \in A} * 1/p(A)$ is bounded. Not being able to drive the system to $A$ can suffer $O(1)$ sub-optimality gap in this case. This is an instance of reward which may require exploration in the full reachability set as we do.
> > > > > > > > > Generally, we agree that providing sample complexity bound on the zero-shot performance for any reward is not possible unless more structural assumptions on the reward are made. This is one of the reasons why we focus on learning the dynamics and giving **guarantees only on the model error**, i.e., we do not provide and neither claim to give zero-shot guarantees in the main paper. The intuition here is that once we have learned the true dynamics model, we can use it to *control any task*. Giving guarantees on the zero-shot performance will require additional assumptions which we believe do not fit our setting. For example, already for a very common text-book case of linear systems with quadratic cost (LQR), the reward/costs are not bounded. In our experiments, we consider the zero-shot performance because this is also extensively done by prior work in curious exploration. However, we also perform an experiment that directly studies our theoretical guarantees (cf., Figure 1). Lastly, if we were to make assumptions on our reward function such as Lipschitzness continuity, we can derive zero-shot guarantees. The analysis is similar to H-UCRL ([1], Lemma 5), where we can replace the right-hand side of the inequality with our model error. We will add it to the Appendix. However, this is a special case, and our method is motivated from the perspective where we have no knowledge/assumptions about the reward.
> > > > > > > > >
> > > > > > > > > *2. Right dimension to motivate the work*: We agree that this is a key advantage of our work, and we also acknowledge that in the related work of the revised paper, where we also list the works on reward-free RL discussed here.
> > > > > > > > >
> > > > > > > > > In summary, 1) the reviewer has correctly mentioned that **structural assumptions on the reward and transition** are required to give zero-shot guarantees. 2) Because we do not make any such assumptions, we believe the best approach is to learn the dynamics model globally first and then use it for control. This is why we (and many other works) study the problem of system identification.
> > > > > > > > >
> > > > > > > > > We hope this clarifies the reviewer's concerns. Based on the reviewer's last comment,  we believe the reviewer has acknowledged our contribution to the field of practical active exploration and system ID. Accordingly, we would appreciate it if they will reevaluate our work.
> > > > > > > > >
> > > > > > > > > [1] Curi, Sebastian, et al., "Efficient model-based reinforcement learning through optimistic policy search and planning." Advances in Neural Information Processing Systems 33 (2020).

---

> > > > > > > > > > ### Comment · Reviewer_Y1JX · 2023-08-19
> > > > > > > > > > **Re: Response to the reviewer**
> > > > > > > > > >
> > > > > > > > > > First, I want to thank the authors for their effort in finding some common ground between our divergent opinions.
> > > > > > > > > >
> > > > > > > > > > To be honest, I think their argument on the provided example is becoming rather sophistic, but I can agree on the implications *"providing sample complexity bound on the zero-shot performance for any reward is not possible unless more structural assumptions on the reward are made"* and *"giving guarantees on the zero-shot performance will require additional assumptions which we believe do not fit our setting"*. Indeed, I believe reward-free exploration is the right framework to provide those guarantees, while this paper is doing something else.
> > > > > > > > > >
> > > > > > > > > > That said, to acknowledge the merits of the paper in terms of providing theoretical ground for curiosity-driven approaches, I will drop my strict reject recommendation in favour of a borderline evaluation. However, to clear the bar for acceptance I think the paper should
> > > > > > > > > > - Revise the abstract and introduction to better highlight the motivation for this work other than zero-shot guarantees, e.g., avoiding phrases like "How should we explore an unknown dynamical system such that the estimated model allows us to solve multiple downstream tasks in a zero-shot manner?"
> > > > > > > > > > - Provide a thorough discussion on the differences between reward-free literature and this works, and why they both matters, as well as a more direct connection with heuristic methods in active model estimation.
> > > > > > > > > >
> > > > > > > > > > I will leave to the judgment of the AC to assess whether those changes can be carried out for a camera-ready version, or they require a new submission and evaluation.
> > > > > > > > > >
> > > > > > > > > > Best wishes,
> > > > > > > > > > Reviewer Y1JX

---

> > > > > > > > > > > ### Author Response · Authors · 2023-08-21
> > > > > > > > > > > **Response to the reviewer 2**
> > > > > > > > > > >
> > > > > > > > > > > We would also like to thank the reviewer for their active engagement in the rebuttal period and for their feedback on our work.
> > > > > > > > > > >
> > > > > > > > > > > We have adapted the revised paper to emphasize more on our objective of active exploration. For this, we have also introduced a clear problem statement in the paper, which formally states our objective of reducing the model epistemic uncertainty/error globally in the reachability set. Furthermore, we have revised the related works to add a discussion on reward-free RL where we highlight the importance of this line of work and emphasize the difference between the setting we consider. This is in line with the discussion we have had with the reviewer during the rebuttal period.
> > > > > > > > > > >
> > > > > > > > > > > Best wishes, Authors

---

### Official Review · Reviewer_EYiX · 2023-07-06

**Soundness:** 3 good
**Presentation:** 4 excellent
**Contribution:** 2 fair
**Rating:** 5
**Confidence:** 2

**Summary:**

The paper presents an active exploration method (OPAX) for dynamics model learning. The method seeks to maximize information gain, while being optimistic about unknown dynamics with respect to the achievable information gain. Theoretical results establish a connection between information gain and model complexity, and a convergence guarantee for GP models. Experimental results compare OPAX to baselines in several control and manipulation domains.

**Strengths:**

To my knowledge, the paper's principled approach to derive the planning objective in eqs. (6) and (7) from the optimal design perspective is novel. The paper is carefully written and easy to follow. The experiments are relevant and the selection of baselines seems fair. I believe that improving the data efficiency of learning dynamics models for zero-shot task generalization is a relevant research objective.

**Weaknesses:**

The approach does not really outperform the relatively the baseline methods used in the experiments. I don't think that's a huge problem, as I believe the main contribution of the paper is its theoretical part. Still, I think the paper could be improved by explaining better why the optimism does not lead to a significant difference in model performance (OPAX vs PETS-AE and Mean-AE).

**Questions:**

- Can you identify experimental regimes in which the optimism has a significant impact on the performance of the resulting model? Even if such an experiment might be a little cherry-picked, I think it would still be useful for the reader.

**Limitations:**

There is no detailed discussion of limitations anywhere in the paper. I think adding this would improve the paper.

---

> ### Author Rebuttal · Authors · 2023-08-07
>
> We thank the reviewer for the feedback and acknowledge that the reviewer correctly highlighted the strengths of our work. Below we have addressed the reviewer’s concerns.
>
> ## Weaknesses:
> *W1 Outperforming baselines*: We are happy that the reviewer recognized that the main contributions of the paper are theoretical. In [1], the benefits of optimism are discussed in more detail. Particularly, they show that optimism helps in settings with scarce rewards, whereas for non-scarce rewards PETS and optimistic planning perform on par. Since our exploration objective by definition is not scarce (i.e. the reward is the uncertainty which is mostly non-zero everywhere), we observe that OPAX performs similarly to the baselines. We have added this discussion to the revised paper.
>
> ## Additional Questions:
> *Q1 Experimental regimes where optimism helps*:  As highlighted in our response to W1, optimism has an impact on settings with scarce rewards. Our proposed objective is not scarce by definition. However, we could penalize large actions in our objective, i.e., augment our reward to:
> $$
> r(s_t, a_t) = \log\left(1 + \frac{\sigma_n^2(s_t, a_t)}{\sigma^2}\right) - \lambda ||a_t||^2.
> $$
> In this setting, scarcity in the rewards can be induced by picking large values for $\lambda$. Practically,  we penalize/avoid exerting large actions on the system. We compare OpAx to PETS-AE on the pendulum environment for a specific choice of $\lambda=5$ to give the reviewer an intuition, cf., Figure 1 in the attached document. In the figure, it is visible that in this setting, optimism helps in solving the problem faster. We have also added this simple experiment to the appendix of the revised paper.
>
> ## Reference
> [1] Curi, S., Berkenkamp, F., and Krause, A. (2020). Efficient model-based reinforcement learning

---

### Official Review · Reviewer_zJUZ · 2023-07-07

**Soundness:** 3 good
**Presentation:** 3 good
**Contribution:** 3 good
**Rating:** 5
**Confidence:** 3

**Summary:**

The paper presents some insights into active exploration for model-based reinforcement learning. For certain kinds of environments, the authors show a convergence guarantee for model uncertainty. They augment their analysis with an empirical study of an agent using their approach OpAX.

**Strengths:**

The authors point out an interesting research direction. They motivate their approach well and built up almost all of their mathematical framework. The empirical study manages to give a broad picture of the approach's performance.

**Weaknesses:**

The empirical part of this paper is connected only weakly to the theoretical part. Where are the proven convergence properties in the study? Picking domains where edge cases of the convergence properties might be observed, would help the paper. The authors state that the approach they show in the empirical study is very similar to other approaches found in literature, but they show no direct comparison. Why is the empirical study part of this paper, then?

The authors' interpretation of the results is somewhat lavish. The performance of the main approach is very similar to other shown approaches and clear advantage is not shown. The comparison on downstream tasks is important to stress the overall benefit of intrinsic reward, but also to be expected. It is not discussed how the competitiveness or even advantage of the said approaches arises for the "high-dimensional task" (Fig. 4). Is there no price to pay for the more involved approach? Why are the baselines shown without training times? At some point, the authors argue that their approach is not limited to domains where simple assumption of physicality might help. But why are all tested domains strictly physical, then?

Errors:
- throughout the paper: "c.f." --> "cf."
- line 65: "since" --> "when"
- line 133: "since" --> "since," (add comma)
- lines 168ff switch to $f^\star$  from $f^*$
- line 203: "Theorem 1;" --> "Theorem 1:" (use colon)
- line 212: "RKHS, however"--> "RHKS; however," (use semicolon and comma)
- line 222f: "deep mind" --> "DeepMind"

**Questions:**

(also see "weaknesses")

Why does the presented approach work as well as it does? Are there no trade-offs involved?

What does the theoretical argument say about other types of intrinsic reward?

Where is the novelty in the empirical study?

What experiments would be necessary to show the borderline cases of the convergence property?

**Limitations:**

The authors discuss the approach's limitations quite well.

---

> ### Author Rebuttal · Authors · 2023-08-07
>
> We thank the reviewer for the feedback and for pointing out our typos. We have addressed them in the paper.
>
> ### Weaknesses
>
> *W1 Empirical evaluation of theoretical results*: We designed the pendulum experiment (line 242) on GPs, i.e., RKHS setting, precisely to evaluate our theoretical findings empirically. In our experiments, we show in the example of the pendulum environment how fast our algorithm reduces the model epistemic uncertainty in the full state space of the pendulum. Moreover, in this experiment, we consider both GP models with an RBF kernel and BNNs using probabilistic ensembles. In Figure 1., we show that our algorithm monotonously decreases the epistemic uncertainty with the number of episodes N for the GP case, i.e., validating the findings of theorem 2. Furthermore, for the BNN case, we also show the reduction in the epistemic uncertainty which corresponds to an empirical study of theorem 1.
>
> *W2 Similarity of baselines*: We compare with a similar baseline which is [1]. Our intrinsic reward, the sum of log epistemic uncertainty, is similar to the one used by [1, 2] (they use the sum of epistemic uncertainty instead of the log). However, [1, 2] do not use the optimistic planner. Moreover, the two most commonly used planners in RL are; 1) mean (using the mean model) and 2) PETS [3]. We compare our algorithm to these for the same intrinsic reward (baselines: MEAN-AE, PETS-AE), where we show that our method performs on par with these baselines while also providing theoretical guarantees.
> Furthermore, in Appendix D, we also compare the exact intrinsic reward from [1, 2], i.e., the sum of epistemic uncertainty without the log to ours. Here, we demonstrate that both choices of intrinsic rewards perform similarly. Additionally, we show that the intrinsic reward from [1, 2] is also theoretically sound (Lemma 11, Appendix D) when combined with optimistic planning from OpAx.
>
> *W3 Empirical performance of OpAx and baselines*: The strength of our work lies in the theoretical guarantees. In the empirical results, we show that our algorithm performs at least on par with our baselines that do not yield such theoretical guarantees.
>
> *W4 High-dimensional task study*: For the high-dimensional task, the state space is considerably larger (58D), which makes optimizing over hallucinated controls $\eta$ challenging (dimension of $\eta$ is equal to the dimension of the state). In Appendix C.2.2, we propose a heuristic variant for OpAx, which does not scale with the state space size, and therefore can handle high-dimensional systems. Our results indicate that the heuristic variant also performs well. We have updated the main paper to clarify this further.
>
> *W5 Training Budget of OpAx vs Baselines*: All algorithms are given equal compute budget (cf., Appendix C). Our approach requires policy optimization over a larger domain space that includes the actions and the hallucinated actions $\eta$. However, as we list in Appendix C, for all our baselines we use the same hyperparameters (training, optimization steps, optimizer samples, etc.) for fairness. Therefore, all methods had equal training budgets.
>
> *W6 Limitation to domains where the simple assumption of physicality hold and evaluation*: We would appreciate it if the reviewer refers us to the line where the argument is made. Then we can address the comment in detail. We study general dynamical systems, which we also consider in our evaluation.
>
> ## Additional Questions:
> *Q1 Trade-offs for OpAx*: The main practical trade-off of the algorithm lies in the optimization problem, which is now performed over a larger space that includes the hallucinated controls. In theory, we assume an oracle that solves the problem, in practice this is challenging, for instance, due to compute limitations. To this end, we propose a heuristic variant of our approach in Appendix C.2.2 and show that it works for the high-dimensional setting.
>
> *Q2 Other intrinsic rewards*: Essential to our algorithm and theoretical analysis is the choice of our intrinsic reward. Could the reviewer clarify this question? Is the reviewer asking about theoretical guarantees of another algorithm with another objective?
>
> *Q3 Borderline case of convergence*: Could the reviewer clarify what is meant by a borderline case? Our results give a bound on the model epistemic uncertainty in the worse case (including the borderline case). For kernels such as RBF, this implies convergence for any function in its RKHS.
>
> Having addressed all of the questions provided by the reviewer, and given the contributions of this paper, we kindly ask the reviewer to reconsider the assessment for our paper. We would be happy to answer any remaining questions or concerns.
>
> ## References
> [1] Pathak et al (2019). Self-supervised exploration via disagreement. In International conference on machine learning.
>
> [2] Sekar, R. et al (2020). Planning to explore via self-supervised world models.  In International Conference on Machine Learning.
>
> [3] Chua, K., et al. (2018) Deep reinforcement learning in a handful of trials using probabilistic dynamics models. Advances in neural information processing systems.

---

### Official Review · Reviewer_V3Yd · 2023-07-07

**Soundness:** 3 good
**Presentation:** 3 good
**Contribution:** 2 fair
**Rating:** 6
**Confidence:** 4

**Summary:**

This paper studies provable exploration in model-based reinforcement learning and proposes an algorithm with optimistic active exploration based on information gain. Theoretical results and experimental results are provided to support their method.

**Strengths:**

1. The paper presents a practical algorithmic implementation of active exploration with optimism and information gain, by incorporating the techniques from Curi et al. [1] to introduce a hallucinate policy.
2. The authors provide thorough theoretical and experimental analysis.
3. The paper is well written and easy to follow.

[1] Sebastian Curi et al. Efficient Model-Based Reinforcement Learning through Optimistic Policy Search and Planning.

**Weaknesses:**

1. My biggest concern is the novelty of this paper. Provable exploration based on optimism (mutual information or uncertainty) is not novel. Although the proposed algorithm provides an efficient way for practical implementation, which is good, the novelty is still limited compared to H-UCRL, which is also an optimism-based MBRL algorithm with the same hallucinate policy technique.
2. The authors claim that their method can achieve better zero-shot performance compared to baselines. But it is not clear why is this. Can the authors explain in more detail? The current theory does not indicate this result.
3. What makes the algorithm better in terms of generalization ability compared to H-UCRL? They seem to be developed from very similar optimism perspectives with the same hallucinate technique.
4. From the experimental results, it seems that H-UCRL outperforms the proposed algorithm in most training tasks, despite that the proposed algorithm generalizes better (again, more explanation needed). Can the authors comment on this?
5. Can the authors provide the training curves of H-UCRL and CEE-US instead of asymptotic performance?

**Questions:**

See the weakness section above.

**Limitations:**

More discussions on related works and discussions on the zero-shot generalizability are needed to better characterize the contribution of this paper.

---

> ### Author Rebuttal · Authors · 2023-08-07
>
> We thank the reviewer for the feedback. Below we address the reviewer’s concerns.
>
> ### Weaknesses
>
> *W1 Novelty of our work*: We refer the reviewer to the author rebuttal section. If the reviewer is still concerned with the novelty of our work, we’d be happy to receive more detailed feedback on this concern. Additionally, a key component of our algorithm is active exploration which H-UCRL does not perform, i.e., our algorithm takes a generalist/uniform approach towards exploration of dynamical systems, whereas H-UCRL takes a specialist one (see W3).
>
> *W2 Baselines*: We consider four baselines PETS-AE, MEAN-AE, random, and H-UCRL. Random uniformly samples actions from the action space and does not use any intrinsic rewards to guide exploration. Accordingly, it underperforms. PETS-AE and MEAN-AE perform well in all environments. They use our proposed intrinsic rewards but use a greedy (not optimistic) planning approach that has no theoretical guarantees, except for linear systems [1]. Our method comes with strong theoretical guarantees, while empirically performing on par with SOTA baselines.
>
> *W3 H-UCRL Baseline and zero-shot performance*: H-UCRL is a task-specific model-based RL algorithm therefore its exploration is guided for the specific task it is trained for. Accordingly, its learned model does not generalize to novel unseen tasks. This is particularly observable in Figure 2, where we show that H-UCRL does not perform well on new tasks.
> OpAx is a task-agnostic algorithm that performs undirected exploration. Therefore, OpAx is a generalist. Accordingly, OpAx achieves on-par performance with H-UCRL on the tasks H-UCRL is trained for, however, it outperforms H-UCRL on novel unseen tasks.
>
> *W4 Training curves for H-UCRL and CEE-US*: We provide the training curves for H-UCRL for the swimmer environment in order to explain this trade-off between the specialists H-UCRL and generalists OpAx (cf., Figure 2 in the attached document). From the figure, it is noticeable that H-UCRL achieves higher rewards for the tasks it is trained for faster but fails to solve unseen/novel downstream tasks.
> We have added these plots in the appendix of the revised version of the paper as well for further clarification. We also provide training curves for CEE-US (cf., Figure 3).
>
> Thank you for your valuable feedback on our paper. We hope we addressed your concerns and if they're resolved, we kindly request you to consider revising our score upwards. We are happy to provide further clarification.
>
> ## Reference
> [1] Simchowitz, Max, and Dylan Foster. "Naive exploration is optimal for online lqr." International Conference on Machine Learning. PMLR, 2020.

---

> > ### Comment · Reviewer_V3Yd · 2023-08-17
> >
> > Thank the authors for the response. My concern regarding W4 is addressed. However, I'm still confused why the proposed method has better zero-shot generalization ability. Can the authors provide insights beyond experimental results? I didn't seem to find the corresponding theoretical justifications. Besides, can the authors explain in more detail why OpAx performs "undirected exploration"? And the pseudocode also does not indicate the multi-task feature that the authors claim.

---

> > > ### Author Response · Authors · 2023-08-17
> > > **Response on comment**
> > >
> > > We are happy to provide more insights.
> > > We are operating in a setting where the dynamics of the system are unknown.
> > >
> > > *Standard model-based RL approaches such as H-UCRL*: Given a model estimate, standard model-based RL (MBRL) approaches, optimize for a fixed control task with a known reward function to obtain a policy. They execute this policy on the true system to gather data and update the learned model with the collected data. This process is repeated till satisfactory performance on the control task is achieved. Accordingly, they tend to learn the model well only in regions of the state action space where the rewards for the control task are high instead of learning a globally accurate model.
> > >
> > > *OpAx*: For OpAx we consider the problem of active exploration, where the goal is to explore the dynamics globally. To this end, we use the model epistemic uncertainty as our reward (cf., Eq 6). The model epistemic uncertainty is high in regions where we have less data. Accordingly, at each episode, we plan a policy that drives the system to regions with high model epistemic uncertainty. We collect data in these regions and update our model. Since we only consider the epistemic uncertainty as a reward and not a specific control task (as for standard MBRL methods), our algorithm explores the dynamics globally. This is the objective of active exploration. In Theorem 1 and 2, we theoretically justify our algorithm and show that our method provably explores the whole state-action space. To the best of our knowledge, we are the first to give such guarantees.
> > >
> > > When standard MBRL approaches are evaluated on the control task they see during training, they generally achieve better performance faster. This is because they focus on learning a model in regions where the rewards for the task are high. However, since these approaches learn in regions specific to the control-task, they do not perform well on new unseen tasks. In particular, when the unseen tasks have high rewards in different regions of state action space. On the contrary, OpAx explores the whole domain and therefore it performs better on new tasks, i.e., zero-shot generalization.
> > >
> > > It is much appreciated if you could please reconsider your assessment, or respond with questions/suggestions so that we can improve the paper.

---

> > > > ### Comment · Reviewer_V3Yd · 2023-08-18
> > > >
> > > > Thank the authors for the response. This address most of my previous problems. I would consider raise the score if the authors can address my additional concerns:
> > > >
> > > > 1. I strongly recommend the authors to make a more formal statement about their problem. Specifically, there is no need to learn a globally accurate model in standard RL settings that aims to achieve high returns in single tasks, and a model that is locally accurate in high-reward regions suffices. There is no enough incentive to explore the low-reward regions according to the upper confidence bound. Therefore, it is not clear if the property provided in Theorem 2 is desired. This result only makes sense if the objective is to learn policies that are generalizable to new reward functions (but same dynamics), instead of traditional single-task RL objectives.
> > > > 2. The zero-shot generalizability should be the main focus of this paper, which however, does not appear in the problem setting, analysis, and pseudocode. The authors can make one step further by formally stating the generalization objective and the results to make the paper much better.
> > > > 3. The claim that this paper solves multiple downstream tasks in a zero-shot manner is not precise. I would recommend the authors to state it as MDPs with the same dynamics but different reward functions, to distinguish from the meta-RL and multi-task RL papers. Again, it would make the paper more clear if more formal statement and analysis is provided.

---

> > > > > ### Author Response · Authors · 2023-08-18
> > > > > **Response to reviewer's comment**
> > > > >
> > > > > We thank the reviewer for their active engagement and are happy to hear that the reviewer is willing to increase their score.
> > > > >
> > > > > Following the reviewer's comments, we have made the following changes:
> > > > >
> > > > > 1. In the updated version, we adapted the introduction and problem setting section to clearly state that we are in a setting where we do not know the reward of the underlying control task, and therefore would like to learn the dynamics globally.
> > > > > 2. We define a problem statement in the problem setting section, where we formally motivate our objective which is to learn the dynamics globally, i.e., similar to the statement in Theorem 2. This is followed by the motivation discussed above.
> > > > > 3. For the sake of space restrictions, we now theoretically analyze what our global sample complexity bound implies about zero-shot generalization wrt to some reward function in the appendix and have also adapted the pseudocode to include the zero-shot generalization task.
> > > > > 4. In the experiments, we again clarify that we consider systems with the same dynamics but different reward functions.
> > > > >
> > > > > We hope this addresses the concerns of our reviewer.

---

> > > > > > ### Comment · Reviewer_V3Yd · 2023-08-18
> > > > > >
> > > > > > Can the authors elaborate the specific changes/formulas?

---

> > > > > > > ### Author Response · Authors · 2023-08-21
> > > > > > > **Follow up reviewer V3Yd's comments**
> > > > > > >
> > > > > > > We thank the reviewer for the discussion and below we answer the question.
> > > > > > >
> > > > > > > ## Can the authors elaborate the specific changes/formulas?
> > > > > > >
> > > > > > > **Differentiating our setting from traditional model-based RL**:
> > > > > > >
> > > > > > > We adapted the first paragraph of the introduction to: “Model-based RL algorithms, such as (Chua et al. (2018); Curi et al. (2020); Kakade et al. (2020)), excel in efficiently exploring the dynamical system as they direct the exploration in regions with high rewards. However, due to the directional bias, their underlying learned dynamics model fails to generalize in other areas of the state-action space. While this is sufficient if only one control task is considered, it does not scale to the setting where the system is used to perform several tasks, i.e., under the same dynamics optimized for different reward functions. In this work, we consider the setting where we have no knowledge about the underlying reward and its structure, and have to potentially solve multiple control problems for the same system but with different rewards. In this setting, methods to actively explore and learn the dynamical system are preferred (Shyam et al. (2019); Pathak et al. (2019); Sekar et al. (2020); Sancaktar et al. (2022)). Particularly, once an accurate dynamics model is learned, it can be used to optimize for any reward function using traditional control approaches such as trajectory optimization (Biagiotti and Melchiorri, 2008) and model-predictive control (García et al., 1989) in a zero-shot manner. This raises the question: *how should we interact with the system to efficiently learn its dynamics globally?*
> > > > > > >
> > > > > > > **Problem statement**:
> > > > > > > We study the problem of actively exploring unknown dynamical systems. To this end, we propose an algorithm $\textit{Alg}$, that at each episode $n$ leverages the data acquired thus far, i.e., $\mathcal{D}_{1:n-1}$, to determine a policy $\pi_n \in \Pi$
> > > > > > > for the next step of data collection.
> > > > > > >
> > > > > > > That is, $\textit{Alg}(\mathcal{D}_{1:n-1}, n) \to \pi_n$.
> > > > > > > The collected data is then used to learn an estimate $\mu_n$ of the true dynamics $f^*$. The objective of this work is to propose an algorithm that is consistent, i.e., $\mu_n (z) \to f^*(z)$ with $n \to \infty$ for all $z \in \mathcal{R}$, where $\mathcal{R}$ is the reachability set, i.e.,
> > > > > > > $\mathcal{R}= \\{ z \in \mathcal{Z} \mid \exists \pi \in \Pi, t \leq T, \text { s.t. }, p\left(z_t=z \mid \pi, f^*\right)>0 \\}$,
> > > > > > > and give rates on how fast the estimate $\mu_n$ converges to the true dynamics $f^*$.
> > > > > > >
> > > > > > > **Bound on zero-shot performance**: To keep the response short, we give a quick description of our approach/result. Let $\hat{\pi}$ denote our proposed policy and $\pi^*$ the optimal one. We show that if the rewards satisfy Lipschitz continuity property then $|J(\pi^*) - J(\hat{\pi}) |  \leq \mathcal{O}(\epsilon)$, after we have run the algorithm for long enough such that the uncertainty is reduced to less than $\epsilon$. Here, $J(\pi)$ represents the performance of the policy $\pi$ on the control task. To show this, we consider the RKHS setting and leverage Lemma 5 from H-UCRL [Curi et al, 2020]. Once we have globally reduced the uncertainty to less than $\epsilon$, we can replace the RHS of the equation in [Lemma 5, Curi et al 2022] with $\epsilon$. We also give how many samples are required to achieve less than $\epsilon$ uncertainty with OpAx using our result on sample complexity from Theorem 2.
> > > > > > >
> > > > > > > **Clarification in the experiments**: In our experiments, we change line 254, ''we consider several tasks''  to we consider several dynamical systems for which we study multiple tasks, i.e., different rewards but with the same dynamics.
> > > > > > >
> > > > > > > We hope we could address the reviewer's concerns adeptly and are happy to answer any further questions.

---

> > > > > > > > ### Comment · Reviewer_V3Yd · 2023-08-21
> > > > > > > >
> > > > > > > > Thank the authors for the response. I have raised my score. The updated presentation is better compared to the original manuscript, and more clearly shows the motivation of the paper. However, it could be better if the analysis and problem statement can be formatted from the multi-task performance perspective. It is still not that clear why we need a globally accurate model unless in a multi-task setting. I don't think it's a hard step to connect the model's accuracy to the multi-task performance. I encourage the authors to do so in a later version of their paper.

---

> > > > > > > > > ### Author Response · Authors · 2023-08-21
> > > > > > > > > **Acknowledgement of reviewer V3Yd**
> > > > > > > > >
> > > > > > > > > We thank the reviewer for their valuable feedback and for increasing our score.
> > > > > > > > > In the paper, we focus the problem statement on reducing model epistemic uncertainty globally as this is also the objective of other works on active/curious exploration. Nonetheless, if the reviewer prefers we are happy to move the proposition on the bound of zero-shot performance for Lipschitz continuous rewards to the main paper.

---

> > > > > > > > > > ### Comment · Reviewer_V3Yd · 2023-08-21
> > > > > > > > > >
> > > > > > > > > > That would be great. Along with this, I also encourage the authors to more clearly state the zero-shot performance objective in the problem statement section, instead of only the model's accuracy.

---

### Official Review · Reviewer_meUw · 2023-07-20

**Soundness:** 3 good
**Presentation:** 3 good
**Contribution:** 3 good
**Rating:** 6
**Confidence:** 3

**Summary:**

This paper proposes a task-agnostic active exploration algorithm for non-linear dynamic systems as long as it can be well calibrated. By combining the optimistic exploration principle and some standard baysian techinques, they give a general convergence bound as well as the more specfic bound in gaussian process case. Besides the theoretical proofs, they also provide several downstream experiments, showing their algorihthm can achieve better performance.

**Strengths:**

1. This paper gives the first theoretical gaurantees on general non-linear dynamic models, which is significant.
2. The bayesian-based framework is somewhat novel and can be extended in many cases.
3. Their theoretical analysis are solid. Their results on guassian process model helps reader to further understand this problem.
4. I am not familiar with control experiments, but according to what stated in their paper, their approach give nontrivial improvements.

**Weaknesses:**

1. There is no discussion on the computational complexity. For eqn.(7), it is unclear to me how to efficiently solve $\pi$, $\eta$ when the policy space is large or the $T$ is large.
2. The techinque itself is not very surprising to me. Exploration by optimism is a widely used in all the RL related paper, and the estimation on confidence bound seems to me just a standard derivation from exsiting techiniques.
3. There is no dicussion on the lower-bound, therefore it is hard for me to understand how good this result is.

But I am willing to raise my scores if my questions are addressed, or there are some techinical difficulty I am missing, or someone else want to advocate contribution of the experiment results.

**Questions:**

Main questions:
1. Can you give more dicussion on the computational complexity. Or maybe give me some examples of exsiting oracles?
2. Currently paper aims to upper bound the maximum expected information gain (or the epistemic uncertainty). While I believe it makes sense, it is hard for me to connect that with previous paper. For example, [Wagenmaker and Jamieson, 2020] gives the upper bound ob $||\hat{A} - A^*||$, which is very easy to understand. Can you show what this maximum expected information gain implies in those simple models so reader can have more intuition on your results?
3. Can you give some discussions on the lower bound?
4. Can you give more dicussions on Assumpton 2 (line 114-115) and Line 160 "Incorporating constraints Equation (7) can impose input constraints by considering a restrictive class of policies $\Pi$. "  I understand this is one standard assumption. But theoretically, it is still unclear to me how difficult it is to lift this assumption in order to get the generalization result, and how it will affect the computational complexity.

---

> ### Author Rebuttal · Authors · 2023-08-07
>
> Thank you for your feedback, our response follows.
>
> ## Weaknesses
> *W1 Computational complexity*:  Solving an optimization problem for general nonlinear systems is challenging, and out of scope for this work. For our problem formulation, standard trajectory optimizers such as iLQR[1], iCEM[2] or policy optimizers such as BPTT [3], and SAC [4] can be and have been commonly used. We also provide a heuristic variant of our algorithm in Appendix C.2.2., which, unlike the theoretical variant, does not scale with the dimension of the state space. This heuristic variant was used for the experiments in the “Fetch, Pick & Place Construction” environment. We have added this explanation in the updated version of the paper.
>
> *W2 Novelty of technique*: As we highlight in the general summary above, the novelty of our method lies in leveraging ideas from Bayesian experiment design, and the OFU paradigm, to provide novel and first-of-its-kind analysis for active learning. We relate the regret of planning under unknown dynamics and the information gained over a whole trajectory, to give a novel sample complexity bound for active/curious exploration. To our knowledge, we are the first to do so. This lack of theoretical results is also acknowledged by concurrent works [6, 7].
>
> *W3 There is no discussion on the lower bound*: Providing lower bounds is a challenging open research problem. However, in more restrictive classes of problems [7, 8, 9] there exist nearly minimax optimal lower bounds, which compared to the upper bound, only vary in constants and logarithmic terms (wrt to N). In our setting, we would expect a similar outcome, but we acknowledge that this is yet to be solved as future work.
>
> ## Additional Questions:
>
> *Q1 Intuition of the upper bound*: We are happy to give some intuition. Mainly, for the general RKHS setting we give a bound on how fast the epistemic uncertainty decays (Theorem 2).
> In Lemma 2, we show that with high probability for all $j \in \{1, \dots, d_x}$
>
> $$
> |\mu_{n, j}(z) - f_j^*(z)|  \leq \beta_n(\delta) \sigma_{n, j}(z) \leq \mathcal{O}\left(\sqrt{\frac{\gamma^{T+1}_n}{n}}\right)
> $$
>
> Therefore, the decay rate of the epistemic uncertainty quantifies how fast our mean estimate converges to the true function. Our result holds for general RKHS kernels. Example for the linear kernel (with known matrix B): $f^*(z) = Ax + Bu; \mu_n(z) = \hat{A}x + Bu$.
> Then our bound is of the following sort:
>
> $$
> ||\mu_{n, j}(z) - f^*(z)||_1 = ||(\hat{A} - A)x||_1 \leq \mathcal{O}\left(\sqrt{\frac{d^{T} \log^T(n)}{n}}\right)
> $$
>
> for all $x$. Note, this bound holds for a more general setting than the one considered in [Wagenmaker and Jamieson, 2020]. Furthermore, under stronger assumptions on the noise (Gaussian, instead of sub-Gaussian), we can avoid the exponential dependence on $T$ (see Theorem 4 Appendix B for more detail).
>
> *Q2 Constraints on inputs*: For input constraints like $||a|| \leq a_{\max}$, we can restrict our search to policies that only map to actions between $[-a_{\max}, a_{\max}]$. Most RL optimizers, such as SAC, squash actions to a fixed range.
>
> Having addressed all of the questions provided by the reviewer, and given the contributions of this paper, we would appreciate it if the reviewer would increase their score for our paper. We would be happy to answer any remaining questions or concerns.
>
> ## References
> [1] Li, Weiwei, and Emanuel Todorov (2004). Iterative linear quadratic regulator design for nonlinear biological movement systems. First International Conference on Informatics in Control, Automation and Robotics.
>
> [2] Pinneri, C., et al (2021). Sample-efficient cross-entropy method for real-time planning. Conference on Robot Learning.
>
> [3] Clavera, I. et al (2020). Model-augmented actor-critic: Backpropagating through paths. arXiv preprint arXiv:2005.08068 .
>
> [4] Haarnoja, T., et al (2018). Soft actor-critic: Off-policy maximum entropy deep reinforcement learning with a stochastic actor. International conference on machine learning.
>
> [5] Kakade, S., et al (2020). Information theoretic regret bounds for online nonlinear control. Advances in Neural Information Processing Systems.
>
> [6] Chakraborty, S. et al (2023). STEERING: Stein Information Directed Exploration for Model-Based Reinforcement Learning. arXiv preprint arXiv:2301.12038.
>
> [7] Wagenmaker, A. et al (2023). Optimal Exploration for Model-Based RL in Nonlinear Systems. arXiv preprint arXiv:2306.09210.
>
> [8] Wagenmaker, A. et al (2020). Active learning for identification of linear dynamical systems. In Conference on Learning Theory.
>
> [9] Scarlett, J., et al (2017). Lower bounds on regret for noisy gaussian process bandit optimization. Conference on Learning Theory.

---

> > ### Comment · Reviewer_meUw · 2023-08-17
> >
> > Thanks for clarify on my additional question.
> >
> > I agree this paper has some solid contributions, including a novel problem formulation, some sound proofs and corresponding experiment results as supplementary to their computional problem. Meanwhile, I still think 1\ the OFU techiniques are not novel enough 2\ Although there are many paper only consider sample complexity instead of computational complexity, but they all have substantial novelty on their techinques while this paper not.
> >
> > Based on this, I raise my score to 5 because both pros and cons are obvious. It depends on AC to judge which shall weight more. If this paper get accepted, I will suggest add some explaination including the one mentioned in Q1 and the exponential horizon thing so readers can better understand this result in the overal literature.

---

> > > ### Author Response · Authors · 2023-08-17
> > > **Follow up on reviewer's comment**
> > >
> > > We thank you for increasing our score. Following the suggestion, we have added the additional explanation in the updated version of the paper.
> > >
> > > As recognized, the paper has solid contributions including *being the first one* to give sample complexity of learning unknown dynamical systems with continuous state and action spaces. This is recognized as an open problem ([1, Paragraph 1 in the Introduction] [2, Paragraph 2 in the Related Work]).
> > >
> > > The OFU technique has been used in several settings such as analyzing cumulative regret. However, it has not been used for model-based active exploration of dynamical systems with continuous state and action spaces.
> > > Particularly, in our setting, the pure OFU technique is not enough to prove sample complexity. Our active exploration objective is integral to the theoretical analysis. The objective we propose is the *information gained over a trajectory/rollout*, which we simplify (to a practical intrinsic reward) in Lemma 1. *This has also not been done by prior work*. The combination of our objective and the OFU principle gives the SOTA sample complexity guarantees. Our theoretical analysis goes beyond just applying the OFU principle and the proving technique is of independent interest for the active exploration community. Accordingly, we believe that our technique has substantial novelty beyond just the OFU principle.
> > >
> > > Given your second comment, we are curious about which papers you are referring to. Since, as we highlight above, we (as well as [1, 2]) are not aware of any sample complexity bounds for our general case.
> > >
> > > We hope to have addressed your concerns regarding our contribution and would appreciate a reevaluation of our work.
> > >
> > >
> > >
> > >
> > >
> > >
> > >
> > >
> > > [1] Chakraborty, S. et al (2023). STEERING: Stein Information Directed Exploration for Model-Based Reinforcement Learning. arXiv preprint arXiv:2301.12038.
> > >
> > > [2] Wagenmaker, A. et al (2023). Optimal Exploration for Model-Based RL in Nonlinear Systems." arXiv preprint arXiv:2306.09210.

---

> > > > ### Comment · Area_Chair_5jrX · 2023-08-18
> > > >
> > > > Dear authors,
> > > >
> > > > As a point of clarification, are you able to give an indication of the computation time (empirically) needed by the various algorithms under comparison, e.g. on average per episode or iteration over the training process? (or did I miss that somewhere in the paper or appendix?). If you provide an indication, please add what hardware was used and whether this was the same for all algorithms under consideration.
> > > >
> > > > -the AC

---

> > > > > ### Author Response · Authors · 2023-08-18
> > > > > **Response to the AC's comments**
> > > > >
> > > > > Dear AC,
> > > > >
> > > > > We highly appreciate your active engagement in our rebuttal process and thank you accordingly.
> > > > >
> > > > > We do not report the computation times. Instead, we report the hyperparameters used for each of the algorithms in the Appendix. The hyperparameters such as optimization steps and gradient steps are identical for each algorithm. Since the training and optimization procedure for all the algorithms is identical (all require learning a Bayesian model and optimizing a policy), they all are allocated equal computational budgets.
> > > > >
> > > > > We hope this clarifies the question.
> > > > >
> > > > > Thanks,
> > > > > Authors

---

### Official Review · Reviewer_kKQP · 2023-07-20

**Soundness:** 3 good
**Presentation:** 2 fair
**Contribution:** 2 fair
**Rating:** 6
**Confidence:** 4

**Summary:**

This paper proposes and studies a rather intuitive algorithm for active learning in nonlinear dynamical systems in the episodic setting. They establish consistency (in terms of mutual information) and provide supporting numerical experiments.

**Strengths:**

* The proposed algorithm is intuitive and it is satisfying that it "works" (in terms of consistency).

* The paper is generally quite well-written and I did not have (m)any issues in terms of clarity and level of writing.

* The exact setting is relatively novel and well-motivated (some caveats below) and the question is interesting and definitely deserves further study.

* The experiments look relatively thorough (but I am not the best person to judge their significance).


**Weaknesses:**


* While I am sympathetic to the fact it can be hard to keep track of everything appearing in ML conferences,  unfortunately the manuscript does miss some of the most closely related recent references especially when it comes to learning in nonlinear dynamical systems or general mixing processes. My concern here is that missing these makes the contribution of the present paper appear larger than it actually is, since the below references [A,B,C,D] also treat learning in rather general dynamical systems/time-series.  In this light, the last part of the stated contributions (cf. line 52) are  somewhat misleading as it states that related work is not general enough to treat the present system dynamics. However, [A,B,C,D] are actually general enough (or at least very close to general enough) and the authors might want to complement their references  listed starting from line 50 (which certainly aren't the most general setups considered in the recent literature). In particular [B,C] also explicitly study RKHS-like dynamical systems.


[A] Roy, Abhishek, Krishnakumar Balasubramanian, and Murat A. Erdogdu. "On empirical risk minimization with dependent and heavy-tailed data." Advances in Neural Information Processing Systems 34 (2021): 8913-8926.

[B] Ziemann, Ingvar M., Henrik Sandberg, and Nikolai Matni. "Single trajectory nonparametric learning of nonlinear dynamics." conference on Learning Theory. PMLR, 2022.

[C] Ziemann, Ingvar, and Stephen Tu. "Learning with little mixing." Advances in Neural Information Processing Systems, 2022.

[D] Li, Yingcong, et al. "Transformers as algorithms: Generalization and stability in in-context learning." International Conference on Machine Learning. 2023.

* As the above references do not treat the active learning setting, this limitation can easily be overcome by including the above references and making the qualifying distinction that the contributions are not "first in system identification/supervised learning of nonlinear systems" but rather first in terms of experiment design/active learning.

* The exponential dependence on the horizon in the main results appears overly pessimistic to me and I question whether the the bound is informative at all in any meaningful setting. While convergence is not guaranteed in terms of MI the above references achieve convergence even from a single trajectory, at least indicating at a glance that such a dependence ought to be removable. This exponential dependence on the horizon in the provided bounds is a major caveat here---I have therefore estimated the theoretical results to be asymptotic consistency results and not bona fide finite sample guarantees.


**Questions:**

* Could the authors comment on the necessity of the exponential dependence on the horizon in their main result? It would be interesting to see if this indeed necessary or whether it can be overcome by an improved proof method/algo.

* Is the dependence on $\delta$ hidden in the main results (and if so what is it)? Seems a little strange to me as the result is claimed to hold wp at least $1-\delta$.

* Are there any interesting cases where $ L \times \max \beta \leq 1$ or is the exponential dependence always present?

**Limitations:**

N/A.

---

> ### Author Rebuttal · Authors · 2023-08-07
>
> We thank the reviewer for their feedback and for providing additional references.
> As the reviewer highlighted, the shared references do not consider the challenging active learning/unsupervised learning setting and focus more on the supervised learning problem. We have included the references in the revised paper as suggested
>
> ## Weaknesses
> *W1. Exponential dependency on horizon T in regret bound*: The exponential dependency may be removed, however, in accordance to prior work, we believe it will require further assumptions (cf. [1] last paragraph, page 7). For instance, restricting the noise to be sampled from a Gaussian distribution (as opposed to sub-Gaussian) gives a $\mathcal{O}(\beta_N\sqrt{T\frac{\gamma_N}{N}})$ bound in the RKHS setting. This analysis was presented in Appendix B, Theorem 4. In the revised paper, we moved this result to the main text.
>
> *W2. Comparison to the single trajectory setting*: The single trajectory works mostly deal with linear systems that are stable or assume knowledge of a stabilizing controller [2]. We make no such assumptions. Works that first learn a stabilizing controller from the data, also suffer from an exponential growth in the cost until a controller is learned [3]. Our general setting will require stronger assumptions to avoid the exponential dependence (see response W1.). We also emphasize that we give a worst-case theoretical upper bound. In practice, the epistemic uncertainty decreases much faster as depicted in Figure 1 in the paper.
>
> ## Additional Questions
> *Q1 Dependence on $\delta$*: The calibration factor $\beta_n$ depends on $\delta$ (see definition 2). For the RKHS case, the dependence of $\beta_n$ on $\delta$ is well studied ( $\beta_n \propto \sqrt{\log(1/\delta)}$) [e.g. in 4].
>
>
> Having addressed your concerns we kindly ask you to consider revising our score. For any remaining questions, we are happy to provide further clarification.
>
> ## References
> [1] Curi, S., Berkenkamp, F., and Krause, A. (2020). Efficient model-based reinforcement learning through optimistic policy search and planning. Advances in Neural Information Processing Systems.
>
> [2] Simchowitz, M. and Foster, D. (2020). Naive exploration is optimal for online lqr. In International Conference on Machine Learning, pages 8937–8948. PMLR.
>
> [3] Chen, Xinyi, and Elad Hazan. "Black-box control for linear dynamical systems." In Conference on Learning Theory, pp. 1114-1143. PMLR, 2021.
>
> [4] Chowdhury, S.R., et al (2017). On kernelized multi-armed bandits. International Conference on Machine Learning.

---

> > ### Comment · Reviewer_kKQP · 2023-08-10
> >
> > I have read the rebuttal and am happy to raise my score to a 6.
> >
> > In general I think bounds with exponential time dependencies are prohibitive so I am glad that the authors showcase their result removing this into the main body---even though it appears restricted to the Gaussian setting. Although I have not verified it sounds a little to me like this exponential dependence is driven by a lack of hypercontractivity/log-concavity for general subgaussians. I'd be curious to hear the authors' thoughts on this/what drives this dependence.

---

> > > ### Author Response · Authors · 2023-08-10
> > >
> > > We thank the reviewer for increasing our score.
> > >
> > > Indeed, this could be an explanation. Moreover, together with the dynamics and noise we induce a stochastic process that represents the evolution of the dynamical system. This process does not satisfy any contraction properties, i.e., in general, the system may be unstable.
> > >
> > > When analyzing our bound; a key quantity that we study is how the two trajectories (optimistic and true) evolve over a horizon of $T$. In general, this can diverge exponentially with horizon $T$. However, for the special case of Gaussian noise due to its support on the whole domain and because of the boundedness of our exploration objective (epistemic uncertainty is always upper-bounded) we can perform a tighter analysis similar to [1].
> > >
> > > [1] Kakade, S., et al (2020). Information theoretic regret bounds for online nonlinear control. Advances in Neural Information Processing Systems.
> > >
> > > We are happy to answer further questions.

---

### Author Rebuttal · Authors · 2023-08-07

We thank the reviewers for their feedback. It seems that some key contributions to our work have missed the attention of our reviewers. We have made our contributions more clear in the revised paper and included additional references. We highlight our contributions below for clarification.

1. We consider the problem of learning a dynamical system in a reward-free/task-agnostic manner over *continuous state and action spaces*. We adhere to the general textbook definition of a dynamical system, e.g. [1].

2. We introduce a practical exploration objective, based on the concept of information in Bayesian experiment design. Our *novel* derivation technique may be of *independent interest* in active learning and experiment design.

3. We utilize the principle of optimism in the face of uncertainty (OFU) and give a PAC bound on the epistemic uncertainty of any visited trajectory (Theorem 2).

4. To our knowledge, this paper is the first to present active learning guarantees on *continuous domains* and for *generic dynamical systems in an RKHS*. Prior work is limited to strictly simpler classes of systems such as finite, linear, and low-rank MDPs [2, 3, 4, 5], linear systems [6], and nonlinear systems with finite-dimensional feature spaces [7]. In fact, concurrent work on model-based RL [8, 9] explicitly highlights the *lack of theoretical guarantees* within the context of curious/active exploration of general dynamical systems. Our paper directly targets this gap.

5. We validate our method extensively over 6 RL tasks, to demonstrate that *in addition to enjoying strong theoretical guarantees*, OPAX performs on par with the state-of-the-art baselines, none of which are supported by theory.

We have revised our Related Works section and added additional experiments in the Appendix to demonstrate the benefits of active learning and optimism, as asked for by our reviewers. Furthermore, some of our reviewers raised concerns about the exponential dependence of our bound on the horizon $T$. In accordance with prior work [10], we believe stronger assumptions are needed to alleviate this. We had discussed this in Appendix B, where in Theorem 4 we give a tighter bound (polynomial in $T$) under additional assumptions on the noise. We have now moved the theorem to the main paper for more visibility.

We would be happy to further update the paper, if there are any remaining questions or feedback by the reviewers.

## References
[1] Khalil, H. K. (2015). Nonlinear control, volume 406. Pearson New York.

[2] Jin, C. et al (2020). Reward-free exploration for reinforcement learning. International Conference on Machine Learning.

[3] Tarbouriech, J. et al (2020). Active model estimation in markov decision processes. Conference on Uncertainty in Artificial Intelligence.

[4] Wagenmaker, Andrew J (2022). et al. Reward-free rl is no harder than reward-aware rl in linear markov decision processes.

[5] Chen, J. et al (2023). On the statistical efficiency of reward-free exploration in non-linear rl. Advances in Neural Information Processing Systems.

[6] Wagenmaker, A. et al (2020). Active learning for identification of linear dynamical
systems. In Conference on Learning Theory.

[7] Mania, H. et al (2022). Active learning for nonlinear system identification with guarantees. The Journal of Machine Learning Research.

[8] Chakraborty, S. et al (2023). STEERING: Stein Information Directed Exploration for Model-Based Reinforcement Learning. arXiv preprint arXiv:2301.12038.

[9] Wagenmaker, A. et al (2023). Optimal Exploration for Model-Based RL in Nonlinear Systems." arXiv preprint arXiv:2306.09210.

[10] Curi, S., Berkenkamp, F., and Krause, A. (2020). Efficient model-based reinforcement learning through optimistic policy search and planning. Advances in Neural Information Processing Systems.

---

> ### Comment · Area_Chair_5jrX · 2023-08-18
>
> Dear authors,
>
> Thanks for your rebuttal. Unfortunately, not all reviewers have responded to your rebuttal yet. I just wanted to let you know that I'll keep trying to get the to reply to your rebuttal. In any case, the points raised in the rebuttal will be taken into account in the private reviewer-AC discussion and in the final decision making.
>
> -The AC

---

### Decision · Program_Chairs · 2023-09-21

**Decision:**

Accept (poster)

**Comment:**

There was quite a discussion about this paper in relation to ‘zero shot’ RL methods. The submitted paper does not explicitly maximize for future zero-shot performance (rather, for learning a model), which has its own strengths and weaknesses. However, the introduction does seem to claim zero-shot performance of the method, without evaluating this aspect against existing methods. The authors should review such (implied) claims to make them in line with the paper's actual results.

There was also a discussion on novelty. Several reviewers argued the method is not very novel. However, the authors argue that it is the analysis presented that is novel and valuable. On the contribution, there were also questions about the usefulness of the bound and the strength of the connection between theory and practice. The authors pointed at a result from the appendix avoiding exponential dependence, and gave an indication of what a lower bound might look like, alleviating the concerns on this aspect.